# UNMASKING TRANSFORMERS: A THEORETICAL APPROACH TO DATA RECOVERY VIA ATTENTION WEIGHTS

## ABSTRACT

In the realm of deep learning, transformers have emerged as a dominant architecture, particularly in natural language processing tasks. However, with their widespread adoption, concerns regarding the security and privacy of the data processed by these models have arisen. In this paper, we address a pivotal question: Can the data fed into transformers be recovered using their attention weights and outputs? We introduce a theoretical framework to tackle this problem. Specifically, we present an algorithm that aims to recover the input data $X \in \mathbb{R}^{d \times n}$ from given attention weights $W = QK^\top \in \mathbb{R}^{d \times d}$ and output $B \in \mathbb{R}^{n \times n}$ by minimizing the loss function $L(X)$. This loss function captures the discrepancy between the expected output and the actual output of the transformer. Our findings have significant implications for the Localized Layer-wise Mechanism (LLM), suggesting potential vulnerabilities in the model's design from a security and privacy perspective. This work underscores the importance of understanding and safeguarding the internal workings of transformers to ensure the confidentiality of processed data.

## 1 INTRODUCTION

In the intricate and constantly evolving domain of deep learning, the transformer architecture has emerged as a game-changing innovation Vaswani et al. (2017). This novel architecture has propelled the state-of-the-art performance in a myriad of tasks, and its potency lies in the underlying mechanism known as the "attention mechanism." The essence of this mechanism can be distilled into its unique interaction between three distinct matrices: the **Query** ($Q$), the **Key** ($K$), and the **Value** ($V$), where the **Query** matrix ($Q$) represents the questions or the aspects we're interested in, the **Key** matrix ($K$) denotes the elements against which these questions are compared or matched, and the he **Value** matrix ($V$) encapsulates the information we want to retrieve based on the comparisons. These matrices are not just mere multidimensional arrays; they play vital roles in encoding, comparing, and extracting pertinent information from the data.

Given this context, the attention mechanism can be mathematically captured as follows:

**Definition 1.1** (Attention matrix computation)**.** *Let* $Q, K \in \mathbb{R}^{n \times d}$ *be two matrices that respectively represent the query and key. Similarly, for a matrix* $V \in \mathbb{R}^{n \times d}$ *denoting the value, the attention matrix is defined as*

$$\mathrm{Att}(Q, K, V) := D^{-1} A V,$$

*In this equation, two matrices are introduced:* $A \in \mathbb{R}^{n \times n}$ *and* $D \in \mathbb{R}^{n \times n}$*, defined as:*

$$A := \exp(QK^\top) \ \text{ and } \ D := \mathrm{diag}(A\mathbf{1}_n).$$

Here, the matrix $A$ represents the relationship scores between the query and key, and $D$ ensures normalization, ensuring that the attention weights sum to one. The computation hence, deftly combines these relationships with the value matrix to output the final attended representation.

In practical large-scale language models ChatGPT (2022); OpenAI (2023), there might be multi-levels of the attention computation. For those multi-level architecture, the feed-forward can be represented

as

$$\underbrace{X_{\ell+1}^\top}_{n \times d} \leftarrow \underbrace{D(X_\ell)^{-1} \exp(X_\ell^\top Q_\ell K_\ell X_\ell)}_{n \times n} \underbrace{X_\ell^\top}_{n \times d} \underbrace{V_\ell}_{d \times d}$$

where $X_\ell$ is the input of $\ell$-th layer, and $X_{\ell+1}$ is the output of $\ell$-th layer, and $Q_\ell, K_\ell, V_\ell$ are the attention weights in $\ell$-th layer.

This architecture has particularly played a pivotal role in driving progress across various sub-disciplines of natural language processing (NLP). It has profoundly influenced sectors such as machine translation Firat et al. (2016); Choi et al. (2018), sentiment analysis Usama et al. (2020); Naseem et al. (2020), language modeling Martin et al. (2019), and even the generation of creative text ChatGPT (2022); OpenAI (2023). This trajectory of influence is most prominently embodied by the creation and widespread adoption of Large Language Models (LLMs) like GPT Radford et al. (2018) and BERT Devlin et al. (2018). These models, along with their successive versions, e.g., GPT-2 Radford et al. (2019), GPT-3 Brown et al. (2020), PaLM Chowdhery et al. (2022), OPT Zhang et al. (2022), are hallmarks in the field due to their staggering number of parameters and complex architectural designs. These LLMs have achieved unparalleled performance levels, setting new standards in machine understanding and automated text generation ChatGPT (2022); OpenAI (2023). Moreover, their emergence has acted as a catalyst for rethinking what algorithms are capable of, spurring new lines of inquiry and scrutiny within both academic and industrial circles Ray (2023). As these LLMs find broader application across an array of sectors, gaining a thorough understanding of their intricate internal mechanisms is evolving from a topic of scholarly interest into a crucial requirement for their effective and responsible deployment.

Yet, the very complexity and architectural sophistication that propel the success of transformers come with a host of consequential challenges, making their effective and responsible usage nontrivial. Prominent among these challenges is the overarching imperative of ensuring data security and privacy Pan et al. (2020); Brown et al. (2022); Kandpal et al. (2022). Within the corridors of the research community, an increasingly pertinent question is emerging regarding the inherent vulnerabilities of these architectures. Specifically,

*is it possible to know the input data by analyzing the attention weights and model outputs?*

To put it in mathematical terms, given a language model represented as $Y = f(W; X)$, if one has access to the output $Y$ and the attention weights $W$, is it possible to mathematically invert the model to obtain the original input data $X$?

Addressing this line of inquiry extends far beyond the realm of academic speculation; it has direct and significant implications for practical, real-world applications. This is especially true when these transformer models interact with data that is either sensitive in nature, like personal health records Cascella et al. (2023), or proprietary, as in the financial sector Wu et al. (2023). With the broader deployment of Large Language Models (LLMs) into environments that adhere to stringent data confidentiality regulations, the mandate for achieving absolute data security becomes unequivocally critical. In this work, we aim to delve deeply into this paramount issue, striving to offer a nuanced understanding of these potential vulnerabilities while suggesting pathways for ensuring safety in the development, training, and utilization of transformer technologies.

In this study, we address a distinct problem that differs from the conventional task of finding optimal weights for a given input and output. Specifically, we assume that the weights are already known, and our objective is to invert the input to recover the original data. The key focus of our investigation lies in identifying the conditions under which successful inversion of the original input is feasible. This problem holds significant relevance in the context of addressing security concerns associated with attention networks.

To provide a formal definition of our training objective for data recovery, we aim to optimize a specific criterion that enables effective inversion of the input. By formulating and solving this objective, we aim to gain valuable insights into the security implications and vulnerabilities of attention networks.

**Definition 1.2** (Regression model). *Given the attention weights $W = KQ^\top \in \mathbb{R}^{d \times d}$, $V \in \mathbb{R}^{d \times d}$ and output $B \in \mathbb{R}^{n \times d}$, the goal is find $X \in \mathbb{R}^{d \times n}$ such that*

$$L(X) := \| \underbrace{D(X)^{-1} \exp(X^\top W X)}_{n \times n} \underbrace{X^\top}_{n \times d} \underbrace{V}_{d \times d} - \underbrace{B}_{n \times d} \|_F^2$$

*where*

- $D(X) = \mathrm{diag}(\exp(X^\top W X)\mathbf{1}_n) \in \mathbb{R}^{n \times n}$

Figure 1: Visualization of our loss function.

In order to establish an understanding of attacking on the above model, we present our main result in the following section.

## 1.1 OUR RESULT

We state our result as follows:

**Theorem 1.3** (Informal version of Theorem J.1). *Given a model with several layers of attention. For each layer, we have parameters $Q \in \mathbb{R}^{d \times d}, K \in \mathbb{R}^{d \times d}, V \in \mathbb{R}^{d \times d}$. We denote $W := KQ^\top$. Given a desired output $B \in \mathbb{R}^{d \times n}$, then we can denote the training data input*

$$X^* = \arg\min_X \|D(X)^{-1} \exp(X^\top W X) X^\top V - B\|_F^2 + L_{\mathrm{reg}}$$

*Next, we choose a good initial point $X_0$ that is close enough to $X^*$. Assume that there exists a scalar $R > 1$ such that $\|W\|_F \leq R$, $\|V\|_F \leq R$, $|b_{i,j}| \leq R$ where $b_{i,j}$ denotes the $i,j$-th entry of $B$ for all $i \in [n], j \in [d]$.*

*Then, for any accuracy parameter $\epsilon \in (0, 0.1)$ and a failure probability $\delta \in (0, 0.1)$, an algorithm based on the Newton method can be employed to recover the initial data. The result of this algorithm guarantee within $T = O(\log(\|X_0 - X^*\|_F / \epsilon))$ executions, it outputs a matrix $\widetilde{X} \in \mathbb{R}^{d \times n}$ satisfying $\|\widetilde{X} - X^*\|_F \leq \epsilon$ with a probability of at least $1 - \delta$.*

**Roadmap.** We arrange the rest of our paper as follows. In Section 2 we present some works related our topic. In Section 3 we provide a preliminary for our work. In Section 4, we state an overview of our techniques, summarizing the method we use to recover data via attention weights. We conclude our work and propose some future directions in Section 5.

## 2 RELATED WORKS

**Attention Computation Theory.** Following the rise of LLM, numerous studies have emerged on attention computation Kitaev et al. (2020); Tay et al. (2020); Chen et al. (2021); Zandieh et al. (2023); Tarzanagh et al. (2023); Sanford et al. (2023); Panigrahi et al. (2023a); Zhang et al. (2020a); Arora & Goyal (2023); Tay et al. (2021); Deng et al. (2023b). LSH techniques approximate attention, and based on them, the KDEformer offers a notable dot-product attention approximation Zandieh et al. (2023). Recent works Alman & Song (2023); Brand et al. (2023); Deng et al. (2023c) explored diverse attention computation methods and strategies to enhance model efficiency. On the optimization front,

Zhang et al. (2020b) highlighted that adaptive methods excel over SGD due to heavy-tailed noise distributions. Other insights include the emergence of the KTIW property Snell et al. (2021) and various regression problems inspired by attention computation Gao et al. (2023a); Li et al. (2023c;b), revealing deeper nuances of attention models.

**Security concerns about LLM.** Amid LLM advancements, concerns about misuse have arisen Pan et al. (2020); Brown et al. (2022); Kandpal et al. (2022); Kirchenbauer et al. (2023); Vyas et al. (2023); Chu et al. (2023); Xu et al. (2023); Gao et al. (2023c); Kirchenbauer et al. (2023); He et al. (2022a;b). Pan et al. (2020) assesses the privacy risks of capturing sensitive data with eight models and introduces defensive strategies, balancing performance and privacy. Brown et al. (2022) asserts that current methods fall short in guaranteeing comprehensive privacy for language models, recommending training on publicly intended text. Kandpal et al. (2022) reveals that the vulnerability of large language models to privacy attacks is significantly tied to data duplication in training sets, emphasizing that deduplicating this data greatly boosts their resistance to such breaches. Kirchenbauer et al. (2023) devised a way to watermark LLM output without compromising quality or accessing LLM internals. Meanwhile, Vyas et al. (2023) introduced near access-freeness (NAF), ensuring generative models, like transformers and image diffusion models, don't closely mimic copyrighted content by over $k$-bits.

**Inverting the neural network.** Originating from the explosion of deep learning, there have been a series of works focused on inverting the neural network Jensen et al. (1999); Lu et al. (1999); Mahendran & Vedaldi (2015); Dosovitskiy & Brox (2016); Zhang et al. (2020d). Jensen et al. (1999) surveys various techniques for neural network inversion, which involves finding input values that produce desired outputs, and highlights its applications in query-based learning, sonar performance analysis, power system security assessment, control, and codebook vector generation. Lu et al. (1999) presents a method for inverting trained neural networks by formulating the problem as a mathematical programming task, enabling various network inversions and enhancing generalization performance.. Mahendran & Vedaldi (2015) explores the reconstruction of image representations, including CNNs, to assess the extent to which it's possible to recreate the original image, revealing that certain layers in CNNs retain accurate visual information with varying degrees of geometric and photometric invariance. Zhang et al. (2020d) presents a novel generative model-inversion attack method that can effectively reverse deep neural networks, particularly in the context of face image reconstruction, and explores the connection between a model's predictive ability and vulnerability to such attacks while noting limitations in using differential privacy for defense.

**Attacking the Neural Networks.** During the development of artificial intelligence, there have been many works on attaching the neural networks Zhu et al. (2019); Wei et al. (2020); Rigaki & Garcia (2020); Huang et al. (2020); Yin et al. (2021); Huang et al. (2021b). Several studies Zhu et al. (2019); Wei et al. (2020); Rigaki & Garcia (2020); Yin et al. (2021) have warned that local training data can be compromised using only exchanged gradient information. These methods start with dummy data and gradients, and through gradient descent, they empirically show that the original data can be fully reconstructed. A follow-up study Zhao et al. (2020) specifically focuses on classification tasks and finds that the real labels can also be accurately recovered. Other types of attacks include membership and property inference Shokri et al. (2017); Melis et al. (2019), the use of Generative Adversarial Networks (GANs) Hitaj et al. (2017); Goodfellow et al. (2014), and additional machine-learning techniques McPherson et al. (2016); Papernot et al. (2016). A recent paper Wang et al. (2023) uses tensor decomposition for gradient leakage attacks but is limited by its inefficiency and focus on over-parametrized networks.

**Theoretical Approaches to Understanding LLMs.** Recent strides have been made in understanding and optimizing regression models using various activation functions. Research on over-parameterized neural networks has examined exponential and hyperbolic activation functions for their convergence properties and computational efficiency Gao et al. (2023a); Li et al. (2023c); Deng et al. (2023b); Gao et al. (2023c); Li et al. (2023a). Modifications such as regularization terms and algorithmic innovations, like a convergent approximation Newton method, have been introduced to enhance their performance Li et al. (2023c); Deng et al. (2022). Studies have also leveraged tensor tricks to vectorize regression models, allowing for advanced Lipschitz and time-complexity analyses Gao et al. (2023b); Deng et al. (2023a). Simultaneously, the field is seeing innovations in

optimization algorithms tailored for LLMs. Techniques like block gradient estimators have been employed for huge-scale optimization problems, significantly reducing computational complexity Cai et al. (2021). Unique approaches like Direct Preference Optimization bypass the need for reward models, fine-tuning LLMs based on human preference data Rafailov et al. (2023). Additionally, advancements in second-order optimizers have relaxed the conventional Lipschitz Hessian assumptions, providing more flexibility in convergence proofs Liu et al. (2023). Also, there is a series of work on understanding fine-tuning Malladi et al. (2023a;b); Panigrahi et al. (2023b). Collectively, these theoretical contributions are refining our understanding and optimization of LLMs, even as they introduce new techniques to address challenges such as non-guaranteed Hessian Lipschitz conditions.

**Optimization and Convergence of Deep Neural Networks.** Prior research Li & Liang (2018); Du et al. (2018); Allen-Zhu et al. (2019a;b); Arora et al. (2019a;b); Song & Yang (2019); Cai et al. (2019); Zhang et al. (2019); Cao & Gu (2019); Zou & Gu (2019); Oymak & Soltanolkotabi (2020); Ji & Telgarsky (2019); Lee et al. (2020); Huang et al. (2021a); Zhang et al. (2020c); Brand et al. (2020); Zhang et al. (2020a); Song et al. (2021); Alman et al. (2023); Munteanu et al. (2022); Zhang (2022); Gao et al. (2023a); Li et al. (2023c); Qin et al. (2023) on the optimization and convergence of deep neural networks has been crucial in understanding their exceptional performance across various tasks. These studies have also contributed to enhancing the safety and efficiency of AI systems. In Gao et al. (2023a) they define a neural function using an exponential activation function and apply the gradient descent algorithm to find optimal weights. In Li et al. (2023c), they focus on the exponential regression problem inspired by the attention mechanism in large language models. They address the non-convex nature of standard exponential regression by considering a regularization version that is convex. They propose an algorithm that leverages input sparsity to achieve efficient computation. The algorithm has a logarithmic number of iterations and requires nearly linear time per iteration, making use of the sparsity of the input matrix.

## 3 Preliminary

In this section, we present the preliminary concepts and introductions to the background of our research that form the foundation of our paper. We begin by introducing the notations we utilize in Section 3.1. In Section 3.2, we introduce a solid method to attack neural networks by inverting their weights and outputs. In Section 3.3, we use a regression form to simplify the training process when transformer implements back-propagation.

### 3.1 Notations

We used $\mathbb{R}$ to denote real numbers. We use $A \in \mathbb{R}^{n \times d}$ to denote an $n \times d$ size matrix where each entry is a real number. For any positive integer $n$, we use $[n]$ to denote $\{1, 2, \cdots, n\}$. For a matrix $A \in \mathbb{R}^{n \times d}$, we use $a_{i,j}$ to denote the an entry of $A$ which is in $i$-th row and $j$-th column of $A$, for each $i \in [n]$, $j \in [d]$. We use $A_{i,j} \in \mathbb{R}^{n \times d}$ to denote a matrix such that all of its entries equal to $0$ except for $a_{i,j}$. We use $\mathbf{1}_n$ to denote a length-$n$ vector where all the entries are ones. For a vector $w \in \mathbb{R}^n$, we use $\mathrm{diag}(w) \in \mathbb{R}^{n \times n}$ denote a diagonal matrix where $(\mathrm{diag}(w))_{i,i} = w_i$ and all other off-diagonal entries are zero. Let $D \in \mathbb{R}^{n \times n}$ be a diagonal matrix, we use $D^{-1} \in \mathbb{R}^{n \times n}$ to denote a diagonal matrix where $i$-th entry on diagonal is $D_{i,i}$ and all the off-diagonal entries are zero. Given two vectors $a, b \in \mathbb{R}^n$, we use $(a \circ b) \in \mathbb{R}^n$ to denote the length-$n$ vector where $i$-th entry is $a_i b_i$. For a matrix $A \in \mathbb{R}^{n \times d}$, we use $A^\top \in \mathbb{R}^{d \times n}$ to denote the transpose of matrix $A$. For a vector $x \in \mathbb{R}^n$, we use $\exp(x) \in \mathbb{R}^n$ to denote a length-$n$ vector where $\exp(x)_i = \exp(x_i)$ for all $i \in [n]$. For a matrix $X \in \mathbb{R}^{n \times n}$, we use $\exp(X) \in \mathbb{R}^{n \times n}$ to denote matrix where $\exp(X)_{i,j} = \exp(X_{i,j})$. For any matrix $A \in \mathbb{R}^{n \times d}$, we define $\|A\|_F := (\sum_{i=1}^n \sum_{j=1}^d A_{i,j}^2)^{1/2}$. For a vector $a, b \in \mathbb{R}^n$, we use $\langle a, b \rangle$ to denote $\sum_{i=1}^n a_i b_i$.

### 3.2 Model Inversion Attack

A model inversion attack is a type of adversarial attack in which a malicious user attempts to recover the private dataset used to train a supervised machine learning model . The goal of a model inversion attack is to generate realistic and diverse samples that accurately describe each class in the private dataset.

The attacker typically has access to the trained model and can use it to make predictions on input data . By carefully crafting input data and observing the model's predictions, the attacker can infer information about the training data.

Model inversion attacks can be a significant privacy concern, as they can potentially reveal sensitive information about individuals or organizations. These attacks exploit vulnerabilities in the model's behavior and can be used to extract information that was not intended to be disclosed.

Model inversion attacks can be formulated as an optimization problem. Given the output $Y$, the model function $f_\theta$ with parameters $\theta$, and the loss function $\mathcal{L}$, the objective of a model inversion attack is to find an input data $X^*$ that minimizes the loss between the model's prediction $f_\theta(X)$ and the target output $Y$. Mathematically, this can be expressed as:

$$X^* = \arg\min_X \mathcal{L}(f_\theta(X), Y)$$

Since the loss function $\mathcal{L}(f_\theta(X), Y)$ is convex with respect to optimizing $X$, we can employ a specific method for model inversion attack, which involves the following steps:

1. Initialize an input data $X$.
2. Compute the gradient $\nabla_X \mathcal{L}(f_\theta(X), Y)$.
3. Optimize $X$ using a learning rate $\eta$ by updating $X = X - \eta \nabla_X \mathcal{L}(f_\theta(X), Y)$.

This iterative process aims to find an input $X$ that minimizes the loss between the model's prediction and the target output. By updating $X$ in the direction opposite to the gradient, the attack can potentially converge to an input that generates a prediction close to the desired output, thereby inverting the model. In this work, we focus our effort on the Attention models (which is natural due to the explosive development of LLMs). In this case, the parameters $\theta$ in our model are considered to consist of $\{Q, K, V\}$. During the script, to avoid the abuse of notations, we use $B = Y$ to denote the ground truth label.

### 3.3 Regression Problem Inspired by Attention Computation

In this paper, we extend the prior work of Gao et al. (2023b) and focus on the training process of the attention mechanism in the context of the Transformer model. We decompose the training procedure into a regression form based on the insights provided by Deng et al. (2023b).

Specifically, we investigate the training process for a specific layer, denoted as the $l$-th layer, and consider the case of single-headed attention. In this setting, we have an input matrix represented as $X \in \mathbb{R}^{d \times n}$ and a target matrix denoted as $B \in \mathbb{R}^{d \times n}$. Given $Q \in \mathbb{R}^{d \times d}, K \in \mathbb{R}^{d \times d}, V \in \mathbb{R}^{d \times d}$ as the trained weights of attention architecture. The objective of the training process in the Transformer model is to minimize the loss function by utilizing back-propagation.

The loss function, denoted as $L(X)$, is defined as follows:

$$L(X) = \|D^{-1} \exp(X^\top K^\top Q X) X^\top V - B\|_F^2,$$

where $D := \mathrm{diag}(\exp(X^\top K^\top Q X) \mathbf{1}_n)$ and each row of $D^{-1} \exp()$ corresponds to a softmax function.

The goal of minimizing this loss function is to align the predicted output, obtained by applying the attention mechanism, with the target matrix $B$.

## 4 Recovering Data via Attention Weights

In this section, we propose our theoretical method to recover the training data from trained transformer weights and outputs. Besides, we solve our method by proving hessian of our training objective is Lipschitz-continuous and positive definite. In Section 4.1, we provide a detailed description of our approach. In Section 4.3, we show our result that proving hessian of training objective is Lipschitz-continuous. In Section 4.4, we show our result that the hessian of training objective is positive definite.

### 4.1 Training Objective of Attention Inversion Attack

In this study, we propose a novel technique for inverting the attention weights of a transformer model using Hessian decomposition. Our aim is to find the input $X \in \mathbb{R}^{d \times n}$ that minimizes the Frobenius norm of the difference between $D(X)^{-1} \exp(X^\top W X)V$ and $B$, where $W = KQ^\top \in \mathbb{R}^{d \times d}$ represents the attention weights, $B \in \mathbb{R}^{n \times d}$ is the desired output, and $D(X) = \mathrm{diag}(\exp(X^\top W X)) \in \mathbb{R}^{n \times n}$ is a diagonal matrix.

To achieve this, we introduce an algorithm that minimizes the loss function $L(X)$, defined as follows:

$$L(X) := \|D(X)^{-1} \exp(X^\top W X)X^\top V - B\|_F^2 + L_{\text{reg}}, \tag{1}$$

where $V \in \mathbb{R}^{d \times d}$ is a matrix of values, and $L_{\text{reg}}$ captures any additional regularization terms. This loss function quantifies the discrepancy between the expected output and the actual output of the transformer.

In our approach, we leverage Hessian decomposition to efficiently compute the Hessian matrix and apply a second-order method to approximate the optimal input $X$. By utilizing the Hessian, we can gain insights into the curvature of the loss function and improve the efficiency of optimization. This approach enables us to efficiently find an approximate solution for the input $X$ that minimizes the loss function, thereby inverting the attention weights of the transformer model.

By integrating Hessian decomposition and second-order optimization techniques (Anstreicher (2000); Lee et al. (2019); Cohen et al. (2019); Jiang et al. (2021); Huang et al. (2022); Gu & Song (2022); Gu et al. (2023)), our proposed algorithm provides a promising approach for addressing the challenging task of inverting attention weights in transformer models.

Due to the complexity of the loss function (Eq. (1)), directly computing its Hessian is challenging or even impossible. To simplify the computation, we introduce several notations (See Figure 2 for visualization):

$$\text{Exponential Function: } u(X)_i := \exp(X^\top W X_{*,i})$$
$$\text{Sum of Softmax: } \alpha(X)_i := \langle u(X)_i, \mathbf{1}_n \rangle$$
$$\text{Softmax Probability: } f(X)_i := \alpha(X)_i^{-1} u(X)_i$$
$$\text{Value Function: } h(X)_j := X^\top V_{*,j}$$
$$\text{One-unit Loss Function: } c(X)_{i,j} := \langle f(X)_i, h(X)_j \rangle - b_{i,j}.$$

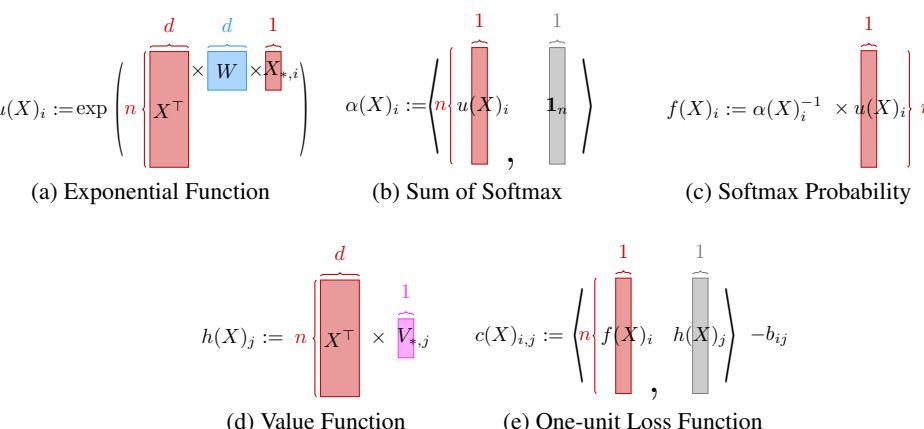

(a) Exponential Function     (b) Sum of Softmax     (c) Softmax Probability

(d) Value Function     (e) One-unit Loss Function

Figure 2: Visualization of Notations We Defined

Using these terms, we can express the loss function $L(X)$ as the sum over all elements:

$$L(X) = \sum_{i=1}^{n} \sum_{j=1}^{d} (c(X)_{i,j})^2$$

This allows us to break down the computation into several steps. Specifically, we start by computing the gradients of the predefined terms. Given two integers $i_0 \in [n]$ and $j_0 \in [d]$, we define $c(X)_{i_0,j_0}$ as a matrix where all entries are zero except for the entry $c_{i_0,j_0}$. Additionally, we denote $i_1 \in [n]$ and $j_1 \in [d]$ as two other integers, and use $x_{i_1,j_1}$ to represent the entry in $X$ corresponding to the $i_1$-th row and $j_1$-th column.

We can now express $\frac{\mathrm{d}c(X)_{i_0,j_0}}{\mathrm{d}x_{i_1,j_1}}$ (the gradient of $c(X)_{i_0,j_0}$) in two cases:

- *Case 1:* The situation when $i_0 = i_1$.
- *Case 2:* The situation when $i_0 \neq i_1$.

By decomposing the Hessian into several cases (See Section F for details), we can calculate the final Hessian. Similar to the approach used when computing the gradients, we introduce two additional integers $i_2 \in [n]$ and $j_2 \in [d]$. The Hessian can then be expressed as $\frac{\mathrm{d}^2 c(X)_{i_0,j_0}}{\mathrm{d}x_{i_1,j_1}\mathrm{d}x_{i_2,j_2}}$. We can further break down the computation into four cases to handle different scenarios:

- *Case 1:* The situation when $i_0 = i_1 = i_2$.
- *Case 2:* The situation when $i_0 = i_1 \neq i_2$.
- *Case 3:* The situation when $i_0 \neq i_1$, $i_0 \neq i_2$ and $i_1 = i_2$.
- *Case 4:* The situation when $i_0 \neq i_1$, $i_0 \neq i_2$ and $i_1 \neq i_2$.

It is worth mentioning that there is a case that $i_0 \neq i_1$, $i_0 = i_2$, is equivalent to the case that $i_0 = i_1 \neq i_2$. By considering these four cases, we can calculate the Hessian for each element in $X$. This allows us to gain further insights into the curvature of the loss function and optimize the parameters more effectively.

### 4.2 HESSIAN DECOMPOSITION

By considering different conditions of Hessian, we have the following decomposition.

**Definition 4.1** (Hessian of functions of matrix). *We define the Hessian of $c(X)_{i_0,j_0}$ by considering its Hessian with respect to $x = \mathrm{vec}(X)$. This means that, $\nabla^2 c(X)_{i_0,j_0}$ is a $nd \times nd$ matrix with its $i_1 \cdot j_1, i_2 \cdot j_2$-th entry being $\frac{\mathrm{d}c(X)_{i_0,j_0}}{\mathrm{d}x_{i_1,j_2}x_{i_2,j_2}}$.*

**Definition 4.2** (Hessian split). *We split the hessian of $c(X)_{i_0,j_0}$ into following cases*

- $i_0 = i_1 = i_2 : H_1^{(i_1,i_2)}$

- $i_0 = i_1, i_0 \neq i_2 : H_2^{(i_1,i_2)}$

- $i_0 \neq i_1, i_0 = i_2 : H_3^{(i_1,i_2)}$

- $i_0 \neq i_1, i_0 \neq i_2 : H_4^{(i_1,i_2)}$

*In above, $H_i^{(i_1,i_2)}$ is a $d \times d$ matrix with its $j_1, j_2$-th entry being $\frac{\mathrm{d}c(X)_{i_0,j_0}}{\mathrm{d}x_{i_1,j_2}x_{i_2,j_2}}$.*

Utilizing above definitions, we split the Hessian to a $n \times n$ partition with its $i_1, i_2$-th component being $H_i(i_1, i_2)$.

**Definition 4.3.** *We define $\nabla^2 c(X)_{i_0,j_0}$ to be as following*

$$\nabla^2 c(X)_{i_0,j_0} = \begin{bmatrix} H_4^{(1,1)} & H_4^{(1,2)} & H_4^{(1,3)} & \cdots & H_3^{(1,i_0)} & \cdots & H_4^{(1,n)} \\ H_4^{(2,1)} & H_4^{(2,2)} & H_4^{(2,3)} & \cdots & H_3^{(2,i_0)} & \cdots & H_4^{(2,n)} \\ H_4^{(3,1)} & H_4^{(3,2)} & H_4^{(3,3)} & \cdots & H_3^{(3,i_0)} & \cdots & H_4^{(3,n)} \\ \vdots & \vdots & \vdots & \ddots & \vdots & \ddots & \vdots \\ H_2^{(i_0,1)} & H_2^{(i_0,2)} & H_2^{(i_0,3)} & \cdots & H_1^{(i_0,i_0)} & \cdots & H_2^{(i_0,n)} \\ \vdots & \vdots & \vdots & \ddots & \vdots & \ddots & \vdots \\ H_4^{(n,1)} & H_4^{(n,2)} & H_4^{(n,3)} & \cdots & H_3^{(n,i_0)} & \cdots & H_4^{(n,n)} \end{bmatrix}$$

### 4.3 HESSIAN OF $L(X)$ IS LIPSCHITZ- CONTINUOUS

We present our findings that establish the Lipschitz continuity property of the Hessian of $L(X)$, which is a highly desirable characteristic in optimization. This property signifies that the second derivatives of $L(X)$ exhibit smooth changes within a defined range. Leveraging this Lipschitz property enables us to employ gradient-based methods with guaranteed convergence rates and enhanced stability. Consequently, our results validate the feasibility of utilizing the proposed training objective to achieve convergence in the model inversion attack. This finding holds significant promise for the development of efficient and effective optimization strategies in this context.

**Lemma 4.4** (informal version of Lemma H.10). *Under following conditions*

- *Assumption G.1 (bounded parameter) holds*

- *Let $c(X)_{i_0,j_0}$ be defined as Definition B.8*

*For $X, Y \in \mathbb{R}^{d \times n}$, we have*

$$\|\nabla^2 L(X) - \nabla^2 L(Y)\| \leq O(n^3 d^3 R^{10}) \|X - Y\|_F$$

### 4.4 HESSIAN OF $L(X)$ IS POSITIVE DEFINITE

After computing the Hessian of $L(X)$, we now show our result that can confirm it is positive definite under proper regularization. Therefore, we can apply a modified Newton's method to approach the optimal solution.

**Lemma 4.5** (PSD bounds for $\nabla^2 L(X)$). *Under following conditions,*

- *Let $L(X)$ be defined as in Definition B.9*

- *Let Assumption G.1 (bounded parameter) be satisfied*

*we have*

$$\nabla^2 L(X) \succeq -O(ndR^8) \cdot \mathbf{I}_{nd}$$

Therefore, we define the regulatization term as follows to have the PSD guarantee.

**Definition 4.6** (Regularization). *Let $\gamma = O(-ndR^8)$, we define*

$$L_{\text{reg}}(X) := \gamma \cdot \|\text{vec}(X)\|_2^2$$

With above properties of the loss function, we have the convergence result in Theorem 1.3.

## 5 CONCLUSION AND FUTURE DISCUSSION

In this study, we have presented a theoretical approach for inverting input data using weights and outputs. Our investigation delved into the mathematical frameworks that underpin the attention mechanism, with the aim of determining whether knowledge of attention weights and model outputs could enable the reconstruction of sensitive information from the input data. The insights gained from this research are intended to deepen our understanding and facilitate the development of more secure and robust transformer models. By doing so, we strive to foster responsible and ethical advancements in the field of deep learning.

This work lays the groundwork for future research and development aimed at fortifying transformer technologies against potential threats and vulnerabilities. Our ultimate goal is to enhance the safety and effectiveness of these groundbreaking models across a wide range of applications. By addressing potential risks and ensuring the integrity of sensitive information, we aim to create a more secure and trustworthy environment for the deployment of transformer models.

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
