**Roadmap.** We arrange the appendix as follows. In Section A, we provide several preliminary notations. In Section B we provide details of computing the gradients. In Section C and Section D we provide detail of computing Hessian for two cases. In Section E we show how to split the Hessian matrix. In Section F we combine the results before and compute the Hessian for the loss function. In Section G we bound the basic functions to be used later. In Section H we provide proof for the Lipschitz property of the loss function. We provide our final result in Section J.

## A    NOTATIONS

We used $\mathbb{R}$ to denote real numbers. We use $A \in \mathbb{R}^{n \times d}$ to denote an $n \times d$ size matrix where each entry is a real number. For any positive integer $n$, we use $[n]$ to denote $\{1, 2, \cdots, n\}$. For a matrix $A \in \mathbb{R}^{n \times d}$, we use $a_{i,j}$ to denote the an entry of $A$ which is in $i$-th row and $j$-th column of $A$, for each $i \in [n]$, $j \in [d]$. We use $A_{i,j} \in \mathbb{R}^{n \times d}$ to denote a matrix such that all of its entries equal to $0$ except for $a_{i,j}$. We use $\mathbf{1}_n$ to denote a length-$n$ vector where all the entries are ones. For a vector $w \in \mathbb{R}^n$, we use $\mathrm{diag}(w) \in \mathbb{R}^{n \times n}$ denote a diagonal matrix where $(\mathrm{diag}(w))_{i,i} = w_i$ and all other off-diagonal entries are zero. Let $D \in \mathbb{R}^{n \times n}$ be a diagonal matrix, we use $D^{-1} \in \mathbb{R}^{n \times n}$ to denote a diagonal matrix where $i$-th entry on diagonal is $D_{i,i}$ and all the off-diagonal entries are zero. Given two vectors $a, b \in \mathbb{R}^n$, we use $(a \circ b) \in \mathbb{R}^n$ to denote the length-$n$ vector where $i$-th entry is $a_i b_i$. For a matrix $A \in \mathbb{R}^{n \times d}$, we use $A^\top \in \mathbb{R}^{d \times n}$ to denote the transpose of matrix $A$. For a vector $x \in \mathbb{R}^n$, we use $\exp(x) \in \mathbb{R}^n$ to denote a length-$n$ vector where $\exp(x)_i = \exp(x_i)$ for all $i \in [n]$. For a matrix $X \in \mathbb{R}^{n \times n}$, we use $\exp(X) \in \mathbb{R}^{n \times n}$ to denote matrix where $\exp(X)_{i,j} = \exp(X_{i,j})$. For any matrix $A \in \mathbb{R}^{n \times d}$, we define $\|A\|_F := (\sum_{i=1}^n \sum_{j=1}^d A_{i,j}^2)^{1/2}$. For a vector $a, b \in \mathbb{R}^n$, we use $\langle a, b \rangle$ to denote $\sum_{i=1}^n a_i b_i$.

## B    GRADIENTS

Here in this section, we provide analysis for the gradient computation. In Section B.1 we state some facts to be used. In Section B.2 we provide some definitions. In Sections B.3, B.4, B.5, B.6, B.7, B.8 and B.9 we compute the gradient for the terms defined respectively. Finally in Section B.10 we compute the gradient for $L(X)$.

### B.1    FACTS

**Fact B.1** (Basic algebra). *We have*

- $\langle u, v \rangle = \langle v, u \rangle = u^\top v = v^\top u.$

- $\langle u \circ v, w \rangle = \langle u \circ v \circ w, \mathbf{1}_n \rangle$

- $u^\top (v \circ w) = u^\top \mathrm{diag}(v) w$

**Fact B.2** (Basic calculus rule). *We have*

- $\frac{\mathrm{d}\langle f(x), g(x) \rangle}{\mathrm{d}t} = \langle \frac{\mathrm{d}f(x)}{\mathrm{d}t}, g(x) \rangle + \langle f(x), \frac{\mathrm{d}g(x)}{\mathrm{d}t} \rangle$ *(here t can be any variable)*

- $\frac{\mathrm{d}y^z}{\mathrm{d}x} = z \cdot y^{z-1} \frac{\mathrm{d}y}{\mathrm{d}x}$

- $u \cdot v = v \cdot u$

- $\frac{\mathrm{d}x}{\mathrm{d}x_j} = e_j$ *where $e_j$ is a vector that only $j$-th entry is $1$ and zero everywhere else.*

- *Let $x \in \mathbb{R}^d$, let $y \in \mathbb{R}$ be independent of $x$, we have $\frac{\mathrm{d}x}{\mathrm{d}y} = \mathbf{0}_d$.*

- *Let $f(x), g(x) \in \mathbb{R}$, we have $\frac{\mathrm{d}(f(x)g(x))}{\mathrm{d}t} = \frac{\mathrm{d}f(x)}{\mathrm{d}t} g(x) + f(x) \frac{\mathrm{d}g(x)}{\mathrm{d}t}$*

- *Let $x \in \mathbb{R}$, $\frac{\mathrm{d}}{\mathrm{d}x} \exp(x) = \exp(x)$*

- *Let $f(x) \in \mathbb{R}^n$, we have $\frac{\mathrm{d}\exp(f(x))}{\mathrm{d}t} = \exp(f(x)) \circ \frac{\mathrm{d}f(x)}{\mathrm{d}t}$*

### B.2 DEFINITIONS

**Definition B.3** (Simplified notations)**.** *We have following definitions*

- *We use $u(X)_{i_0,i_1}$ to denote the $i_1$-th entry of $u(X)_{i_0}$.*

- *We use $f(X)_{i_0,i_1}$ to denote the $i_1$-th entry of $f(X)_{i_0}$.*

- *We define $W_{j_1,*}$ to denote the $j_1$-th row of $W$. (In the proof, we treat $W_{j_1,*}$ as a column vector).*

- *We define $W_{*,j_1}$ to denote the $j_1$-th column of $W$.*

- *We define $w_{j_1,j_0}$ to denote the scalar equals to the entry in $j_1$-th row, $j_0$-th column of $W$.*

- *We define $V_{*,j_1}$ to denote the $j_1$-th column of $V$.*

- *We define $v_{j_1,j_0}$ to denote the scalar equals to the entry in $j_1$-th row, $j_0$-th column of $V$.*

- *We define $X_{*,i_0}$ to denote the $i_0$-th column of $X$.*

- *We define $x_{i_1,j_1}$ to denote the scalar equals to the entry in $i_1$**-th column,** $j_1$**-th row** of $X$.*

**Definition B.4** (Exponential function $u$)**.** *If the following conditions hold*

- *Let $X \in \mathbb{R}^{d \times n}$*

- *Let $W \in \mathbb{R}^{d \times d}$*

*For each $i_0 \in [n]$, we define $u(X)_{i_0} \in \mathbb{R}^n$ as follows*

$$u(X)_{i_0} = \exp(X^\top W X_{*,i_0})$$

**Definition B.5** (Sum function of softmax $\alpha$)**.** *If the following conditions hold*

- *Let $X \in \mathbb{R}^{d \times n}$*

- *Let $u(X)_{i_0}$ be defined as Definition B.4*

*We define $\alpha(X)_{i_0} \in \mathbb{R}$ for all $i_0 \in [n]$ as follows*

$$\alpha(X)_{i_0} = \langle u(X)_{i_0}, \mathbf{1}_n \rangle$$

**Definition B.6** (Softmax probability function $f$)**.** *If the following conditions hold*

- *Let $X \in \mathbb{R}^{d \times n}$*

- *Let $u(X)_{i_0}$ be defined as Definition B.4*

- *Let $\alpha(X)_{i_0}$ be defined as Definition B.5*

*We define $f(X)_{i_0} \in \mathbb{R}^n$ for each $i_0 \in [n]$ as follows*

$$f(X)_{i_0} := \alpha(X)_{i_0}^{-1} u(X)_{i_0}$$

**Definition B.7** (Value function $h$)**.** *If the following conditions hold*

- *Let $X \in \mathbb{R}^{d \times n}$*

- *Let $V \in \mathbb{R}^{d \times d}$*

*We define $h(X)_{j_0} \in \mathbb{R}^n$ for each $j_0 \in [n]$ as follows*

$$h(X)_{j_0} := X^\top V_{*,j_0}$$

**Definition B.8** (One-unit loss function $c$)**.** *If the following conditions hold*

- *Let $f(X)_{i_0}$ be defined as Definition B.6*

- *Let $h(X)_{j_0}$ be defined as Definition B.7*

*We define $c(X) \in \mathbb{R}^{n \times d}$ as follows*

$$c(X)_{i_0,j_0} := \langle f(X)_{i_0}, h(X)_{j_0} \rangle - b_{i_0,j_0}, \forall i_0 \in [n], j_0 \in [d]$$

**Definition B.9** (Overall function $L$). *If the following conditions hold*

- *Let $c(X)_{i_0,j_0}$ be defined as Definition B.8*

*We define $L(X) \in \mathbb{R}$ as follows*

$$L(X) := \sum_{i_0=1}^{n} \sum_{j_0=1}^{d} (c(X)_{i_0,j_0})^2$$

## B.3 Gradient for each column of $X^\top W X_{*,i_0}$

**Lemma B.10.** *We have*

- **Part 1.** *Let $i_0 = i_1 \in [n]$, $j_1 \in [d]$*

$$\underbrace{\frac{\mathrm{d} X^\top W X_{*,i_0}}{\mathrm{d} x_{i_1,j_1}}}_{n \times 1} = \underbrace{e_{i_0}}_{n \times 1} \cdot \underbrace{\langle W_{j_1,*}, X_{*,i_0} \rangle}_{\text{scalar}} + \underbrace{X^\top}_{n \times d} \underbrace{W_{*,j_1}}_{d \times 1}$$

- **Part 2** *Let $i_0 \neq i_1 \in [n]$, $j_1 \in [d]$*

$$\underbrace{\frac{\mathrm{d} X^\top W X_{*,i_0}}{\mathrm{d} x_{i_1,j_1}}}_{n \times 1} = \underbrace{e_{i_1}}_{n \times 1} \cdot \underbrace{\langle W_{j_1,*}, X_{*,i_0} \rangle}_{\text{scalar}}$$

*Proof.* **Proof of Part 1.**

$$\underbrace{\frac{\mathrm{d} X^\top W X_{*,i_0}}{\mathrm{d} x_{i_1,j_1}}}_{n \times 1} = \underbrace{\frac{\mathrm{d} X^\top}{\mathrm{d} X_{i_1,j_1}}}_{n \times d} \underbrace{W}_{d \times d} \underbrace{X_{*,i_0}}_{d \times 1} + \underbrace{X^\top}_{n \times d} \underbrace{W}_{d \times d} \underbrace{\frac{\mathrm{d} X_{*,i_0}}{\mathrm{d} X_{i_1,j_1}}}_{d \times 1}$$

$$= \underbrace{e_{i_1}}_{n \times 1} \underbrace{e_{j_1}^\top}_{1 \times d} \underbrace{W}_{d \times d} \underbrace{X_{*,i_0}}_{d \times 1} + \underbrace{X^\top}_{n \times d} \underbrace{W}_{d \times d} \underbrace{e_{j_1}}_{d \times 1}$$

$$= \underbrace{e_{i_1}}_{n \times 1} \cdot \underbrace{\langle W_{j_1,*}, X_{*,i_0} \rangle}_{\text{scalar}} + \underbrace{X^\top}_{n \times d} \underbrace{W_{*,j_1}}_{d \times 1}$$

$$= \underbrace{e_{i_0}}_{n \times 1} \cdot \underbrace{\langle W_{j_1,*}, X_{*,i_0} \rangle}_{\text{scalar}} + \underbrace{X^\top}_{n \times d} \underbrace{W_{*,j_1}}_{d \times 1}$$

where the 1st step follows from Fact B.2, the 2nd step follows from simple derivative rule, the 3rd is simple algebra, the 4th step ie because $i_0 = i_1$.

**Proof of Part 2**

$$\underbrace{\frac{\mathrm{d} X^\top W X_{*,i_0}}{\mathrm{d} x_{i_1,j_1}}}_{n \times 1} = \underbrace{\frac{\mathrm{d} X^\top}{\mathrm{d} x_{i_1,j_1}}}_{n \times d} \underbrace{W}_{d \times d} \underbrace{X_{*,i_0}}_{d \times 1} + \underbrace{X^\top}_{n \times d} \underbrace{W}_{d \times d} \underbrace{\frac{\mathrm{d} X_{*,i_0}}{\mathrm{d} x_{i_1,j_1}}}_{d \times 1}$$

$$= \underbrace{e_{i_1}}_{n \times 1} \underbrace{e_{j_1}^\top}_{1 \times d} \underbrace{W}_{d \times d} \underbrace{X_{*,i_0}}_{d \times 1} + \underbrace{X^\top}_{n \times d} \underbrace{W}_{d \times d} \underbrace{\mathbf{0}_d}_{d \times 1}$$

$$= \underbrace{e_{i_1}}_{n \times 1} \cdot \underbrace{\langle W_{j_1,*}, X_{*,i_0} \rangle}_{\text{scalar}}$$

where the 1st step follows from Fact B.2, the 2nd step follows from simple derivative rule, the 3rd is simple algebra. □

### B.4 GRADIENT FOR $u(X)_{i_0}$

**Lemma B.11.** *Under following conditions*

- *Let $u(X)_{i_0}$ be defined as Definition B.4*

*We have*

- **Part 1.** *For each $i_0 = i_1 \in [n]$, $j_1 \in [d]$*

$$\underbrace{\frac{\mathrm{d}u(X)_{i_0}}{\mathrm{d}x_{i_1,j_1}}}_{n \times 1} = u(X)_{i_0} \circ (e_{i_0} \cdot \langle W_{j_1,*}, X_{*,i_0} \rangle + X^\top W_{*,j_1})$$

- **Part 2** *For each $i_0 \neq i_1 \in [n]$, $j_1 \in [d]$*

$$\underbrace{\frac{\mathrm{d}u(X)_{i_0}}{\mathrm{d}x_{i_1,j_1}}}_{n \times 1} = \underbrace{u(X)_{i_0}}_{n \times 1} \circ (e_{i_1} \cdot \langle W_{j_1,*}, X_{*,i_0} \rangle)$$

*Proof.*

**Proof of Part 1**

$$\underbrace{\frac{\mathrm{d}u(X)_{i_0}}{\mathrm{d}x_{i_1,j_1}}}_{n \times 1} = \underbrace{\frac{\mathrm{d}\exp(X^\top W X_{*,i_0})}{\mathrm{d}x_{i_1,j_1}}}_{n \times 1}$$

$$= \exp(\underbrace{X^\top}_{n \times d} \underbrace{W}_{d \times d} \underbrace{X_{*,i_0}}_{d \times 1}) \circ \underbrace{\frac{\mathrm{d}X^\top W X_{*,i_0}}{\mathrm{d}x_{i_1,j_1}}}_{n \times 1}$$

$$= \underbrace{u(X)_{i_0}}_{n \times 1} \circ \underbrace{\frac{\mathrm{d}X^\top W X_{*,i_0}}{\mathrm{d}x_{i_1,j_1}}}_{n \times 1}$$

$$= \underbrace{u(X)_{i_0}}_{n \times 1} \circ (\underbrace{e_{i_0}}_{n \times 1} \cdot \underbrace{\langle W_{j_1,*}, X_{*,i_0} \rangle}_{\text{scalar}} + \underbrace{X^\top}_{n \times d} \underbrace{W_{*,j_1}}_{d \times 1})$$

where the 1st step and the 3rd step follow from Definition of $u(X)_{i_0}$ (see Definition B.4), the 2nd step follows from Fact B.2, the 4th step follows by Lemma B.10.

**Proof of Part 2**

$$\underbrace{\frac{\mathrm{d}u(X)_{i_0}}{\mathrm{d}x_{i_1,j_1}}}_{n \times 1} = \underbrace{\frac{\mathrm{d}\exp(X^\top W X_{*,i_0})}{\mathrm{d}x_{i_1,j_1}}}_{n \times 1}$$

$$= \exp(\underbrace{X^\top}_{n \times d} \underbrace{W}_{d \times d} \underbrace{X_{*,i_0}}_{d \times 1}) \circ \underbrace{\frac{\mathrm{d}X^\top W X_{*,i_0}}{\mathrm{d}x_{i_1,j_1}}}_{n \times 1}$$

$$= \underbrace{u(X)_{i_0}}_{n \times 1} \circ \underbrace{\frac{\mathrm{d}X^\top W X_{*,i_0}}{\mathrm{d}x_{i_1,j_1}}}_{n \times 1}$$

$$= \underbrace{u(X)_{i_0}}_{n \times 1} \circ (\underbrace{e_{i_1}}_{n \times 1} \cdot \underbrace{\langle W_{j_1,*}, X_{*,i_0} \rangle}_{\text{scalar}})$$

where the 1st step and the 3rd step follow from Definition of $u(X)_{i_0}$ (see Definition B.4), the 2nd step follows from Fact B.2, the 4th step follows by Lemma B.10.

$\square$

### B.5 Gradient Computation for $\alpha(X)_{i_0}$

**Lemma B.12** (A generalization of Lemma 5.6 in Deng et al. (2023b)). *If the following conditions hold*

- *Let $\alpha(X)_{i_0}$ be defined as Definition B.5*

*Then, we have*

- **Part 1.** *For each $i_0 = i_1 \in [n]$, $j_1 \in [d]$*

$$\underbrace{\frac{\mathrm{d}\alpha(X)_{i_0}}{\mathrm{d}x_{i_1,j_1}}}_{\text{scalar}} = u(X)_{i_0,i_0} \cdot \langle W_{j_1,*}, X_{*,i_0} \rangle + \langle u(X)_{i_0}, X^\top W_{*,j_1} \rangle$$

- **Part 2.** *For each $i_0 \neq i_1 \in [n]$, $j_1 \in [d]$*

$$\underbrace{\frac{\mathrm{d}\alpha(X)_{i_0}}{\mathrm{d}x_{i_1,j_1}}}_{\text{scalar}} = u(X)_{i_0,i_1} \cdot \langle W_{j_1,*}, X_{*,i_0} \rangle$$

*Proof.* **Proof of Part 1.**

$$\underbrace{\frac{\mathrm{d}\alpha(X)_{i_0}}{\mathrm{d}x_{i_1,j_1}}}_{\text{scalar}} = \underbrace{\frac{\mathrm{d}\langle u(X)_{i_0}, \mathbf{1}_n \rangle}{\mathrm{d}x_{i_1,j_1}}}_{\text{scalar}}$$

$$= \langle \underbrace{\frac{\mathrm{d}u(X)_{i_0}}{\mathrm{d}x_{i_1,j_1}}}_{n \times 1}, \underbrace{\mathbf{1}_n}_{n \times 1} \rangle$$

$$= \langle \underbrace{u(X)_{i_0}}_{n \times 1} \circ (e_{i_0} \cdot \langle W_{j_1,*}, X_{*,i_0} \rangle + X^\top W_{*,j_1}), \underbrace{\mathbf{1}_n}_{n \times 1} \rangle$$

$$= \langle \underbrace{u(X)_{i_0}}_{n \times 1} \circ e_{i_0}, \mathbf{1}_n \rangle \cdot \langle W_{j_1,*}, X_{*,i_0} \rangle + \langle u(X)_{i_0} \circ (X^\top W_{*,j_1}), \underbrace{\mathbf{1}_n}_{n \times 1} \rangle$$

$$= \langle \underbrace{u(X)_{i_0}}_{n \times 1}, e_{i_0} \rangle \cdot \langle W_{j_1,*}, X_{*,i_0} \rangle + \langle u(X)_{i_0}, X^\top W_{*,j_1} \rangle$$

$$= u(X)_{i_0,i_0} \cdot \langle W_{j_1,*}, X_{*,i_0} \rangle + \langle u(X)_{i_0}, X^\top W_{*,j_1} \rangle$$

where the 1st step follows from the definition of $\alpha(X)_{i_0}$ (see Definition B.5), the 2nd step follows from Fact B.2, the 3rd step follows from Lemma B.11, the 4th step is rearrangement, the 5th step is derived by Fact B.1, the last step is by the definition of $U(X)_{i_0,i_0}$.

**Proof of Part 2.**

$$\underbrace{\frac{\mathrm{d}\alpha(X)_{i_0}}{\mathrm{d}x_{i_1,j_1}}}_{\text{scalar}} = \underbrace{\frac{\mathrm{d}\langle u(X)_{i_0}, \mathbf{1}_n \rangle}{\mathrm{d}x_{i_1,j_1}}}_{\text{scalar}}$$

$$= \langle \underbrace{\frac{\mathrm{d}u(X)_{i_0}}{\mathrm{d}x_{i_1,j_1}}}_{n \times 1}, \underbrace{\mathbf{1}_n}_{n \times 1} \rangle$$

$$= \langle \underbrace{u(X)_{i_0}}_{n \times 1} \circ (e_{i_1} \cdot \langle W_{j_1,*}, X_{*,i_0} \rangle), \underbrace{\mathbf{1}_n}_{n \times 1} \rangle$$

$$= \langle \underbrace{u(X)_{i_0}}_{n \times 1} \circ e_{i_1}, \underbrace{\mathbf{1}_n}_{n \times 1} \rangle \cdot \langle W_{j_1,*}, X_{*,i_0} \rangle$$

$$= \underbrace{u(X)_{i_0,i_1}}_{\text{scalar}} \cdot \langle W_{j_1,*}, X_{*,i_0} \rangle$$

where the 1st step follows from the definition of $\alpha(X)_{i_0}$ (see Definition B.5), the 2nd step follows from Fact B.2, the 3rd step follows from Lemma B.11, the 4th step is rearrangement, the 5th step is derived by Fact B.1.

$\square$

## B.6 GRADIENT COMPUTATION FOR $\alpha(X)_{i_0}^{-1}$

**Lemma B.13** (A generalization of Lemma 5.6 in Deng et al. (2023b)). *If the following conditions hold*

- *Let $\alpha(X)_{i_0}$ be defined as Definition B.5*

*we have*

- **Part 1.** *For $i_0 = i_1 \in [n]$, $j_1 \in [d]$*

$$\underbrace{\frac{\mathrm{d}\alpha(X)_{i_0}^{-1}}{\mathrm{d}x_{i_1,j_1}}}_{\text{scalar}} = -\alpha(X)_{i_0}^{-1} \cdot (f(X)_{i_0,i_0} \cdot \langle W_{j_1,*}, X_{*,i_0} \rangle + \langle f(X)_{i_0}, X^\top W_{*,j_1} \rangle))$$

- **Part 2.** *For $i_0 \neq i_1 \in [n]$, $j_1 \in [d]$*

$$\underbrace{\frac{\mathrm{d}\alpha(X)_{i_0}^{-1}}{\mathrm{d}x_{i_1,j_1}}}_{\text{scalar}} = -\alpha(X)_{i_0}^{-1} \cdot f(X)_{i_0,i_1} \cdot \langle W_{j_1,*}, X_{*,i_0} \rangle$$

*Proof.* **Proof of Part 1.**

$$\begin{aligned}
\underbrace{\frac{\mathrm{d}\alpha(X)_{i_0}^{-1}}{\mathrm{d}x_{i_1,j_1}}}_{\text{scalar}} &= \underbrace{-1}_{\text{scalar}} \cdot \underbrace{(\alpha(X)_{i_0})^{-2}}_{\text{scalar}} \cdot \underbrace{\frac{\mathrm{d}(\alpha(X)_{i_0})}{\mathrm{d}x_{i_1,j_1}}}_{\text{scalar}} \\
&= -\underbrace{(\alpha(X)_{i_0})^{-2}}_{\text{scalar}} \cdot (u(X)_{i_0,i_0} \cdot \langle W_{j_1,*}, X_{*,i_0} \rangle + \langle u(X)_{i_0}, X^\top W_{*,j_1} \rangle) \\
&= -\alpha(X)_{i_0}^{-1} \cdot (f(X)_{i_0,i_0} \cdot \langle W_{j_1,*}, X_{*,i_0} \rangle + \langle f(X)_{i_0}, X^\top W_{*,j_1} \rangle)
\end{aligned}$$

where the 1st step follows from Fact B.2, the 2nd step follows by Lemma B.12.

**Proof of Part 2.**

$$\begin{aligned}
\underbrace{\frac{\mathrm{d}\alpha(X)_{i_0}^{-1}}{\mathrm{d}x_{i_1,j_1}}}_{\text{scalar}} &= \underbrace{-1}_{\text{scalar}} \cdot \underbrace{(\alpha(X)_{i_0})^{-2}}_{\text{scalar}} \cdot \underbrace{\frac{\mathrm{d}(\alpha(X)_{i_0})}{\mathrm{d}x_{i_1,j_1}}}_{\text{scalar}} \\
&= -\underbrace{(\alpha(X)_{i_0})^{-2}}_{\text{scalar}} \cdot u(X)_{i_0,i_1} \cdot \langle W_{j_1,*}, X_{*,i_0} \rangle \\
&= -\alpha(X)_{i_0}^{-1} \cdot f(X)_{i_0,i_1} \cdot \langle W_{j_1,*}, X_{*,i_0} \rangle
\end{aligned}$$

where the 1st step follows from Fact B.2, the 2nd step follows from result from Lemma B.12.

$\square$

### B.7 GRADIENT FOR $f(X)_{i_0}$

**Lemma B.14.** *If the following conditions hold*

- *Let $f(X)_{i_0}$ be defined as Definition B.6*

*Then, we have*

- **Part 1.** *For all $i_0 = i_1 \in [n]$, $j_1 \in [d]$*

$$\underbrace{\frac{\mathrm{d}f(X)_{i_0}}{\mathrm{d}x_{i_1,j_1}}}_{n \times 1} = -\underbrace{f(X)_{i_0}}_{n \times 1} \cdot \underbrace{(f(X)_{i_0,i_0} \cdot \langle W_{j_1,*}, X_{*,i_0} \rangle + \langle f(X)_{i_0}, X^\top W_{*,j_1} \rangle)}_{\text{scalar}}$$
$$+ \underbrace{f(X)_{i_0} \circ (e_{i_0} \cdot \langle W_{j_1,*}, X_{*,i_0} \rangle + X^\top W_{*,j_1})}_{n \times 1}$$

- **Part 2.** *For all $i_0 \neq i_1 \in [n]$, $j_1 \in [d]$*

$$\underbrace{\frac{\mathrm{d}f(X)_{i_0}}{\mathrm{d}x_{i_1,j_1}}}_{n \times 1} = -\underbrace{f(X)_{i_0}}_{n \times 1} \cdot \underbrace{f(X)_{i_0,i_1} \cdot \langle W_{j_1,*}, X_{*,i_0} \rangle}_{\text{scalar}}$$
$$+ \underbrace{f(X)_{i_0} \circ (e_{i_1} \cdot \langle W_{j_1,*}, X_{*,i_0} \rangle)}_{n \times 1}$$

*Proof.* **Proof of Part 1.**

$$\underbrace{\frac{\mathrm{d}f(X)_{i_0}}{\mathrm{d}x_{i_1,j_1}}}_{n \times 1} = \underbrace{\frac{\mathrm{d}\alpha(X)_{i_0}^{-1}u(X)_{i_0}}{\mathrm{d}x_{i_1,j_1}}}_{n \times 1}$$

$$= \underbrace{u(X)_{i_0}}_{n \times 1} \cdot \underbrace{\frac{\mathrm{d}}{\mathrm{d}x_{i_1,j_1}}\alpha(X)_{i_0}^{-1}}_{\text{scalar}} + \underbrace{\alpha(X)_{i_0}^{-1}}_{\text{scalar}} \cdot \underbrace{\frac{\mathrm{d}}{\mathrm{d}x_{i_1,j_1}}u(X)_{i_0}}_{n \times 1}$$

$$= -\underbrace{u(X)_{i_0}}_{n \times 1} \cdot \underbrace{(\alpha(X)_{i_0})^{-1} \cdot (f(X)_{i_0,i_0} \cdot \langle W_{j_1,*}, X_{*,i_0} \rangle + \langle f(X)_{i_0}, X^\top W_{*,j_1} \rangle)}_{\text{scalar}}$$
$$+ \underbrace{\alpha(X)_{i_0}^{-1}}_{\text{scalar}} \cdot \underbrace{\frac{\mathrm{d}}{\mathrm{d}x_{i_1,j_1}}u(X)_{i_0}}_{n \times 1}$$

$$= -\underbrace{u(X)_{i_0}}_{n \times 1} \cdot \underbrace{(\alpha(X)_{i_0})^{-1} \cdot (f(X)_{i_0,i_0} \cdot \langle W_{j_1,*}, X_{*,i_0} \rangle + \langle f(X)_{i_0}, X^\top W_{*,j_1} \rangle)}_{\text{scalar}}$$
$$+ \underbrace{\alpha(X)_{i_0}^{-1}}_{\text{scalar}} \cdot \underbrace{(u(X)_{i_0} \circ (e_{i_0} \cdot \langle W_{j_1,*}, X_{*,i_0} \rangle + X^\top W_{*,j_1}))}_{n \times 1}$$

$$= -\underbrace{f(X)_{i_0}}_{n \times 1} \cdot \underbrace{(f(X)_{i_0,i_0} \cdot \langle W_{j_1,*}, X_{*,i_0} \rangle + \langle f(X)_{i_0}, X^\top W_{*,j_1} \rangle)}_{\text{scalar}}$$
$$+ \underbrace{f(X)_{i_0} \circ (e_{i_0} \cdot \langle W_{j_1,*}, X_{*,i_0} \rangle + X^\top W_{*,j_1})}_{n \times 1}$$

where the 1st step follows from the definition of $f(X)_{i_0}$ (see Definition B.6), the 2nd step follows from Fact B.2, the 3rd step follows from Lemma B.13, the 4th step follows from result from Lemma B.11, the 5th step from the definition of $f(X)_{i_0}$ (see Definition B.6).

**Proof of Part 2.**

$$\underbrace{\frac{\mathrm{d}f(X)_{i_0}}{\mathrm{d}x_{i_1,j_1}}}_{n\times 1} = \underbrace{\frac{\mathrm{d}\alpha(X)_{i_0}^{-1}u(X)_{i_0}}{\mathrm{d}x_{i_1,j_1}}}_{n\times 1}$$

$$= \underbrace{u(X)_{i_0}}_{n\times 1} \cdot \underbrace{\frac{\mathrm{d}}{\mathrm{d}x_{i_1,j_1}}\alpha(X)_{i_0}^{-1}}_{\text{scalar}} + \underbrace{\alpha(X)_{i_0}^{-1}}_{\text{scalar}} \cdot \underbrace{\frac{\mathrm{d}}{\mathrm{d}x_{i_1,j_1}}u(X)_{i_0}}_{n\times 1}$$

$$= -\underbrace{u(X)_{i_0}}_{n\times 1} \cdot \underbrace{(\alpha(X)_{i_0})^{-2}\cdot u(X)_{i_0,i_1}\cdot \langle W_{j_1,*}, X_{*,i_0}\rangle}_{\text{scalar}}$$

$$+ \underbrace{\alpha(X)_{i_0}^{-1}}_{\text{scalar}} \cdot \underbrace{\frac{\mathrm{d}}{\mathrm{d}x_{i_1,j_1}}u(X)_{i_0}}_{n\times 1}$$

$$= -\underbrace{u(X)_{i_0}}_{n\times 1} \cdot \underbrace{(\alpha(X)_{i_0})^{-2}\cdot u(X)_{i_0,i_1}\cdot \langle W_{j_1,*}, X_{*,i_0}\rangle}_{\text{scalar}}$$

$$+ \underbrace{\alpha(X)_{i_0}^{-1}}_{\text{scalar}} \cdot \underbrace{(u(X)_{i_0}\circ (e_{i_1}\cdot \langle W_{j_1,*}, X_{*,i_0}\rangle))}_{n\times 1}$$

$$= -\underbrace{f(X)_{i_0}}_{n\times 1} \cdot \underbrace{f(X)_{i_0,i_1}\cdot \langle W_{j_1,*}, X_{*,i_0}\rangle}_{\text{scalar}}$$

$$+ e_{i_1}\cdot \underbrace{f(X)_{i_0,i_1}\cdot \langle W_{j_1,*}, X_{*,i_0}\rangle)}_{\text{scalar}}$$

where the 1st step follows from the definition of $f(X)_{i_0}$ (see Definition B.6), the 2nd step follows from Fact B.2, the 3rd step follows from Lemma B.13, the 4th step follows from result from Lemma B.11, the 5th step from the definition of $f(X)_{i_0}$ (see Definition B.6). $\qquad\square$

### B.8 GRADIENT FOR $h(X)_{j_0}$

**Lemma B.15.** *If the following conditions hold*

- *Let $h(X)_{j_0}$ be defined as Definition B.7*

*Then, for all $i_1 \in [n]$, $j_0, j_1 \in [d]$, we have*

$$\underbrace{\frac{\mathrm{d}h(X)_{j_0}}{\mathrm{d}x_{i_1,j_1}}}_{n\times 1} = e_{i_1}\cdot v_{j_1,j_0}$$

*Proof.*

$$\underbrace{\frac{\mathrm{d}h(X)_{j_0}}{\mathrm{d}x_{i_1,j_1}}}_{n\times 1} = \underbrace{\frac{\mathrm{d}X^\top V_{*,j_0}}{\mathrm{d}x_{i_1,j_1}}}_{n\times 1}$$

$$= \underbrace{\frac{\mathrm{d}X^\top}{\mathrm{d}x_{i_1,j_1}}}_{n\times d}\cdot \underbrace{V_{*,j_0}}_{d\times 1}$$

$$= \underbrace{e_{i_1}}_{n\times 1}\cdot \underbrace{e_{j_1}^\top}_{1\times d}\cdot \underbrace{V_{*,j_0}}_{d\times 1}$$

$$= \underbrace{e_{i_1}}_{n\times 1}\cdot \underbrace{v_{j_1,j_0}}_{\text{scalar}}$$

where the first step is by definition of $h(X)_{j_0}$ (see Definition B.7), the 2nd and the 3rd step are by differentiation rules, the 4th step is by simple algebra. $\qquad\square$

## B.9 Gradient for $c(X)_{i_0,j_0}$

**Lemma B.16.** *If the following conditions hold*

- *Let $c(X)_{i_0}$ be defined as Definition B.8*
- *Let $s(X)_{i_0,j_0} := \langle f(X)_{i_0}, h(X)_{j_0} \rangle$*

*Then, we have*

- **Part 1.** *For all $i_0 = i_1 \in [n]$, $j_0, j_1 \in [d]$*

$$\frac{\mathrm{d}c(X)_{i_0,j_0}}{\mathrm{d}x_{i_1,j_1}} = C_1(X) + C_2(X) + C_3(X) + C_4(X) + C_5(X)$$

  *where we have definitions:*

  - $C_1(X) := -s(X)_{i_0,j_0} \cdot f(X)_{i_0,i_0} \cdot \langle W_{j_1,*}, X_{*,i_0} \rangle$
  - $C_2(X) := -s(X)_{i_0,j_0} \cdot \langle f(X)_{i_0}, X^\top W_{*,j_1} \rangle$
  - $C_3(X) := f(X)_{i_0,i_0} \cdot h(X)_{j_0,i_0} \cdot \langle W_{j_1,*}, X_{*,i_0} \rangle$
  - $C_4(X) := \langle f(X)_{i_0} \circ (X^\top W_{*,j_1}), h(X)_{j_0} \rangle$
  - $C_5(X) := f(X)_{i_0,i_0} \cdot v_{j_1,j_0}$

- **Part 2.** *For all $i_0 \neq i_1 \in [n]$, $j_0, j_1 \in [d]$*

$$\frac{\mathrm{d}c(X)_{i_0,j_0}}{\mathrm{d}x_{i_1,j_1}} = C_6(X) + C_7(X) + C_8(X)$$

  *where we have definitions:*

  - $C_6(X) := -s(X)_{i_0,j_0} \cdot f(X)_{i_0,i_1} \cdot \langle W_{j_1,*}, X_{*,i_0} \rangle$
    * *This is corresponding to $C_1(X)$*
  - $C_7(X) := f(X)_{i_0,i_1} \cdot h(X)_{j_0,i_1} \cdot \langle W_{j_1,*}, X_{*,i_0} \rangle$
    * *This is corresponding to $C_3(X)$*
  - $C_8(X) := f(X)_{i_0,i_1} \cdot v_{j_1,j_0}$
    * *This is corresponding to $C_5(X)$*

*Proof.* **Proof of Part 1**

$$\underbrace{\frac{\mathrm{d}c(X)_{i_0,j_1}}{\mathrm{d}x_{i_1,j_1}}}_{\text{scalar}} = \underbrace{\frac{\mathrm{d}(\langle f(X)_{i_0}, h(X)_{j_0} \rangle - b_{i_0,j_0})}{\mathrm{d}x_{i_1,j_1}}}_{\text{scalar}}$$

$$= \underbrace{\frac{\mathrm{d}\langle f(X)_{i_0}, h(X)_{j_0} \rangle}{\mathrm{d}x_{i_1,j_1}}}_{\text{scalar}}$$

$$= \langle \underbrace{\frac{\mathrm{d}f(X)_{i_0}}{\mathrm{d}x_{i_1,j_1}}}_{n \times 1}, \underbrace{h(X)_{j_0}}_{n \times 1} \rangle + \langle \underbrace{f(X)_{i_0}}_{n \times 1}, \underbrace{\frac{\mathrm{d}h(X)_{j_0}}{\mathrm{d}x_{i_1,j_1}}}_{n \times 1} \rangle$$

$$= \langle \underbrace{\frac{\mathrm{d}f(X)_{i_0}}{\mathrm{d}x_{i_1,j_1}}}_{n \times 1}, \underbrace{h(X)_{j_0}}_{n \times 1} \rangle + \langle \underbrace{f(X)_{i_0}}_{n \times 1}, \underbrace{e_{i_1}}_{n \times 1} \cdot \underbrace{v_{j_1,j_0}}_{\text{scalar}} \rangle$$

$$= \langle - \underbrace{f(X)_{i_0}}_{n \times 1} \cdot \underbrace{(f(X)_{i_0,i_0} \cdot \langle W_{j_1,*}, X_{*,i_0} \rangle + \langle f(X)_{i_0}, X^\top W_{*,j_1} \rangle)}_{\text{scalar}}$$

$$+ \underbrace{f(X)_{i_0} \circ (e_{i_0} \cdot \langle W_{j_1,*}, X_{*,i_0} \rangle + X^\top W_{*,j_1})}_{n \times 1}, \underbrace{h(X)_{j_0}}_{n \times 1} \rangle + \langle \underbrace{f(X)_{i_0}}_{n \times 1}, \underbrace{e_{i_1}}_{n \times 1} \cdot \underbrace{v_{j_1,j_0}}_{\text{scalar}} \rangle$$

$$
\begin{aligned}
= & - s(X)_{i_0,j_0} \cdot f(X)_{i_0,i_0} \cdot \langle W_{j_1,*}, X_{*,i_0} \rangle \\
& - s(X)_{i_0,j_0} \cdot \langle f(X)_{i_0}, X^\top W_{*,j_1} \rangle \\
& + f(X)_{i_0,i_0} h(X)_{j_0,i_0} \langle W_{j_1,*}, X_{*,i_0} \rangle \\
& + \langle f(X)_{i_0} \circ (X^\top W_{*,j_1}), h(X)_{j_0} \rangle \\
& + f(X)_{i_0,i_1} v_{j_1,j_0} \\
:= & \ C_1(X) + C_2(X) + C_3(X) + C_4(X) + C_5(X)
\end{aligned}
$$

where the first step is by definition of $c(X)_{i_0,j_0}$ (see Definition B.8), the 2nd step is because $b_{i_0,j_0}$ is independent of $X$, the 3rd step is by Fact B.2, the 4th step uses Lemma B.15, the 5th step uses Lemma B.14, the 6th and 8th step are rearrangement of terms, the 7th step holds by the definition of $f(X)_{i_0}$ (see Definition B.6).

**Proof of Part 2**

$$
\underbrace{\frac{\mathrm{d}c(X)_{i_0,j_1}}{\mathrm{d}x_{i_1,j_1}}}_{\text{scalar}} = \underbrace{\frac{\mathrm{d}(\langle f(X)_{i_0}, h(X)_{j_0} \rangle - b_{i_0,j_0})}{\mathrm{d}x_{i_1,j_1}}}_{\text{scalar}}
$$

$$
= \underbrace{\frac{\mathrm{d}\langle f(X)_{i_0}, h(X)_{j_0} \rangle}{\mathrm{d}x_{i_1,j_1}}}_{\text{scalar}}
$$

$$
= \langle \underbrace{\frac{\mathrm{d}f(X)_{i_0}}{\mathrm{d}x_{i_1,j_1}}}_{n \times 1}, \underbrace{h(X)_{j_0}}_{n \times 1} \rangle + \langle \underbrace{f(X)_{i_0}}_{n \times 1}, \underbrace{\frac{\mathrm{d}h(X)_{j_0}}{\mathrm{d}x_{i_1,j_1}}}_{n \times 1} \rangle
$$

$$
= \langle \underbrace{\frac{\mathrm{d}f(X)_{i_0}}{\mathrm{d}x_{i_1,j_1}}}_{n \times 1}, \underbrace{h(X)_{j_0}}_{n \times 1} \rangle + \langle \underbrace{f(X)_{i_0}}_{n \times 1}, \underbrace{e_{i_1}}_{n \times 1} \cdot \underbrace{v_{j_1,j_0}}_{\text{scalar}} \rangle
$$

$$
= \langle - \underbrace{(\alpha(X)_{i_0})^{-1}}_{\text{scalar}} \cdot \underbrace{f(X)_{i_0}}_{n \times 1} \cdot \underbrace{u(X)_{i_0,i_1} \cdot \langle W_{j_1,*}, X_{*,i_0} \rangle}_{\text{scalar}}
$$

$$
+ \underbrace{f(X)_{i_0} \circ (e_{i_1} \cdot \langle W_{j_1,*}, X_{*,i_0} \rangle)}_{n \times 1}, \underbrace{h(X)_{j_0}}_{n \times 1} \rangle + \langle \underbrace{f(X)_{i_0}}_{n \times 1}, \underbrace{e_{i_1}}_{n \times 1} \cdot \underbrace{v_{j_1,j_0}}_{\text{scalar}} \rangle
$$

$$
= - \underbrace{(\alpha(X)_{i_0})^{-1} \cdot \langle f(X)_{i_0}, h(X)_{j_0} \rangle \cdot u(X)_{i_0,i_1} \cdot \langle W_{j_1,*}, X_{*,i_0} \rangle}_{\text{scalar}}
$$

$$
+ \underbrace{\langle f(X)_{i_0} \circ e_{i_1}, h(X)_{j_0} \rangle \cdot \langle W_{j_1,*}, X_{*,i_0} \rangle}_{\text{scalar}}
$$

$$
+ \langle \underbrace{f(X)_{i_0}}_{n \times 1}, \underbrace{e_{i_1}}_{n \times 1} \cdot \underbrace{v_{j_1,j_0}}_{\text{scalar}} \rangle
$$

$$
= - s(X)_{i_0,j_0} \cdot f(X)_{i_0,i_1} \cdot \langle W_{j_1,*}, X_{*,i_0} \rangle
$$

$$
+ f(X)_{i_0,i_1} \cdot h(X)_{j_0,i_1} \cdot \langle W_{j_1,*}, X_{*,i_0} \rangle
$$

$$
+ f(X)_{i_0,i_1} \cdot v_{j_1,j_0}
$$

$$
:= C_6(X) + C_7(X) + C_8(X)
$$

where the first step is by definition of $c(X)_{i_0,j_0}$ (see Definition B.8), the 2nd step is because $b_{i_0,j_0}$ is independent of $X$, the 3rd step is by Fact B.2, the 4th step uses Lemma B.15, the 5th step uses Lemma B.14, the 6th and 7th step are rearrangement of terms. $\square$

## B.10 Gradient for $L(X)$

**Lemma B.17.** *If the following holds*

- *Let $L(X)$ be defined as Definition B.9*

*For $i_1 \in [n]$, $j_1 \in [d]$, we have*

$$\frac{\mathrm{d}L(X)}{\mathrm{d}x_{i_1,j_1}} = \sum_{i_0=1}^{n} \sum_{j_0=1}^{d} c(X)_{i_0,j_0} \cdot \frac{\mathrm{d}c(X)_{i_0,j_0}}{\mathrm{d}x_{i_1,j_1}}$$

*Proof.* The result directly follows by chain rule. $\square$

## C    HESSIAN CASE 1: $i_0 = i_1$

Here in this section, we provide Hessian analysis for the first case. In Sections C.1, C.2, C.3, C.4, C.5, C.6 and C.8, we calculate the derivative for several important terms. In Section C.9, C.10, C.11, C.12 and C.13 we calculate derivative for $C_1, C_2, C_3, C_4$ and $C_5$ respectively. Finally in Section C.14 we calculate derivative of $\frac{c(X)_{i_0,j_0}}{\mathrm{d}x_{i_1,j_2}}$.

### C.1    DERIVATIVE OF SCALAR FUNCTION $w(X)_{i_0,j_1}$

**Lemma C.1.** *We have*

- **Part 1** *For $i_0 = i_1 = i_2 \in [n]$, $j_1, j_2 \in [d]$*

$$\frac{\mathrm{d}w(X)_{i_0,j_1}}{\mathrm{d}x_{i_2,j_2}} = w_{j_1,j_2}$$

- **Part 2** *For $i_0 = i_1 \neq i_2 \in [n]$, $j_1, j_2 \in [d]$*

$$\frac{\mathrm{d}w(X)_{i_0,j_1}}{\mathrm{d}x_{i_2,j_2}} = 0$$

*Proof.* **Proof of Part 1**

$$\begin{aligned}
\frac{\mathrm{d}w(X)_{i_0,j_1}}{\mathrm{d}x_{i_2,j_2}} &= \langle W_{j_1,*}, \frac{\mathrm{d}X_{*,i_0}}{\mathrm{d}x_{i_2,j_2}} \rangle \\
&= \langle W_{j_1,*}, e_{j_2} \rangle \\
&= w_{j_1,j_2}
\end{aligned}$$

where the first step and the 2nd step are by Fact B.2, the 3rd step is simple algebra.

**Proof of Part 2**

$$\begin{aligned}
\frac{\mathrm{d}w(X)_{i_0,j_1}}{\mathrm{d}x_{i_2,j_2}} &= \langle W_{j_1,*}, \frac{\mathrm{d}X_{*,i_0}}{\mathrm{d}x_{i_2,j_2}} \rangle \\
&= \langle W_{j_1,*}, \mathbf{0}_d \rangle = 0
\end{aligned}$$

where the first step is by Fact B.2, the 2nd step is because $i_0 \neq i_2$. $\square$

### C.2    DERIVATIVE OF VECTOR FUNCTION $X^\top W_{*,j_1}$

**Lemma C.2.** *We have*

- **Part 1** *For $i_0 = i_1 = i_2 \in [n]$, $j_1, j_2 \in [d]$*

$$\frac{\mathrm{d}X^\top W_{*,j_1}}{\mathrm{d}x_{i_2,j_2}} = e_{i_0} \cdot w_{j_2,j_1}$$

- **Part 2** *For $i_0 = i_1 \neq i_2 \in [n]$, $j_1, j_2 \in [d]$*

$$\frac{\mathrm{d}X^\top W_{*,j_1}}{\mathrm{d}x_{i_2,j_2}} = e_{i_2} \cdot w_{j_2,j_1}$$

*Proof.* **Proof of Part 1**

$$
\frac{\mathrm{d}X^\top W_{*,j_1}}{\mathrm{d}x_{i_2,j_2}} = \frac{\mathrm{d}X^\top}{\mathrm{d}x_{i_2,j_2}} \cdot W_{*,j_1}
$$
$$
= e_{i_2} e_{j_2}^\top \cdot W_{*,j_1}
$$
$$
= e_{i_2} \cdot w_{j_2,j_1}
$$
$$
= e_{i_0} \cdot w_{j_2,j_1}
$$

where the first step and the 2nd step are by Fact B.2, the 3rd step is simple algebra, the 4th step holds since $i_0 = i_2$.

**Proof of Part 2**

$$
\frac{\mathrm{d}X^\top W_{*,j_1}}{\mathrm{d}x_{i_2,j_2}} = \frac{\mathrm{d}X^\top}{\mathrm{d}x_{i_2,j_2}} \cdot W_{*,j_1}
$$
$$
= e_{i_2} e_{j_2}^\top \cdot W_{*,j_1}
$$
$$
= e_{i_2} \cdot w_{j_2,j_1}
$$

where the first step and the 2nd step are by Fact B.2, the 3rd step is simple algebra. $\square$

## C.3 DERIVATIVE OF SCALAR FUNCTION $f(X)_{i_0,i_0}$

**Lemma C.3.** *If the following holds:*

- *Let $f(X)_{i_0}$ be defined as Definition B.6*

*We have*

- **Part 1** *For $i_0 = i_2 \in [n]$, $j_1, j_2 \in [d]$*

$$
\frac{\mathrm{d}f(X)_{i_0,i_0}}{\mathrm{d}x_{i_2,j_2}} = - f(X)_{i_0,i_0} \cdot (f(X)_{i_0,i_0} \cdot w(X)_{i_0,j_2} + \langle f(X)_{i_0}, X^\top W_{*,j_2}\rangle)
$$
$$
+ f(X)_{i_0,i_0} \cdot \langle W_{j_2,*} + W_{*,j_2}, X_{*,i_0}\rangle
$$

- **Part 2** *For $i_0 \neq i_2 \in [n]$, $j_1, j_2 \in [d]$*

$$
\frac{\mathrm{d}f(X)_{i_0,i_0}}{\mathrm{d}x_{i_2,j_2}} = -f(X)_{i_0,i_0} \cdot f(X)_{i_0,i_2} \cdot w(X)_{i_0,j_2}
$$

*Proof.* **Proof of Part 1**

$$
\frac{\mathrm{d}f(X)_{i_0,i_0}}{\mathrm{d}x_{i_2,j_2}} = (-(\alpha(X)_{i_0})^{-1} \cdot f(X)_{i_0} \cdot (u(X)_{i_0,i_0} \cdot w(X)_{i_0,j_2} + \langle u(X)_{i_0}, X^\top W_{*,j_2}\rangle)
$$
$$
+ f(X)_{i_0} \circ (e_{i_0} \cdot w(X)_{i_0,j_2} + X^\top W_{*,j_2}))_{i_0}
$$
$$
= -(\alpha(X)_{i_0})^{-1} \cdot f(X)_{i_0,i_0} \cdot (u(X)_{i_0,i_0} \cdot w(X)_{i_0,j_2} + \langle u(X)_{i_0}, X^\top W_{*,j_2}\rangle)
$$
$$
+ (f(X)_{i_0} \circ (e_{i_0} \cdot w(X)_{i_0,j_2}))_{i_0} + (f(X)_{i_0} \circ (X^\top W_{*,j_2}))_{i_0}
$$
$$
= -(\alpha(X)_{i_0})^{-1} \cdot f(X)_{i_0,i_0} \cdot (u(X)_{i_0,i_0} \cdot w(X)_{i_0,j_2} + \langle u(X)_{i_0}, X^\top W_{*,j_2}\rangle)
$$
$$
+ f(X)_{i_0,i_0} \cdot w(X)_{i_0,j_2} + f(X)_{i_0,i_0} \cdot \langle W_{*,j_2}, X_{*,i_0}\rangle
$$
$$
= - f(X)_{i_0,i_0} \cdot (f(X)_{i_0,i_0} \cdot w(X)_{i_0,j_2} + \langle f(X)_{i_0}, X^\top W_{*,j_2}\rangle)
$$
$$
+ f(X)_{i_0,i_0} \cdot w(X)_{i_0,j_2} + f(X)_{i_0,i_0} \cdot \langle W_{*,j_2}, X_{*,i_0}\rangle
$$

where the first step uses Lemma B.14 for $i_0 = i_2$, the following steps are taking the $i_0$-th entry of $f(X)_{i_0}$, the last step is by the definition of $f(X)_{i_0}$ (see Definition B.6).

**Proof of Part 2**

$$
\frac{\mathrm{d}f(X)_{i_0,i_0}}{\mathrm{d}x_{i_2,j_2}} = (-(\alpha(X)_{i_0})^{-1} \cdot f(X)_{i_0} \cdot u(X)_{i_0,i_2} \cdot w(X)_{i_0,j_2}
$$

$$+ f(X)_{i_0} \circ (e_{i_2} \cdot w(X)_{i_0,j_2}))_{i_0}$$
$$= - (\alpha(X)_{i_0})^{-1} \cdot f(X)_{i_0,i_0} \cdot u(X)_{i_0,i_2} \cdot w(X)_{i_0,j_2}$$
$$+ (f(X)_{i_0} \circ (e_{i_2} \cdot w(X)_{i_0,j_2}))_{i_0}$$
$$= - (\alpha(X)_{i_0})^{-1} \cdot f(X)_{i_0,i_0} \cdot u(X)_{i_0,i_2} \cdot w(X)_{i_0,j_2}$$
$$= - f(X)_{i_0,i_0} \cdot f(X)_{i_0,i_2} \cdot w(X)_{i_0,j_2}$$

where the first step uses Lemma B.14 for $i_0 \neq i_2$, the 2nd step is taking the $i_0$-th entry of $f(X)_{i_0}$, the 3rd step is because $i_0 \neq i_2$, the last step is by the definition of $f(X)_{i_0}$ (see Definition B.6). $\qquad\square$

## C.4 Derivative of Scalar Function $h(X)_{j_0,i_0}$

**Lemma C.4.** *If the following holds:*

- *Let $h(X)_{j_0}$ be defined as Definition B.7*

*We have*

- **Part 1** *For $i_0 = i_2 \in [n]$, $j_1, j_2 \in [d]$*

$$\frac{\mathrm{d}h(X)_{j_0,i_0}}{\mathrm{d}x_{i_2,j_2}} = v_{j_2,j_0}$$

- **Part 2** *For $i_0 \neq i_2 \in [n]$, $j_1, j_2 \in [d]$*

$$\frac{\mathrm{d}h(X)_{j_0,i_0}}{\mathrm{d}x_{i_2,j_2}} = 0$$

*Proof.* **Proof of Part 1**

$$\frac{\mathrm{d}h(X)_{j_0,i_0}}{\mathrm{d}x_{i_2,j_2}} = (e_{i_2} \cdot v_{j_2,j_0})_{i_0}$$
$$= v_{j_2,j_0}$$

where the first step is by Lemma B.15, the 2nd step is because $i_0 = i_2$.

**Proof of Part 2**

$$\frac{\mathrm{d}h(X)_{j_0,i_0}}{\mathrm{d}x_{i_2,j_2}} = (e_{i_2} \cdot v_{j_2,j_0})_{i_0}$$
$$= 0$$

where the first step is by Lemma B.15, the 2nd step is because $i_0 \neq i_2$. $\qquad\square$

## C.5 Derivative of Scalar Function $z(X)_{i_0,j_1}$

**Lemma C.5.** *If the following holds:*

- *Let $f(X)_{i_0}$ be defined as Definition B.6*
- *Let $z(X)_{i_0,j_1} := \langle f(X)_{i_0}, X^\top W_{*,j_1} \rangle$*
- *Let $w(X)_{i_0,j_1} = \langle W_{j_1,*}, X_{*,i_0} \rangle$*

*We have*

- **Part 1** *For $i_0 = i_1 = i_2 \in [n]$, $j_1, j_2 \in [d]$*

$$\frac{\mathrm{d}z(X)_{i_0,j_1}}{\mathrm{d}x_{i_2,j_2}}$$
$$= - z(X)_{i_0,j_1} \cdot f(X)_{i_0,i_0} \cdot w(X)_{i_0,j_2}$$

$$- z(X)_{i_0,j_1} \cdot z(X)_{i_0,j_2}$$
$$+ f(X)_{i_0,i_0} \cdot \langle W_{*,j_1}, X_{*,i_0} \rangle \cdot w(X)_{i_0,j_2}$$
$$+ \langle f(X)_{i_0} \circ X^\top W_{*,j_2}, X^\top W_{*,j_1} \rangle$$
$$+ f(X)_{i_0,i_0} \cdot w_{j_2,j_1}$$

- **Part 2** *For $i_0 = i_1 \neq i_2 \in [n]$, $j_1, j_2 \in [d]$*

$$\frac{\mathrm{d}\langle f(X)_{i_0}, X^\top W_{*,j_1} \rangle}{\mathrm{d}x_{i_2,j_2}}$$
$$= - z(X)_{i_0,j_1} \cdot f(X)_{i_0,i_0} \cdot w(X)_{i_0,j_2}$$
$$+ f(X)_{i_0,i_0} \cdot w(X)_{i_0,j_2} \cdot \langle W_{*,j_1}, X_{*,i_0} \rangle$$
$$+ f(X)_{i_0,i_0} \cdot w_{j_2,j_1}$$

*Proof.* **Proof of Part 1**

$$\frac{\mathrm{d}\langle f(X)_{i_0}, X^\top W_{*,j_1} \rangle}{\mathrm{d}x_{i_2,j_2}}$$
$$= \langle \frac{\mathrm{d}f(X)_{i_0}}{\mathrm{d}x_{i_2,j_2}}, X^\top W_{*,j_1} \rangle + \langle f(X)_{i_0}, \frac{\mathrm{d}X^\top W_{*,j_1}}{\mathrm{d}x_{i_2,j_2}} \rangle$$
$$= \langle \frac{\mathrm{d}f(X)_{i_0}}{\mathrm{d}x_{i_2,j_2}}, X^\top W_{*,j_1} \rangle + \langle f(X)_{i_0}, e_{i_0} \cdot w_{j_2,j_1} \rangle$$
$$= \langle \frac{\mathrm{d}f(X)_{i_0}}{\mathrm{d}x_{i_2,j_2}}, X^\top W_{*,j_1} \rangle + f(X)_{i_0,i_0} \cdot w_{j_2,j_1}$$
$$= \langle -(\alpha(X)_{i_0})^{-1} \cdot f(X)_{i_0} \cdot (u(X)_{i_0,i_0} \cdot w(X)_{i_0,j_2} + \langle u(X)_{i_0}, X^\top W_{*,j_2} \rangle)$$
$$+ f(X)_{i_0} \circ (e_{i_0} \cdot w(X)_{i_0,j_2} + X^\top W_{*,j_2}), X^\top W_{*,j_1} \rangle + f(X)_{i_0,i_0} \cdot w_{j_2,j_1}$$
$$= \langle -f(X)_{i_0} \cdot (f(X)_{i_0,i_0} \cdot w(X)_{i_0,j_2} + \langle f(X)_{i_0}, X^\top W_{*,j_2} \rangle)$$
$$+ f(X)_{i_0} \circ (e_{i_0} \cdot w(X)_{i_0,j_2} + X^\top W_{*,j_2}), X^\top W_{*,j_1} \rangle + f(X)_{i_0,i_0} \cdot w_{j_2,j_1}$$
$$= - z(X)_{i_0,j_1} \cdot f(X)_{i_0,i_0} \cdot w(X)_{i_0,j_2}$$
$$- z(X)_{i_0,j_1} \cdot z(X)_{i_0,j_2}$$
$$+ f(X)_{i_0,i_0} \cdot \langle W_{*,j_1}, X_{*,i_0} \rangle \cdot w(X)_{i_0,j_2}$$
$$+ \langle f(X)_{i_0} \circ X^\top W_{*,j_2}, X^\top W_{*,j_1} \rangle$$
$$+ f(X)_{i_0,i_0} \cdot w_{j_2,j_1}$$

where the 1st step is by Fact B.2, the 2nd step uses Lemma C.2, the 3rd step is taking the $i_0$-th entry of $f(X)_{i_0}$, the 4th step uses Lemma B.14, the 5th step is by the definition of $f(X)_{i_0}$ (see Definition B.6).

**Proof of Part 2**

$$\frac{\mathrm{d}\langle f(X)_{i_0}, X^\top W_{*,j_1} \rangle}{\mathrm{d}x_{i_2,j_2}}$$
$$= \langle \frac{\mathrm{d}f(X)_{i_0}}{\mathrm{d}x_{i_2,j_2}}, X^\top W_{*,j_1} \rangle + \langle f(X)_{i_0}, \frac{\mathrm{d}X^\top W_{*,j_1}}{\mathrm{d}x_{i_2,j_2}} \rangle$$
$$= \langle \frac{\mathrm{d}f(X)_{i_0}}{\mathrm{d}x_{i_2,j_2}}, X^\top W_{*,j_1} \rangle + \langle f(X)_{i_0}, e_{i_2} \cdot w_{j_2,j_1} \rangle$$
$$= \langle \frac{\mathrm{d}f(X)_{i_0}}{\mathrm{d}x_{i_2,j_2}}, X^\top W_{*,j_1} \rangle + f(X)_{i_0,i_2} \cdot w_{j_2,j_1}$$
$$= \langle -(\alpha(X)_{i_0})^{-1} \cdot f(X)_{i_0} \cdot u(X)_{i_0,i_0} \cdot w(X)_{i_0,j_2}$$
$$+ f(X)_{i_0} \circ (e_{i_0} \cdot w(X)_{i_0,j_2}), X^\top W_{*,j_1} \rangle + f(X)_{i_0,i_0} \cdot w_{j_2,j_1}$$
$$= \langle -f(X)_{i_0} \cdot f(X)_{i_0,i_0} \cdot w(X)_{i_0,j_2}$$

$$+ f(X)_{i_0} \circ (e_{i_0} \cdot w(X)_{i_0,j_2}), X^\top W_{*,j_1}\rangle + f(X)_{i_0,i_0} \cdot w_{j_2,j_1}$$
$$= - z(X)_{i_0,j_1} \cdot f(X)_{i_0,i_0} \cdot w(X)_{i_0,j_2}$$
$$+ f(X)_{i_0,i_0} \cdot w(X)_{i_0,j_2} \cdot \langle W_{*,j_1}, X_{*,i_0}\rangle$$
$$+ f(X)_{i_0,i_0} \cdot w_{j_2,j_1}$$

where the 1st step is by Fact B.2, the 2nd step uses Lemma C.2, the 3rd step is taking the $i_0$-th entry of $f(X)_{i_0}$, the 4th step uses Lemma B.14, the last step is by the definition of $f(X)_{i_0}$ (see Definition B.6). $\square$

## C.6 DERIVATIVE OF SCALAR FUNCTION $f(X)_{i_0,i_0} \cdot h(X)_{j_0,i_0}$

**Lemma C.6.** *If the following holds:*

- *Let $f(X)_{i_0}$ be defined as Definition B.6*
- *Let $h(X)_{j_0}$ be defined as Definition B.7*

*We have*

- **Part 1** *For $i_0 = i_1 = i_2 \in [n]$, $j_1, j_2 \in [d]$*

$$\frac{\mathrm{d}f(X)_{i_0,i_0} \cdot h(X)_{j_0,i_0}}{\mathrm{d}x_{i_2,j_2}}$$
$$= (-f(X)_{i_0,i_0} \cdot (f(X)_{i_0,i_0} \cdot w(X)_{i_0,j_2} + \langle f(X)_{i_0}, X^\top W_{*,j_2}\rangle)$$
$$+ f(X)_{i_0,i_0} \cdot \langle W_{j_2,*} + W_{*,j_2}, X_{*,i_0}\rangle) \cdot h(X)_{j_0,i_0} + f(X)_{i_0,i_0} \cdot v_{j_2,j_0}$$

- **Part 2** *For $i_0 = i_1 \neq i_2 \in [n]$, $j_1, j_2 \in [d]$*

$$\frac{\mathrm{d}f(X)_{i_0,i_0} \cdot h(X)_{j_0,i_0}}{\mathrm{d}x_{i_2,j_2}} = -f(X)_{i_0,i_0} \cdot f(X)_{i_0,i_2} \cdot w(X)_{i_0,j_2} \cdot h(X)_{j_0,i_0}$$

*Proof.* **Proof of Part 1**

$$\frac{\mathrm{d}f(X)_{i_0,i_0} \cdot h(X)_{j_0,i_0}}{\mathrm{d}x_{i_2,j_2}}$$
$$= \frac{\mathrm{d}f(X)_{i_0,i_0}}{\mathrm{d}x_{i_2,j_2}} \cdot h(X)_{j_0,i_0} + f(X)_{i_0,i_0} \cdot \frac{\mathrm{d}h(X)_{j_0,i_0}}{\mathrm{d}x_{i_2,j_2}}$$
$$= \frac{\mathrm{d}f(X)_{i_0,i_0}}{\mathrm{d}x_{i_2,j_2}} \cdot h(X)_{j_0,i_0} + f(X)_{i_0,i_0} \cdot v_{j_2,j_0}$$
$$= (-(\alpha(X)_{i_0})^{-1} \cdot f(X)_{i_0,i_0} \cdot (u(X)_{i_0,i_0} \cdot w(X)_{i_0,j_2} + \langle u(X)_{i_0}, X^\top W_{*,j_2}\rangle)$$
$$+ f(X)_{i_0,i_0} \cdot \langle W_{j_2,*} + W_{*,j_2}, X_{*,i_0}\rangle) \cdot h(X)_{j_0,i_0} + f(X)_{i_0,i_0} \cdot v_{j_2,j_0}$$
$$= (-f(X)_{i_0,i_0} \cdot (f(X)_{i_0,i_0} \cdot w(X)_{i_0,j_2} + \langle f(X)_{i_0}, X^\top W_{*,j_2}\rangle)$$
$$+ f(X)_{i_0,i_0} \cdot \langle W_{j_2,*} + W_{*,j_2}, X_{*,i_0}\rangle) \cdot h(X)_{j_0,i_0} + f(X)_{i_0,i_0} \cdot v_{j_2,j_0}$$

where the fist step is by Fact B.2, the 2nd step calls Lemma C.4, the 3rd step uses Lemma C.3, the last step is by the definition of $f(X)_{i_0}$ (see Definition B.6).

**Proof of Part 2**

$$\frac{\mathrm{d}f(X)_{i_0,i_0} \cdot h(X)_{j_0,i_0}}{\mathrm{d}x_{i_2,j_2}}$$
$$= \frac{\mathrm{d}f(X)_{i_0,i_0}}{\mathrm{d}x_{i_2,j_2}} \cdot h(X)_{j_0,i_0} + f(X)_{i_0,i_0} \cdot \frac{\mathrm{d}h(X)_{j_0,i_0}}{\mathrm{d}x_{i_2,j_2}}$$
$$= -(\alpha(X)_{i_0})^{-1} \cdot f(X)_{i_0,i_0} \cdot u(X)_{i_0,i_2} \cdot w(X)_{i_0,j_2} \cdot h(X)_{j_0,i_0}$$
$$= -f(X)_{i_0,i_0} \cdot f(X)_{i_0,i_2} \cdot w(X)_{i_0,j_2} \cdot h(X)_{j_0,i_0}$$

where the fist step is by Fact B.2, the 2nd step calls Lemma C.4, the 3rd step uses Lemma C.3, the last step is by the definition of $f(X)_{i_0}$ (see Definition B.6). $\square$

## C.7 DERIVATIVE OF SCALAR FUNCTION $f(X)_{i_0,i_0} \cdot w(X)_{i_0,j_1}$

**Lemma C.7.** *If the following holds:*

- *Let $f(X)_{i_0}$ be defined as Definition B.6*

*We have*

- **Part 1** *For $i_0 = i_1 = i_2 \in [n]$, $j_1, j_2 \in [d]$*

$$\frac{\mathrm{d}f(X)_{i_0,i_0} \cdot w(X)_{i_0,j_1}}{\mathrm{d}x_{i_2,j_2}}$$
$$= (f(X)_{i_0,i_0} \cdot (f(X)_{i_0,i_0} \cdot w(X)_{i_0,j_2} + \langle f(X)_{i_0}, X^\top W_{*,j_2} \rangle)$$
$$+ f(X)_{i_0,i_0} \cdot \langle W_{j_2,*} + W_{*,j_2}, X_{*,i_0} \rangle) \cdot w(X)_{i_0,j_1} + f(X)_{i_0,i_0} \cdot w_{j_1,j_2}$$

- **Part 2** *For $i_0 = i_1 \neq i_2 \in [n]$, $j_1, j_2 \in [d]$*

$$\frac{\mathrm{d}f(X)_{i_0,i_0} \cdot w(X)_{i_0,j_1}}{\mathrm{d}x_{i_2,j_2}} = -f(X)_{i_0,i_0} \cdot f(X)_{i_0,i_2} \cdot w(X)_{i_0,j_2} \cdot w(X)_{i_0,j_1}$$

*Proof.* **Proof of Part 1**

$$\frac{\mathrm{d}f(X)_{i_0,i_0} \cdot w(X)_{i_0,j_1}}{\mathrm{d}x_{i_2,j_2}}$$
$$= \frac{\mathrm{d}f(X)_{i_0,i_0}}{\mathrm{d}x_{i_2,j_2}} \cdot w(X)_{i_0,j_1} + f(X)_{i_0,i_0} \cdot \frac{\mathrm{d}w(X)_{i_0,j_1}}{\mathrm{d}x_{i_2,j_2}}$$
$$= \frac{\mathrm{d}f(X)_{i_0,i_0}}{\mathrm{d}x_{i_2,j_2}} \cdot w(X)_{i_0,j_1} + f(X)_{i_0,i_0} \cdot w_{j_1,j_2}$$
$$= (-(\alpha(X)_{i_0})^{-1} \cdot f(X)_{i_0,i_0} \cdot (u(X)_{i_0,i_0} \cdot w(X)_{i_0,j_2} + \langle u(X)_{i_0}, X^\top W_{*,j_2} \rangle)$$
$$+ f(X)_{i_0,i_0} \cdot \langle W_{j_2,*} + W_{*,j_2}, X_{*,i_0} \rangle) \cdot w(X)_{i_0,j_1} + f(X)_{i_0,i_0} \cdot w_{j_1,j_2}$$
$$= (-f(X)_{i_0,i_0} \cdot (f(X)_{i_0,i_0} \cdot w(X)_{i_0,j_2} + \langle f(X)_{i_0}, X^\top W_{*,j_2} \rangle)$$
$$+ f(X)_{i_0,i_0} \cdot \langle W_{j_2,*} + W_{*,j_2}, X_{*,i_0} \rangle) \cdot w(X)_{i_0,j_1} + f(X)_{i_0,i_0} \cdot w_{j_1,j_2}$$

where step 1 is by Fact B.2, the 2nd step calls Lemma C.1, the 3rd step uses Lemma C.3, the last step is by the definition of $f(X)_{i_0}$ (see Definition B.6).

**Proof of Part 2**

$$\frac{\mathrm{d}f(X)_{i_0,i_0} \cdot w(X)_{i_0,j_1}}{\mathrm{d}x_{i_2,j_2}}$$
$$= \frac{\mathrm{d}f(X)_{i_0,i_0}}{\mathrm{d}x_{i_2,j_2}} \cdot w(X)_{i_0,j_1} + f(X)_{i_0,i_0} \cdot \frac{\mathrm{d}w(X)_{i_0,j_1}}{\mathrm{d}x_{i_2,j_2}}$$
$$= \frac{\mathrm{d}f(X)_{i_0,i_0}}{\mathrm{d}x_{i_2,j_2}} \cdot w(X)_{i_0,j_1}$$
$$= -(\alpha(X)_{i_0})^{-1} \cdot f(X)_{i_0,i_0} \cdot u(X)_{i_0,i_2} \cdot w(X)_{i_0,j_2} \cdot w(X)_{i_0,j_1}$$
$$= -f(X)_{i_0,i_0} \cdot f(X)_{i_0,i_2} \cdot w(X)_{i_0,j_2} \cdot w(X)_{i_0,j_1}$$

where step 1 is by Fact B.2, the 2nd step calls Lemma C.1, the 3rd step uses Lemma C.3, the last step is by the definition of $f(X)_{i_0}$ (see Definition B.6). □

## C.8 DERIVATIVE OF VECTOR FUNCTION $f(X)_{i_0} \circ (X^\top W_{*,j_1})$

**Lemma C.8.** *If the following holds:*

- *Let $f(X)_{i_0}$ be defined as Definition B.6*

*We have*

- **Part 1** *For $i_0 = i_1 = i_2 \in [n]$, $j_1, j_2 \in [d]$*

$$
\frac{\mathrm{d}f(X)_{i_0} \circ (X^\top W_{*,j_1})}{\mathrm{d}x_{i_2,j_2}}
$$
$$
= (-f(X)_{i_0} \cdot (f(X)_{i_0,i_0} \cdot w(X)_{i_0,j_2} + \langle f(X)_{i_0}, X^\top W_{*,j_2} \rangle)
$$
$$
+ f(X)_{i_0} \circ (e_{i_0} \cdot w(X)_{i_0,j_2} + X^\top W_{*,j_2})) \circ (X^\top W_{*,j_1}) + f(X)_{i_0} \circ (e_{i_0} \cdot w_{j_2,j_1})
$$

- **Part 2** *For $i_0 = i_1 \neq i_2 \in [n]$, $j_1, j_2 \in [d]$*

$$
\frac{\mathrm{d}f(X)_{i_0} \circ (X^\top W_{*,j_1})}{\mathrm{d}x_{i_2,j_2}}
$$
$$
= (-f(X)_{i_0} \cdot f(X)_{i_0,i_2} \cdot w(X)_{i_0,j_2}
$$
$$
+ f(X)_{i_0} \circ (e_{i_2} \cdot w(X)_{i_0,j_2})) \circ (X^\top W_{*,j_1}) + f(X)_{i_0} \circ (e_{i_2} \cdot w_{j_2,j_1})
$$

*Proof.* **Proof of Part 1**

$$
\frac{\mathrm{d}f(X)_{i_0} \circ (X^\top W_{*,j_1})}{\mathrm{d}x_{i_2,j_2}}
$$
$$
= \frac{\mathrm{d}f(X)_{i_0}}{\mathrm{d}x_{i_2,j_2}} \circ (X^\top W_{*,j_1}) + f(X)_{i_0} \circ \frac{\mathrm{d}X^\top W_{*,j_1}}{\mathrm{d}x_{i_2,j_2}}
$$
$$
= \frac{\mathrm{d}f(X)_{i_0}}{\mathrm{d}x_{i_2,j_2}} \circ (X^\top W_{*,j_1}) + f(X)_{i_0} \circ (e_{i_0} \cdot w_{j_2,j_1})
$$
$$
= (-(\alpha(X)_{i_0})^{-1} \cdot f(X)_{i_0} \cdot (u(X)_{i_0,i_0} \cdot w(X)_{i_0,j_2} + \langle u(X)_{i_0}, X^\top W_{*,j_2} \rangle)
$$
$$
+ f(X)_{i_0} \circ (e_{i_0} \cdot w(X)_{i_0,j_2} + X^\top W_{*,j_2})) \circ (X^\top W_{*,j_1}) + f(X)_{i_0} \circ (e_{i_0} \cdot w_{j_2,j_1})
$$
$$
= (-f(X)_{i_0} \cdot (f(X)_{i_0,i_0} \cdot w(X)_{i_0,j_2} + \langle f(X)_{i_0}, X^\top W_{*,j_2} \rangle)
$$
$$
+ f(X)_{i_0} \circ (e_{i_0} \cdot w(X)_{i_0,j_2} + X^\top W_{*,j_2})) \circ (X^\top W_{*,j_1}) + f(X)_{i_0} \circ (e_{i_0} \cdot w_{j_2,j_1})
$$

where the 1st step is by Fact B.2, the 2nd step uses Lemma C.2, the 3rd step uses Lemma B.14, the last step is by the definition of $f(X)_{i_0}$ (see Definition B.6).

**Proof of Part 2**

$$
\frac{\mathrm{d}f(X)_{i_0} \circ (X^\top W_{*,j_1})}{\mathrm{d}x_{i_2,j_2}}
$$
$$
= \frac{\mathrm{d}f(X)_{i_0}}{\mathrm{d}x_{i_2,j_2}} \circ (X^\top W_{*,j_1}) + f(X)_{i_0} \circ \frac{\mathrm{d}X^\top W_{*,j_1}}{\mathrm{d}x_{i_2,j_2}}
$$
$$
= \frac{\mathrm{d}f(X)_{i_0}}{\mathrm{d}x_{i_2,j_2}} \circ (X^\top W_{*,j_1}) + f(X)_{i_0} \circ (e_{i_2} \cdot w_{j_2,j_1})
$$
$$
= -((\alpha(X)_{i_0})^{-1} \cdot f(X)_{i_0} \cdot u(X)_{i_0,i_2} \cdot w(X)_{i_0,j_2}
$$
$$
+ f(X)_{i_0} \circ (e_{i_2} \cdot w(X)_{i_0,j_2})) \circ (X^\top W_{*,j_1}) + f(X)_{i_0} \circ (e_{i_2} \cdot w_{j_2,j_1})
$$
$$
= (-f(X)_{i_0} \cdot f(X)_{i_0,i_2} \cdot w(X)_{i_0,j_2}
$$
$$
+ f(X)_{i_0} \circ (e_{i_2} \cdot w(X)_{i_0,j_2})) \circ (X^\top W_{*,j_1}) + f(X)_{i_0} \circ (e_{i_2} \cdot w_{j_2,j_1})
$$

where the 1st step is by Fact B.2, the 2nd step uses Lemma C.2, the 3rd step uses Lemma B.14, the last step is by the definition of $f(X)_{i_0}$ (see Definition B.6). $\square$

## C.9 DERIVATIVE OF $C_1(X)$

**Lemma C.9.** *If the following holds:*

- *Let $C_1(X) \in \mathbb{R}$ be defined as in Lemma B.16*

- *Let $z(X)_{i_0,j_1} = \langle f(X)_{i_0}, X^\top W_{*,j_1} \rangle$*

Table 1: $C_1$ Part 1 Summary

| ID | Term | Symmetric? | Table Name |
|----|------|-----------|-----------|
| 1 | $+2s(X)_{i_0,j_0} \cdot f(X)_{i_0,i_0}^2 \cdot w(X)_{i_0,j_1} \cdot w(X)_{i_0,j_2}$ | Yes | N/A |
| 2 | $-f(X)_{i_0,i_0}^2 \cdot h(X)_{j_0,i_0} \cdot w(X)_{i_0,j_2} \cdot w(X)_{i_0,j_1}$ | Yes | N/A |
| 3 | $-f(X)_{i_0,i_0} \cdot \langle f(X)_{i_0} \circ (X^\top W_{*,j_2}), h(X)_{j_0} \rangle \cdot w(X)_{i_0,j_1}$ | No | Table 4: 1 |
| 4 | $-f(X)_{i_0,i_0}^2 \cdot v_{j_2,j_0} \cdot w(X)_{i_0,j_1}$ | No | Table 5: 1 |
| 5 | $-s(X)_{i_0,j_0} \cdot f(X)_{i_0,i_0} \cdot w(X)_{i_0,j_2} \cdot w(X)_{i_0,j_1}$ | Yes | N/A |
| 6 | $-s(X)_{i_0,j_0} \cdot f(X)_{i_0,i_0} \cdot \langle W_{*,j_2}, X_{*,i_0} \rangle \cdot w(X)_{i_0,j_1}$ | No | Table 2: 7 |
| 7 | $-s(X)_{i_0,j_0} \cdot f(X)_{i_0,i_0} \cdot w_{j_1,j_2}$ | No | Table 2: 9 |
| 8 | $2f(X)_{i_0,i_0} \cdot s(X)_{i_0,j_0} \cdot z(X)_{i_0,j_2} \cdot w(X)_{i_0,j_1}$ | No | Table 2: 1 |

- *Let* $w(X)_{i_0,j_1} = \langle W_{j_1,*}, X_{*,i_0} \rangle$

*We have*

- **Part 1** *For* $i_0 = i_1 = i_2 \in [n]$, $j_1, j_2 \in [d]$

$$\frac{\mathrm{d}C_1(X)}{\mathrm{d}x_{i_2,j_2}}$$
$$= + 2s(X)_{i_0,j_0} \cdot f(X)_{i_0,i_0}^2 \cdot w(X)_{i_0,j_2} \cdot w(X)_{i_0,j_1}$$
$$+ 2f(X)_{i_0,i_0} \cdot s(X)_{i_0,j_0} \cdot z(X)_{i_0,j_2} \cdot w(X)_{i_0,j_1}$$
$$- f(X)_{i_0,i_0}^2 \cdot h(X)_{j_0,i_0} \cdot w(X)_{i_0,j_2} \cdot w(X)_{i_0,j_1}$$
$$- f(X)_{i_0,i_0} \cdot \langle f(X)_{i_0} \circ (X^\top W_{*,j_2}), h(X)_{j_0} \rangle \cdot w(X)_{i_0,j_1}$$
$$- f(X)_{i_0,i_0}^2 \cdot v_{j_2,j_0} \cdot w(X)_{i_0,j_1}$$
$$- s(X)_{i_0,j_0} \cdot f(X)_{i_0,i_0} \cdot w(X)_{i_0,j_2} \cdot w(X)_{i_0,j_1}$$
$$- s(X)_{i_0,j_0} \cdot f(X)_{i_0,i_0} \cdot \langle W_{*,j_2}, X_{*,i_0} \rangle \cdot w(X)_{i_0,j_1}$$
$$- s(X)_{i_0,j_0} \cdot f(X)_{i_0,i_0} \cdot w_{j_1,j_2}$$

- **Part 2** *For* $i_0 = i_1 \neq i_2 \in [n]$, $j_1, j_2 \in [d]$

$$\frac{\mathrm{d}C_1(X)}{\mathrm{d}x_{i_2,j_2}}$$
$$= s(X)_{i_0,j_0} \cdot f(X)_{i_0,i_2} \cdot w(X)_{i_0,j_2} \cdot f(X)_{i_0,i_0} \cdot w(X)_{i_0,j_1}$$
$$- f(X)_{i_0,i_2} \cdot h(X)_{j_0,i_2} \cdot w(X)_{i_0,j_2} \cdot f(X)_{i_0,i_0} \cdot w(X)_{i_0,j_1}$$
$$- f(X)_{i_0,i_2} \cdot v_{j_2,j_0} \cdot f(X)_{i_0,i_0} \cdot w(X)_{i_0,j_1}$$
$$+ s(X)_{i_0,j_0} \cdot f(X)_{i_0,i_0} \cdot f(X)_{i_0,i_2} \cdot w(X)_{i_0,j_2} \cdot w(X)_{i_0,j_1}$$

*Proof.* **Proof of Part 1**

$$\frac{\mathrm{d}C_1(X)}{\mathrm{d}x_{i_2,j_2}}$$
$$= \frac{\mathrm{d} - s(X)_{i_0,j_0} \cdot f(X)_{i_0,i_0} \cdot w(X)_{i_0,j_1}}{\mathrm{d}x_{i_2,j_2}}$$
$$= - \frac{\mathrm{d}s(X)_{i_0,j_0}}{\mathrm{d}x_{i_2,j_2}} \cdot f(X)_{i_0,i_0} \cdot w(X)_{i_0,j_1}$$
$$- s(X)_{i_0,j_0} \cdot \frac{\mathrm{d}f(X)_{i_0,i_0} \cdot w(X)_{i_0,j_1}}{\mathrm{d}x_{i_2,j_2}}$$
$$= - \frac{\mathrm{d}s(X)_{i_0,j_0}}{\mathrm{d}x_{i_2,j_2}} \cdot f(X)_{i_0,i_0} \cdot w(X)_{i_0,j_1}$$

$$- s(X)_{i_0,j_0} \cdot ((-(\alpha(X)_{i_0})^{-1} \cdot f(X)_{i_0,i_0} \cdot (u(X)_{i_0,i_0} \cdot w(X)_{i_0,j_2} + \langle u(X)_{i_0}, X^\top W_{*,j_2} \rangle)$$
$$+ f(X)_{i_0,i_0} \cdot \langle W_{j_2,*} + W_{*,j_2}, X_{*,i_0} \rangle) \cdot w(X)_{i_0,j_1} + f(X)_{i_0,i_0} \cdot w_{j_1,j_2})$$
$$= - (-s(X)_{i_0,j_0} \cdot f(X)_{i_0,i_0} \cdot w(X)_{i_0,j_2} - s(X)_{i_0,j_0} \cdot \langle f(X)_{i_0}, X^\top W_{*,j_2} \rangle$$
$$+ f(X)_{i_0,i_0} \cdot h(X)_{j_0,i_0} \cdot w(X)_{i_0,j_2}$$
$$+ \langle f(X)_{i_0} \circ (X^\top W_{*,j_2}), h(X)_{j_0} \rangle + f(X)_{i_0,i_2} \cdot v_{j_2,j_0}) \cdot f(X)_{i_0,i_0} \cdot w(X)_{i_0,j_1}$$
$$- s(X)_{i_0,j_0} \cdot ((-f(X)_{i_0,i_0} \cdot (f(X)_{i_0,i_0} \cdot w(X)_{i_0,j_2} + \langle f(X)_{i_0}, X^\top W_{*,j_2} \rangle)$$
$$+ f(X)_{i_0,i_0} \cdot \langle W_{j_2,*} + W_{*,j_2}, X_{*,i_0} \rangle) \cdot w(X)_{i_0,j_1} + f(X)_{i_0,i_0} \cdot w_{j_1,j_2})$$
$$= 2s(X)_{i_0,j_0} \cdot f(X)_{i_0,i_0}^2 \cdot w(X)_{i_0,j_2} \cdot w(X)_{i_0,j_1}$$
$$+ 2s(X)_{i_0,j_0} \cdot Z(X)_{i_0,j_2} \cdot f(X)_{i_0,i_0} \cdot w(X)_{i_0,j_1}$$
$$- f(X)_{i_0,i_0}^2 \cdot h(X)_{j_0,i_0} \cdot w(X)_{i_0,j_2} \cdot w(X)_{i_0,j_1}$$
$$- f(X)_{i_0,i_0} \cdot \langle f(X)_{i_0} \circ (X^\top W_{*,j_2}), h(X)_{j_0} \rangle \cdot w(X)_{i_0,j_1}$$
$$- f(X)_{i_0,i_0}^2 \cdot v_{j_2,j_0} \cdot w(X)_{i_0,j_1}$$
$$- s(X)_{i_0,j_0} \cdot f(X)_{i_0,i_0} \cdot \langle W_{j_2,*} + W_{*,j_2}, X_{*,i_0} \rangle \cdot w(X)_{i_0,j_1}$$
$$- s(X)_{i_0,j_0} \cdot f(X)_{i_0,i_0} \cdot w_{j_1,j_2}$$

where the first step is by definition of $C_1(X)$ (see Lemma B.16), the 2nd step is by Fact B.2, the 3rd step is by Lemma C.7, the 4th step is because Lemma B.16, the 5th step is a rearrangement.

**Proof of Part 2**

$$\frac{\mathrm{d}C_1(X)}{\mathrm{d}x_{i_2,j_2}}$$
$$= \frac{\mathrm{d} - s(X)_{i_0,j_0} \cdot f(X)_{i_0,i_0} \cdot w(X)_{i_0,j_1}}{\mathrm{d}x_{i_2,j_2}}$$
$$= - \frac{\mathrm{d}s(X)_{i_0,j_0}}{\mathrm{d}x_{i_2,j_2}} \cdot f(X)_{i_0,i_0} \cdot w(X)_{i_0,j_1}$$
$$\quad - s(X)_{i_0,j_0} \cdot \frac{\mathrm{d}f(X)_{i_0,i_0} \cdot w(X)_{i_0,j_1}}{\mathrm{d}x_{i_2,j_2}}$$
$$= - \frac{\mathrm{d}s(X)_{i_0,j_0}}{\mathrm{d}x_{i_2,j_2}} \cdot f(X)_{i_0,i_0} \cdot w(X)_{i_0,j_1}$$
$$\quad + s(X)_{i_0,j_0} \cdot f(X)_{i_0,i_0} \cdot f(X)_{i_0,i_2} \cdot w(X)_{i_0,j_2} \cdot w(X)_{i_0,j_1}$$
$$= - (-s(X)_{i_0,j_0} \cdot f(X)_{i_0,i_2} \cdot w(X)_{i_0,j_2} + f(X)_{i_0,i_2} \cdot h(X)_{j_0,i_2} \cdot w(X)_{i_0,j_2}$$
$$\quad + f(X)_{i_0,i_2} \cdot v_{j_2,j_0}) \cdot f(X)_{i_0,i_0} \cdot w(X)_{i_0,j_1}$$
$$\quad + s(X)_{i_0,j_0} \cdot f(X)_{i_0,i_0} \cdot f(X)_{i_0,i_2} \cdot w(X)_{i_0,j_2} \cdot w(X)_{i_0,j_1}$$
$$= s(X)_{i_0,j_0} \cdot f(X)_{i_0,i_2} \cdot w(X)_{i_0,j_2} \cdot f(X)_{i_0,i_0} \cdot w(X)_{i_0,j_1}$$
$$\quad - f(X)_{i_0,i_2} \cdot h(X)_{j_0,i_2} \cdot w(X)_{i_0,j_2} \cdot f(X)_{i_0,i_0} \cdot w(X)_{i_0,j_1}$$
$$\quad - f(X)_{i_0,i_2} \cdot v_{j_2,j_0} \cdot f(X)_{i_0,i_0} \cdot w(X)_{i_0,j_1}$$
$$\quad + s(X)_{i_0,j_0} \cdot f(X)_{i_0,i_0} \cdot f(X)_{i_0,i_2} \cdot w(X)_{i_0,j_2} \cdot w(X)_{i_0,j_1}$$

where the first step is by definition of $C_1(X)$ (see Lemma B.16), the 2nd step is by Fact B.2, the 3rd step is by Lemma C.7, the 4th step is because Lemma B.16, the 5th step is a rearrangement. $\square$

## C.10 DERIVATIVE OF $C_2(X)$

**Lemma C.10.** *If the following holds:*

- *Let $C_2(X)$ be defined as in Lemma B.16*

- *We define $z(X)_{i_0,j_1} := \langle f(X)_{i_0}, X^\top W_{*,j_1} \rangle$.*

*We have*

Table 2: $C_2$ Part 1 Summary

| ID | Term | Symmetric Terms | Table Name |
|---|---|---|---|
| 1 | $2s(X)_{i_0,j_0} \cdot f(X)_{i_0,i_0} \cdot w(X)_{i_0,j_2} \cdot z(X)_{i_0,j_1}$ | No | Table 1: 9 |
| 2 | $s(X)_{i_0,j_0} \cdot z(X)_{i_0,j_2} \cdot z(X)_{i_0,j_1}$ | Yes | N/A |
| 3 | $-f(X)_{i_0,i_0} \cdot h(X)_{j_0,i_0} \cdot w(X)_{i_0,j_2} \cdot z(X)_{i_0,j_1}$ | No | Table 3: 3 |
| 4 | $-\langle f(X)_{i_0} \circ (X^\top W_{*,j_2}), h(X)_{j_0}\rangle \cdot z(X)_{i_0,j_1}$ | No | Table 4: 2 |
| 5 | $-f(X)_{i_0,i_0} \cdot v_{j_2,j_0} \cdot z(X)_{i_0,j_1}$ | No | Table 5: 2 |
| 6 | $+s(X)_{i_0,j_0} \cdot z(X)_{i_0,j_1} \cdot f(X)_{i_0,i_0} \cdot z(X)_{i_0,j_2}$ | Yes | N/A |
| 7 | $-s(X)_{i_0,j_0} \cdot f(X)_{i_0,i_0} \cdot \langle W_{*,j_1}, X_{*,i_0}\rangle \cdot w(X)_{i_0,j_2}$ | No | Table 1: 6 |
| 8 | $-s(X)_{i_0,j_0} \cdot \langle f(X)_{i_0} \circ (X^\top W_{*,j_2}), X^\top W_{*,j_1}\rangle$ | Yes | N/A |
| 9 | $-s(X)_{i_0,j_0} \cdot f(X)_{i_0,i_0} \cdot w_{j_2,j_1}$ | No | Table 1: 7 |

- **Part 1** *For $i_0 = i_1 = i_2 \in [n]$, $j_1, j_2 \in [d]$*

$$
\frac{\mathrm{d}C_2(X)}{\mathrm{d}x_{i_2,j_2}}
$$
$$
\begin{aligned}
= \; &+ 2s(X)_{i_0,j_0} \cdot f(X)_{i_0,i_0} \cdot w(X)_{i_0,j_2} \cdot z(X)_{i_0,j_1} \\
&+ s(X)_{i_0,j_0} \cdot z(X)_{i_0,j_2} \cdot z(X)_{i_0,j_1} \\
&- f(X)_{i_0,i_0} \cdot h(X)_{j_0,i_0} \cdot w(X)_{i_0,j_2} \cdot z(X)_{i_0,j_1} \\
&- \langle f(X)_{i_0} \circ (X^\top W_{*,j_2}), h(X)_{j_0}\rangle \cdot z(X)_{i_0,j_1} \\
&- f(X)_{i_0,i_0} \cdot v_{j_2,j_0} \cdot z(X)_{i_0,j_1} \\
&+ s(X)_{i_0,j_0} \cdot z(X)_{i_0,j_1} \cdot f(X)_{i_0,i_0} \cdot z(X)_{i_0,j_2} \\
&- s(X)_{i_0,j_0} \cdot f(X)_{i_0,i_0} \cdot \langle W_{*,j_1}, X_{*,i_0}\rangle \cdot w(X)_{i_0,j_2} \\
&- s(X)_{i_0,j_0} \cdot \langle f(X)_{i_0} \circ (X^\top W_{*,j_2}), X^\top W_{*,j_1}\rangle \\
&- s(X)_{i_0,j_0} \cdot f(X)_{i_0,i_0} \cdot w_{j_2,j_1}
\end{aligned}
$$

- **Part 2** *For $i_0 = i_1 \neq i_2 \in [n]$, $j_1, j_2 \in [d]$*

$$
\frac{\mathrm{d}C_2(X)}{\mathrm{d}x_{i_2,j_2}}
$$
$$
\begin{aligned}
= \; &+ s(X)_{i_0,j_0} \cdot f(X)_{i_0,i_2} \cdot w(X)_{i_0,j_2} \cdot z(X)_{i_0,j_1} \\
&- f(X)_{i_0,i_2} \cdot h(X)_{j_0,i_2} \cdot w(X)_{i_0,j_2} \cdot z(X)_{i_0,j_1} \\
&- f(X)_{i_0,i_2} \cdot v_{j_2,j_0} \cdot z(X)_{i_0,j_1} \\
&+ s(X)_{i_0,j_0} \cdot \langle f(X)_{i_0}, X^\top W_{*,j_1}\rangle \cdot f(X)_{i_0,i_0} \cdot w(X)_{i_0,j_2} \\
&- s(X)_{i_0,j_0} \cdot f(X)_{i_0,i_0} \cdot \langle W_{*,j_1}, X_{*,i_0}\rangle \cdot w(X)_{i_0,j_2} \\
&- s(X)_{i_0,j_0} \cdot f(X)_{i_0,i_0} \cdot w_{j_2,j_1}
\end{aligned}
$$

*Proof.* **Proof of Part 1**

$$
\frac{\mathrm{d} - C_2(X)}{\mathrm{d}x_{i_2,j_2}}
$$
$$
= \frac{\mathrm{d}s(X)_{i_0,j_0} \cdot z(X)_{i_0,j_1}}{\mathrm{d}x_{i_2,j_2}}
$$
$$
= \frac{\mathrm{d}s(X)_{i_0,j_0}}{\mathrm{d}x_{i_2,j_2}} \cdot z(X)_{i_0,j_1} + s(X)_{i_0,j_0} \cdot \frac{\mathrm{d}z(X)_{i_0,j_1}}{\mathrm{d}x_{i_2,j_2}}
$$
$$
\begin{aligned}
= \; &\frac{\mathrm{d}s(X)_{i_0,j_0}}{\mathrm{d}x_{i_2,j_2}} \cdot z(X)_{i_0,j_1} \\
&+ s(X)_{i_0,j_0} \cdot ((\langle -(\alpha(X)_{i_0})^{-1} \cdot f(X)_{i_0} \cdot (u(X)_{i_0,i_0} \cdot w(X)_{i_0,j_2} + \langle u(X)_{i_0}, X^\top W_{*,j_2}\rangle) \\
&+ f(X)_{i_0} \circ (e_{i_0} \cdot w(X)_{i_0,j_2} + X^\top W_{*,j_2}), X^\top W_{*,j_1}\rangle + f(X)_{i_0,i_0} \cdot w_{j_2,j_1})
\end{aligned}
$$

$$
\begin{aligned}
= &(-s(X)_{i_0,j_0} \cdot f(X)_{i_0,i_0} \cdot w(X)_{i_0,j_2} - s(X)_{i_0,j_0} \cdot \langle f(X)_{i_0}, X^\top W_{*,j_2} \rangle \\
&+ f(X)_{i_0,i_0} \cdot h(X)_{j_0,i_0} \cdot w(X)_{i_0,j_2} \\
&+ \langle f(X)_{i_0} \circ (X^\top W_{*,j_2}), h(X)_{j_0} \rangle + f(X)_{i_0,i_2} \cdot v_{j_2,j_0}) \cdot z(X)_{i_0,j_1} \\
&+ s(X)_{i_0,j_0} \cdot ((\langle -(\alpha(X)_{i_0})^{-1} \cdot f(X)_{i_0} \cdot (u(X)_{i_0,i_0} \cdot w(X)_{i_0,j_2} + \langle u(X)_{i_0}, X^\top W_{*,j_2} \rangle) \\
&+ f(X)_{i_0} \circ (e_{i_0} \cdot w(X)_{i_0,j_2} + X^\top W_{*,j_2}), X^\top W_{*,j_1} \rangle + f(X)_{i_0,i_0} \cdot w_{j_2,j_1}) \\
= & -s(X)_{i_0,j_0} \cdot f(X)_{i_0,i_0} \cdot w(X)_{i_0,j_2} \cdot z(X)_{i_0,j_1} \\
&- s(X)_{i_0,j_0} \cdot z(X)_{i_0,j_2} \cdot z(X)_{i_0,j_1} \\
&+ f(X)_{i_0,i_0} \cdot h(X)_{j_0,i_0} \cdot w(X)_{i_0,j_2} \cdot z(X)_{i_0,j_1} \\
&+ \langle f(X)_{i_0} \circ (X^\top W_{*,j_2}), h(X)_{j_0} \rangle \cdot z(X)_{i_0,j_1} \\
&+ f(X)_{i_0,i_2} \cdot v_{j_2,j_0} \cdot z(X)_{i_0,j_1} \\
&- s(X)_{i_0,j_0} \cdot \langle f(X)_{i_0}, X^\top W_{*,j_1} \rangle \cdot f(X)_{i_0,i_0} \cdot w(X)_{i_0,j_2} \\
&- s(X)_{i_0,j_0} \cdot \langle f(X)_{i_0}, X^\top W_{*,j_1} \rangle \cdot f(X)_{i_0,i_0} \cdot \langle f(X)_{i_0}, X^\top W_{*,j_2} \rangle \\
&+ s(X)_{i_0,j_0} \cdot f(X)_{i_0,i_0} \cdot \langle W_{*,j_1}, X_{*,i_0} \rangle \cdot w(X)_{i_0,j_2} \\
&+ s(X)_{i_0,j_0} \cdot \langle f(X)_{i_0} \circ (X^\top W_{*,j_2}), X^\top W_{*,j_1} \rangle \\
&+ s(X)_{i_0,j_0} \cdot f(X)_{i_0,i_0} \cdot w_{j_2,j_1}
\end{aligned}
$$

where the first step is by definition of $C_2(X)$ (see Lemma B.16), the 2nd step is by Fact B.2, the 3rd step is by Lemma C.5, the 4th step is because Lemma B.16, the 5th step is a rearrangement.

**Proof of Part 2**

$$
\begin{aligned}
&\frac{\mathrm{d} - C_2(X)}{\mathrm{d}x_{i_2,j_2}} \\
= &\frac{\mathrm{d}s(X)_{i_0,j_0} \cdot \langle f(X)_{i_0}, X^\top W_{*,j_1} \rangle}{\mathrm{d}x_{i_2,j_2}} \\
= &\frac{\mathrm{d}s(X)_{i_0,j_0}}{\mathrm{d}x_{i_2,j_2}} \cdot z(X)_{i_0,j_1} + s(X)_{i_0,j_0} \cdot \frac{\mathrm{d}\langle f(X)_{i_0}, X^\top W_{*,j_1} \rangle}{\mathrm{d}x_{i_2,j_2}} \\
= &\frac{\mathrm{d}s(X)_{i_0,j_0}}{\mathrm{d}x_{i_2,j_2}} \cdot z(X)_{i_0,j_1} \\
&+ s(X)_{i_0,j_0} \cdot ((\langle -(\alpha(X)_{i_0})^{-1} \cdot f(X)_{i_0} \cdot u(X)_{i_0,i_0} \cdot w(X)_{i_0,j_2} \\
&+ f(X)_{i_0} \circ (e_{i_0} \cdot w(X)_{i_0,j_2}), X^\top W_{*,j_1} \rangle + f(X)_{i_0,i_0} \cdot w_{j_2,j_1}) \\
= &(-s(X)_{i_0,j_0} \cdot f(X)_{i_0,i_2} \cdot w(X)_{i_0,j_2} + f(X)_{i_0,i_2} \cdot h(X)_{j_0,i_2} \cdot w(X)_{i_0,j_2} \\
&+ f(X)_{i_0,i_2} \cdot v_{j_2,j_0}) \cdot z(X)_{i_0,j_1} \\
&+ s(X)_{i_0,j_0} \cdot ((\langle -(\alpha(X)_{i_0})^{-1} \cdot f(X)_{i_0} \cdot u(X)_{i_0,i_0} \cdot w(X)_{i_0,j_2} \\
&+ f(X)_{i_0} \circ (e_{i_0} \cdot w(X)_{i_0,j_2}), X^\top W_{*,j_1} \rangle + f(X)_{i_0,i_0} \cdot w_{j_2,j_1}) \\
= & -s(X)_{i_0,j_0} \cdot f(X)_{i_0,i_2} \cdot w(X)_{i_0,j_2} \cdot z(X)_{i_0,j_1} \\
&+ f(X)_{i_0,i_2} \cdot h(X)_{j_0,i_2} \cdot w(X)_{i_0,j_2} \cdot z(X)_{i_0,j_1} \\
&+ f(X)_{i_0,i_2} \cdot v_{j_2,j_0} \cdot z(X)_{i_0,j_1} \\
&- s(X)_{i_0,j_0} \cdot \langle f(X)_{i_0}, X^\top W_{*,j_1} \rangle \cdot f(X)_{i_0,i_0} \cdot w(X)_{i_0,j_2} \\
&+ s(X)_{i_0,j_0} \cdot f(X)_{i_0,i_0} \cdot \langle W_{*,j_1}, X_{*,i_0} \rangle \cdot w(X)_{i_0,j_2} \\
&+ s(X)_{i_0,j_0} \cdot f(X)_{i_0,i_0} \cdot w_{j_2,j_1}
\end{aligned}
$$

where the first step is by definition of $C_2(X)$ (see Lemma B.16), the 2nd step is by Fact B.2, the 3rd step is by Lemma C.5, the 4th step is because Lemma B.16, the 5th step is a rearrangement. $\qquad\square$

## C.11 Derivative of $C_3(X)$

**Lemma C.11.** *If the following holds:*

- *Let $C_3(X)$ be defined as in Lemma B.16*

Table 3: $C_3$ Part 1 Summary

| ID | Term | Symmetric Terms | Table Name |
|----|------|-----------------|------------|
| 1 | $-f(X)_{i_0,i_0}^2 \cdot h(X)_{j_0,i_0} \cdot w(X)_{i_0,j_2} \cdot w(X)_{i_0,j_1}$ | Yes | N/A |
| 2 | $f(X)_{i_0,i_0} \cdot w(X)_{i_0,j_2} \cdot h(X)_{j_0,i_0} \cdot w(X)_{i_0,j_1}$ | Yes | N/A |
| 3 | $-f(X)_{i_0,i_0} \cdot z(X)_{i_0,j_2} \cdot h(X)_{j_0,i_0} \cdot w(X)_{i_0,j_1}$ | No | Table 2: 3 |
| 4 | $f(X)_{i_0,i_0} \cdot \langle W_{*,j_2}, X_{*,i_0} \rangle \cdot h(X)_{j_0,i_0} \cdot w(X)_{i_0,j_1}$ | No | Table 4: 3 |
| 5 | $f(X)_{i_0,i_0} \cdot v_{j_2,j_0} \cdot w(X)_{i_0,j_1}$ | No | Table 5: 3 |
| 6 | $f(X)_{i_0,i_0} \cdot h(X)_{i_0,i_0} \cdot w_{j_1,j_2}$ | No | Table 4: 5 |

*We have*

- **Part 1** *For $i_0 = i_1 = i_2 \in [n]$, $j_1, j_2 \in [d]$*

$$
\frac{\mathrm{d}C_3(X)}{\mathrm{d}x_{i_2,j_2}}
$$
$$
\begin{aligned}
= & - f(X)_{i_0,i_0}^2 \cdot h(X)_{j_0,i_0} \cdot w(X)_{i_0,j_2} \cdot w(X)_{i_0,j_1} \\
& - f(X)_{i_0,i_0} \cdot z(X)_{i_0,j_2} \cdot h(X)_{j_0,i_0} \cdot w(X)_{i_0,j_1} \\
& + f(X)_{i_0,i_0} \cdot w(X)_{i_0,j_2} \cdot h(X)_{j_0,i_0} \cdot w(X)_{i_0,j_1} \\
& + f(X)_{i_0,i_0} \cdot \langle W_{*,j_2}, X_{*,i_0} \rangle \cdot h(X)_{j_0,i_0} \cdot w(X)_{i_0,j_1} \\
& + f(X)_{i_0,i_0} \cdot v_{j_2,j_0} \cdot w(X)_{i_0,j_1} \\
& + f(X)_{i_0,i_0} \cdot h(X)_{i_0,i_0} \cdot w_{j_1,j_2}
\end{aligned}
$$

- **Part 2** *For $i_0 = i_1 \neq i_2 \in [n]$, $j_1, j_2 \in [d]$*

$$
\frac{\mathrm{d}C_3(X)}{\mathrm{d}x_{i_2,j_2}}
$$
$$
= - f(X)_{i_0,i_0} \cdot f(X)_{i_0,i_2} \cdot w(X)_{i_0,j_2} \cdot h(X)_{j_0,i_0} \cdot w(X)_{i_0,j_1}
$$

*Proof.* **Proof of Part 1**

$$
\frac{\mathrm{d}C_3(X)}{\mathrm{d}x_{i_2,j_2}}
$$
$$
= \frac{\mathrm{d}f(X)_{i_0,i_0} \cdot h(X)_{i_0,i_0} \cdot w(X)_{i_0,j_1}}{\mathrm{d}x_{i_2,j_2}}
$$
$$
= \frac{\mathrm{d}f(X)_{i_0,i_0} \cdot h(X)_{i_0,i_0}}{\mathrm{d}x_{i_2,j_2}} \cdot w(X)_{i_0,j_1} + f(X)_{i_0,i_0} \cdot h(X)_{i_0,i_0} \cdot \frac{\mathrm{d}w(X)_{i_0,j_1}}{\mathrm{d}x_{i_2,j_2}}
$$
$$
= \frac{\mathrm{d}f(X)_{i_0,i_0} \cdot h(X)_{i_0,i_0}}{\mathrm{d}x_{i_2,j_2}} \cdot w(X)_{i_0,j_1} + f(X)_{i_0,i_0} \cdot h(X)_{i_0,i_0} \cdot w_{j_1,j_2}
$$
$$
\begin{aligned}
= & ((-f(X)_{i_0,i_0} \cdot (f(X)_{i_0,i_0} \cdot w(X)_{i_0,j_2} + \langle f(X)_{i_0}, X^\top W_{*,j_2} \rangle) \\
& + f(X)_{i_0,i_0} \cdot \langle W_{j_2,*} + W_{*,j_2}, X_{*,i_0} \rangle) \cdot h(X)_{j_0,i_0} + f(X)_{i_0,i_0} \cdot v_{j_2,j_0}) \cdot w(X)_{i_0,j_1} \\
& + f(X)_{i_0,i_0} \cdot h(X)_{i_0,i_0} \cdot w_{j_1,j_2}
\end{aligned}
$$
$$
\begin{aligned}
= & - f(X)_{i_0,i_0}^2 \cdot h(X)_{j_0,i_0} \cdot w(X)_{i_0,j_2} \cdot w(X)_{i_0,j_1} \\
& - f(X)_{i_0,i_0} \cdot Z(X)_{i_0,j_2} \cdot h(X)_{j_0,i_0} \cdot w(X)_{i_0,j_1} \\
& + f(X)_{i_0,i_0} \cdot \langle W_{j_2,*} + W_{*,j_2}, X_{*,i_0} \rangle \cdot h(X)_{j_0,i_0} \cdot w(X)_{i_0,j_1} \\
& + f(X)_{i_0,i_0} \cdot v_{j_2,j_0} \cdot w(X)_{i_0,j_1} \\
& + f(X)_{i_0,i_0} \cdot h(X)_{i_0,i_0} \cdot w_{j_1,j_2}
\end{aligned}
$$

where the first step is by definition of $C_3(X)$ (see Lemma B.16), the 2nd step is by Fact B.2, the 3rd step is by Lemma C.1, the 4th step is because Lemma C.6, the 5th step is a rearrangement.

Table 4: $C_4$ Part 1 Summary

| ID | Term | Symmetric? | Table Name |
|----|------|-----------|-----------|
| 1 | $-\langle f(X)_{i_0} \circ (X^\top W_{*,j_1}), h(X)_{j_0}\rangle \cdot f(X)_{i_0,i_0} \cdot w(X)_{i_0,j_2}$ | No | Table 1: 3 |
| 2 | $-\langle f(X)_{i_0} \circ (X^\top W_{*,j_1}), h(X)_{j_0}\rangle \cdot Z(X)_{i_0,j_2}$ | No | Table 2: 4 |
| 3 | $f(X)_{i_0,i_0} \cdot h(X)_{j_0,i_0} \cdot \langle W_{*,j_1}, X_{*,i_0}\rangle \cdot w(X)_{i_0,j_2}$ | No | Table 3: 4 |
| 4 | $\langle f(X)_{i_0} \circ (X^\top W_{*,j_2}) \circ (X^\top W_{*,j_1}), h(X)_{j_0}\rangle$ | Yes | N/A |
| 5 | $f(X)_{i_0,i_0} \cdot h(X)_{j_0,i_0} \cdot w_{j_2,j_1}$ | No | Table 3: 6 |
| 6 | $f(X)_{i_0,i_0} \cdot \langle W_{*,j_1}, X_{*,i_0}\rangle \cdot v_{j_2,j_0}$ | No | Table 5:4 |

**Proof of Part 2**

$$\frac{\mathrm{d}C_3(X)}{\mathrm{d}x_{i_2,j_2}}$$

$$= \frac{\mathrm{d}f(X)_{i_0,i_0} \cdot h(X)_{i_0,i_0} \cdot w(X)_{i_0,j_1}}{\mathrm{d}x_{i_2,j_2}}$$

$$= \frac{\mathrm{d}f(X)_{i_0,i_0} \cdot h(X)_{i_0,i_0}}{\mathrm{d}x_{i_2,j_2}} \cdot w(X)_{i_0,j_1} + f(X)_{i_0,i_0} \cdot h(X)_{i_0,i_0} \cdot \frac{\mathrm{d}w(X)_{i_0,j_1}}{\mathrm{d}x_{i_2,j_2}}$$

$$= \frac{\mathrm{d}f(X)_{i_0,i_0} \cdot h(X)_{i_0,i_0}}{\mathrm{d}x_{i_2,j_2}} \cdot w(X)_{i_0,j_1}$$

$$= - f(X)_{i_0,i_0} \cdot f(X)_{i_0,i_2} \cdot w(X)_{i_0,j_2} \cdot h(X)_{j_0,i_0} \cdot w(X)_{i_0,j_1}$$

where the first step is by definition of $C_3(X)$ (see Lemma B.16), the 2nd step is by Fact B.2, the 3rd step is by Lemma C.1, the 4th step is because Lemma C.6, the 5th step is a rearrangement.

$\square$

## C.12 DERIVATIVE OF $C_4(X)$

**Lemma C.12.** *If the following holds:*

- *Let $C_4(X)$ be defined as in Lemma B.16*

*We have*

- **Part 1** *For $i_0 = i_1 = i_2 \in [n]$, $j_1, j_2 \in [d]$*

$$\frac{\mathrm{d}C_4(X)}{\mathrm{d}x_{i_2,j_2}}$$

$$= - \langle f(X)_{i_0} \circ (X^\top W_{*,j_1}), h(X)_{j_0}\rangle \cdot f(X)_{i_0,i_0} \cdot w(X)_{i_0,j_2}$$
$$- \langle f(X)_{i_0} \circ (X^\top W_{*,j_1}), h(X)_{j_0}\rangle \cdot Z(X)_{i_0,j_2}$$
$$+ f(X)_{i_0,i_0} \cdot h(X)_{j_0,i_0} \cdot \langle W_{*,j_1}, X_{*,i_0}\rangle \cdot w(X)_{i_0,j_2}$$
$$+ \langle f(X)_{i_0} \circ (X^\top W_{*,j_2}) \circ (X^\top W_{*,j_1}), h(X)_{j_0}\rangle$$
$$+ f(X)_{i_0,i_0} \cdot h(X)_{j_0,i_0} \cdot w_{j_2,j_1}$$
$$+ f(X)_{i_0,i_0} \cdot \langle W_{*,j_1}, X_{*,i_0}\rangle \cdot v_{j_2,j_0}$$

- **Part 2** *For $i_0 = i_1 \neq i_2 \in [n]$, $j_1, j_2 \in [d]$*

$$\frac{\mathrm{d}C_4(X)}{\mathrm{d}x_{i_2,j_2}}$$

$$= - \langle f(X)_{i_0} \circ (X^\top W_{*,j_1}), h(X)_{j_0}\rangle \cdot f(X)_{i_0,i_2} \cdot w(X)_{i_0,j_2}$$
$$+ f(X)_{i_0,i_2} \cdot h(X)_{j_0,i_2} \cdot \langle W_{*,j_1}, X_{*,i_2}\rangle \cdot w(X)_{i_0,j_2}$$
$$+ f(X)_{i_0,i_2} \cdot h(X)_{j_0,i_2} \cdot w_{j_2,j_1}$$
$$+ f(X)_{i_0,i_2} \cdot \langle W_{*,j_1}, X_{*,i_2}\rangle \cdot v_{j_2,j_0}$$

*Proof.* **Proof of Part 1**

$$\frac{\mathrm{d}C_4(X)}{\mathrm{d}x_{i_2,j_2}}$$

$$= \frac{\mathrm{d}\langle f(X)_{i_0} \circ (X^\top W_{*,j_1}), h(X)_{j_0}\rangle}{\mathrm{d}x_{i_2,j_2}}$$

$$= \langle \frac{\mathrm{d}f(X)_{i_0} \circ (X^\top W_{*,j_1})}{\mathrm{d}x_{i_2,j_2}}, h(X)_{j_0}\rangle + \langle f(X)_{i_0} \circ (X^\top W_{*,j_1}), \frac{\mathrm{d}h(X)_{j_0}}{\mathrm{d}x_{i_2,j_2}}\rangle$$

$$= \langle \frac{\mathrm{d}f(X)_{i_0} \circ (X^\top W_{*,j_1})}{\mathrm{d}x_{i_2,j_2}}, h(X)_{j_0}\rangle + \langle f(X)_{i_0} \circ (X^\top W_{*,j_1}), e_{i_2} \cdot v_{j_2,j_0}\rangle$$

$$= \langle (-f(X)_{i_0} \cdot (f(X)_{i_0,i_0} \cdot w(X)_{i_0,j_2} + \langle f(X)_{i_0}, X^\top W_{*,j_2}\rangle)$$
$$\quad + f(X)_{i_0} \circ (e_{i_0} \cdot w(X)_{i_0,j_2} + X^\top W_{*,j_2})) \circ (X^\top W_{*,j_1}) + f(X)_{i_0} \circ (e_{i_0} \cdot w_{j_2,j_1}), h(X)_{j_0}\rangle$$
$$\quad + \langle f(X)_{i_0} \circ (X^\top W_{*,j_1}), e_{i_0} \cdot v_{j_2,j_0}\rangle$$

$$= -\langle f(X)_{i_0} \circ (X^\top W_{*,j_1}), h(X)_{j_0}\rangle \cdot f(X)_{i_0,i_0} \cdot w(X)_{i_0,j_2}$$
$$\quad - \langle f(X)_{i_0} \circ (X^\top W_{*,j_1}), h(X)_{j_0}\rangle \cdot \langle f(X)_{i_0}, X^\top W_{*,j_2}\rangle$$
$$\quad + f(X)_{i_0,i_0} \cdot h(X)_{j_0,i_0} \cdot \langle W_{*,j_1}, X_{*,i_0}\rangle \cdot w(X)_{i_0,j_2}$$
$$\quad + \langle f(X)_{i_0} \circ (X^\top W_{*,j_2}) \circ (X^\top W_{*,j_1}), h(X)_{j_0}\rangle$$
$$\quad + f(X)_{i_0,i_0} \cdot h(X)_{j_0,i_0} \cdot w_{j_2,j_1}$$
$$\quad + f(X)_{i_0,i_0} \cdot \langle W_{*,j_1}, X_{*,i_0}\rangle \cdot v_{j_2,j_0}$$

where the first step is by definition of $C_4(X)$ (see Lemma B.16), the 2nd step is by Fact B.2, the 3rd step is by Lemma B.15, the 4th step is because Lemma C.8, the 5th step is a rearrangement.

**Proof of Part 2**

$$\frac{\mathrm{d}C_4(X)}{\mathrm{d}x_{i_2,j_2}}$$

$$= \frac{\mathrm{d}\langle f(X)_{i_0} \circ (X^\top W_{*,j_1}), h(X)_{j_0}\rangle}{\mathrm{d}x_{i_2,j_2}}$$

$$= \langle \frac{\mathrm{d}f(X)_{i_0} \circ (X^\top W_{*,j_1})}{\mathrm{d}x_{i_2,j_2}}, h(X)_{j_0}\rangle + \langle f(X)_{i_0} \circ (X^\top W_{*,j_1}), \frac{\mathrm{d}h(X)_{j_0}}{\mathrm{d}x_{i_2,j_2}}\rangle$$

$$= \langle \frac{\mathrm{d}f(X)_{i_0} \circ (X^\top W_{*,j_1})}{\mathrm{d}x_{i_2,j_2}}, h(X)_{j_0}\rangle + \langle f(X)_{i_0} \circ (X^\top W_{*,j_1}), e_{i_2} \cdot v_{j_2,j_0}\rangle$$

$$= \langle -(f(X)_{i_0} \cdot f(X)_{i_0,i_2} \cdot w(X)_{i_0,j_2}$$
$$\quad + f(X)_{i_0} \circ (e_{i_2} \cdot w(X)_{i_0,j_2})) \circ (X^\top W_{*,j_1}) + f(X)_{i_0} \circ (e_{i_2} \cdot w_{j_2,j_1}), h(X)_{j_0}\rangle$$
$$\quad + \langle f(X)_{i_0} \circ (X^\top W_{*,j_1}), e_{i_2} \cdot v_{j_2,j_0}\rangle$$

$$= -\langle f(X)_{i_0} \circ (X^\top W_{*,j_1}), h(X)_{j_0}\rangle \cdot f(X)_{i_0,i_2} \cdot w(X)_{i_0,j_2}$$
$$\quad + f(X)_{i_0,i_2} \cdot h(X)_{j_0,i_2} \cdot \langle W_{*,j_1}, X_{*,i_2}\rangle \cdot w(X)_{i_0,j_2}$$
$$\quad + f(X)_{i_0,i_2} \cdot h(X)_{j_0,i_2} \cdot w_{j_2,j_1}$$
$$\quad + f(X)_{i_0,i_2} \cdot \langle W_{*,j_1}, X_{*,i_2}\rangle \cdot v_{j_2,j_0}$$

where the first step is by definition of $C_4(X)$ (see Lemma B.16), the 2nd step is by Fact B.2, the 3rd step is by Lemma B.15, the 4th step is because Lemma C.8, the 5th step is a rearrangement.

$\square$

## C.13 DERIVATIVE OF $C_5(X)$

**Lemma C.13.** *If the following holds:*

- *Let $C_5(X)$ be defined as in Lemma B.16*

Table 5: $C_5$ Part 1 Summary

| Term | Symmetric Terms | Table Name |
|---|---|---|
| $-f(X)_{i_0,i_0}^2 \cdot w(X)_{i_0,j_2} \cdot v_{j_1,j_0}$ | No | $C_1(X):4$ |
| $-f(X)_{i_0,i_0} \cdot z(X)_{i_0,j_2} \cdot v_{j_1,j_0}$ | No | Table 2: 5 |
| $f(X)_{i_0,i_0} \cdot w(X)_{i_0,j_2} \cdot v_{j_1,j_0}$ | No | Table 3:5 |
| $f(X)_{i_0,i_0} \cdot \langle W_{*,j_2}, X_{*,i_0} \rangle \cdot v_{j_1,j_0}$ | No | Table 4: 6 |

*We have*

- **Part 1** *For $i_0 = i_1 = i_2 \in [n]$, $j_1, j_2 \in [d]$*

$$
\begin{aligned}
\frac{\mathrm{d}C_5(X)}{\mathrm{d}x_{i_2,j_2}} = & -f(X)_{i_0,i_0}^2 \cdot w(X)_{i_0,j_2} \cdot v_{j_1,j_0} \\
& -f(X)_{i_0,i_0} \cdot z(X)_{i_0,j_2} \cdot v_{j_1,j_0} \\
& +f(X)_{i_0,i_0} \cdot w(X)_{i_0,j_2} \cdot v_{j_1,j_0} \\
& +f(X)_{i_0,i_0} \cdot \langle W_{*,j_2}, X_{*,i_0} \rangle \cdot v_{j_1,j_0}
\end{aligned}
$$

- **Part 2** *For $i_0 = i_1 \neq i_2 \in [n]$, $j_1, j_2 \in [d]$*

$$
\frac{\mathrm{d}C_5(X)}{\mathrm{d}x_{i_2,j_2}} = -f(X)_{i_0,i_0} \cdot f(X)_{i_0,i_2} \cdot w(X)_{i_0,j_2} \cdot v_{j_1,j_0}
$$

*Proof.* **Proof of Part 1**

$$
\begin{aligned}
& \frac{\mathrm{d}C_5(X)}{\mathrm{d}x_{i_2,j_2}} \\
= & \frac{\mathrm{d}f(X)_{i_0,i_0} \cdot v_{j_1,j_0}}{\mathrm{d}x_{i_2,j_2}} \\
= & \frac{\mathrm{d}f(X)_{i_0,i_0}}{\mathrm{d}x_{i_2,j_2}} \cdot v_{j_1,j_0} \\
= & (-f(X)_{i_0,i_0} \cdot (f(X)_{i_0,i_0} \cdot w(X)_{i_0,j_2} + \langle f(X)_{i_0}, X^\top W_{*,j_2} \rangle) \\
& + f(X)_{i_0,i_0} \cdot \langle W_{j_2,*} + W_{*,j_2}, X_{*,i_0} \rangle) \cdot v_{j_1,j_0} \\
= & -f(X)_{i_0,i_0}^2 \cdot w(X)_{i_0,j_2} \cdot v_{j_1,j_0} \\
& -f(X)_{i_0,i_0} \cdot \langle f(X)_{i_0}, X^\top W_{*,j_2} \rangle \cdot v_{j_1,j_0} \\
& +f(X)_{i_0,i_0} \cdot \langle W_{j_2,*} + W_{*,j_2}, X_{*,i_0} \rangle \cdot v_{j_1,j_0}
\end{aligned}
$$

where the first step is by definition of $C_5(X)$ (see Lemma B.16), the 2nd step is by Fact B.2, the 3rd step is by Lemma C.3, the 4th step is a rearrangement.

**Proof of Part 2**

$$
\begin{aligned}
& \frac{\mathrm{d}C_5(X)}{\mathrm{d}x_{i_2,j_2}} \\
= & \frac{\mathrm{d}f(X)_{i_0,i_0} \cdot v_{j_1,j_0}}{\mathrm{d}x_{i_2,j_2}} \\
= & \frac{\mathrm{d}f(X)_{i_0,i_0}}{\mathrm{d}x_{i_2,j_2}} \cdot v_{j_1,j_0} \\
= & -f(X)_{i_0,i_0} \cdot f(X)_{i_0,i_2} \cdot w(X)_{i_0,j_2} \cdot v_{j_1,j_0}
\end{aligned}
$$

where the first step is by definition of $C_5(X)$ (see Lemma B.16), the 2nd step is by Fact B.2, the 3rd step is by Lemma C.3. $\square$

## C.14 Derivative of $\frac{c(X)_{i_0,j_0}}{\mathrm{d}x_{i_1,j_1}}$

**Lemma C.14.** *If the following holds:*

- *Let $c(X)_{i_0,j_0}$ be defined as in Definition B.8*

*We have*

- **Part 1** *For $i_0 = i_1 = i_2 \in [n]$, $j_1, j_2 \in [d]$*

$$\frac{\mathrm{d}c(X)_{i_0,j_0}}{\mathrm{d}x_{i_1,j_1} x_{i_2,j_2}} =$$

*where we have following definitions*

$$D_1(X) := 2s(X)_{i_0,j_0} \cdot f(X)^2_{i_0,i_0} \cdot w(X)_{i_0,j_2} \cdot w(X)_{i_0,j_1}$$
$$D_2(X) := 2f(X)_{i_0,i_0} \cdot s(X)_{i_0,j_0} \cdot z(X)_{i_0,j_2} \cdot w(X)_{i_0,j_1}$$
$$+ 2f(X)_{i_0,i_0} \cdot s(X)_{i_0,j_0} \cdot z(X)_{i_0,j_1} \cdot w(X)_{i_0,j_2}$$
$$D_3(X) := -f(X)^2_{i_0,i_0} \cdot h(X)_{j_0,i_0} \cdot w(X)_{i_0,j_2} \cdot w(X)_{i_0,j_1}$$
$$D_4(X) := -f(X)_{i_0,i_0} \cdot \langle f(X)_{i_0} \circ (X^\top W_{*,j_2}), h(X)_{j_0} \rangle \cdot w(X)_{i_0,j_1}$$
$$- f(X)_{i_0,i_0} \cdot \langle f(X)_{i_0} \circ (X^\top W_{*,j_1}), h(X)_{j_0} \rangle \cdot w(X)_{i_0,j_2}$$
$$D_5(X) := -f(X)^2_{i_0,i_0} \cdot v_{j_2,j_0} \cdot w(X)_{i_0,j_1} - f(X)^2_{i_0,i_0} \cdot v_{j_1,j_0} \cdot w(X)_{i_0,j_2}$$
$$D_6(X) := -s(X)_{i_0,j_0} \cdot f(X)_{i_0,i_0} \cdot w(X)_{i_0,j_2} \cdot w(X)_{i_0,j_1}$$
$$D_7(X) := -s(X)_{i_0,j_0} \cdot f(X)_{i_0,i_0} \cdot \langle W_{*,j_2}, X_{*,i_0} \rangle \cdot w(X)_{i_0,j_1}$$
$$- s(X)_{i_0,j_0} \cdot f(X)_{i_0,i_0} \cdot \langle W_{*,j_1}, X_{*,i_0} \rangle \cdot w(X)_{i_0,j_2}$$
$$D_8(X) := -s(X)_{i_0,j_0} \cdot f(X)_{i_0,i_0} \cdot w_{j_1,j_2} - s(X)_{i_0,j_0} \cdot f(X)_{i_0,i_0} \cdot w_{j_2,j_1}$$
$$D_9(X) := s(X)_{i_0,j_0} \cdot z(X)_{i_0,j_2} \cdot z(X)_{i_0,j_1}$$
$$D_{10}(X) := -f(X)_{i_0,i_0} \cdot h(X)_{j_0,i_0} \cdot w(X)_{i_0,j_2} \cdot z(X)_{i_0,j_1}$$
$$- f(X)_{i_0,i_0} \cdot h(X)_{j_0,i_0} \cdot w(X)_{i_0,j_1} \cdot z(X)_{i_0,j_2}$$
$$D_{11}(X) := -\langle f(X)_{i_0} \circ (X^\top W_{*,j_2}), h(X)_{j_0} \rangle \cdot z(X)_{i_0,j_1}$$
$$- \langle f(X)_{i_0} \circ (X^\top W_{*,j_1}), h(X)_{j_0} \rangle \cdot z(X)_{i_0,j_2}$$
$$D_{12}(X) := -f(X)_{i_0,i_0} \cdot v_{j_2,j_0} \cdot z(X)_{i_0,j_1} - f(X)_{i_0,i_0} \cdot v_{j_1,j_0} \cdot z(X)_{i_0,j_2}$$
$$D_{13}(X) := s(X)_{i_0,j_0} \cdot z(X)_{i_0,j_1} \cdot f(X)_{i_0,i_0} \cdot z(X)_{i_0,j_2}$$
$$D_{14}(X) := -s(X)_{i_0,j_0} \cdot \langle f(X)_{i_0} \circ (X^\top W_{*,j_2}), X^\top W_{*,j_1} \rangle$$
$$D_{15}(X) := -f(X)^2_{i_0,i_0} \cdot h(X)_{j_0,i_0} \cdot w(X)_{i_0,j_2} \cdot w(X)_{i_0,j_1}$$
$$D_{16}(X) := f(X)_{i_0,i_0} \cdot w(X)_{i_0,j_2} \cdot h(X)_{j_0,i_0} \cdot w(X)_{i_0,j_1}$$
$$D_{17}(X) := f(X)_{i_0,i_0} \cdot \langle W_{*,j_2}, X_{*,i_0} \rangle \cdot h(X)_{j_0,i_0} \cdot w(X)_{i_0,j_1}$$
$$+ f(X)_{i_0,i_0} \cdot \langle W_{*,j_1}, X_{*,i_0} \rangle \cdot h(X)_{j_0,i_0} \cdot w(X)_{i_0,j_2}$$
$$D_{18}(X) := f(X)_{i_0,i_0} \cdot v_{j_2,j_0} \cdot w(X)_{i_0,j_1} + f(X)_{i_0,i_0} \cdot v_{j_1,j_0} \cdot w(X)_{i_0,j_2}$$
$$D_{19}(X) := f(X)_{i_0,i_0} \cdot h(X)_{i_0,i_0} \cdot w_{j_1,j_2} + f(X)_{i_0,i_0} \cdot h(X)_{i_0,i_0} \cdot w_{j_2,j_1}$$
$$D_{20}(X) := + \langle f(X)_{i_0} \circ (X^\top W_{*,j_2}) \circ (X^\top W_{*,j_1}), h(X)_{j_0} \rangle$$
$$D_{21}(X) := + f(X)_{i_0,i_0} \cdot \langle W_{*,j_2}, X_{*,i_0} \rangle \cdot v_{j_1,j_0} + f(X)_{i_0,i_0} \cdot \langle W_{*,j_1}, X_{*,i_0} \rangle \cdot v_{j_2,j_0}$$

- **Part 2** *For $i_0 = i_1 \neq i_2 \in [n]$, $j_1, j_2 \in [d]$*

$$\frac{\mathrm{d}c(X)_{i_0,j_0}}{\mathrm{d}x_{i_1,j_1} x_{i_2,j_2}} =$$

*Proof.* The proof is a combination of derivatives of $C_i(X)$ in this section.

Notice that the symmetricity for **Part 1** is verified by tables in this section. $\square$

# D  HESSIAN CASE 2: $i_0 \neq i_1$

In this section, we focus on the second case of Hessian. In Sections D.1, D.2, D.3, D.4 and D.5, we calculated derivative of some important terms. In Sections D.6, D.7 and D.8 we calculate derivative of $C_6$, $C_7$ and $C_8$ respectively. And in Section D.9 we calculate the derivative of $\frac{\mathrm{d}c(X)_{i_0,j_1}}{\mathrm{d}x_{i_1,j_1}}$.

## D.1  DERIVATIVE OF SCALAR FUNCTION $f(X)_{i_0,i_1}$

**Lemma D.1.** *If the following holds:*

- *Let $f(X)_{i_0}$ be defined as Definition B.6*

- *For $i_0 \neq i_2 \in [n]$, $j_1, j_2 \in [d]$*

*We have*

$$
\frac{\mathrm{d}f(X)_{i_0,i_1}}{\mathrm{d}x_{i_2,j_2}} = - f(X)_{i_0,i_1} \cdot f(X)_{i_0,i_2} \cdot w(X)_{i_0,j_2}
$$
$$
+ f(X)_{i_0,i_1} \cdot w(X)_{i_0,j_2}
$$

*Proof.*

$$
\frac{\mathrm{d}f(X)_{i_0,i_1}}{\mathrm{d}x_{i_2,j_2}} = (-(\alpha(X)_{i_0})^{-1} \cdot f(X)_{i_0} \cdot u(X)_{i_0,i_2} \cdot \langle W_{j_2,*}, X_{*,i_0} \rangle
$$
$$
+ f(X)_{i_0} \circ (e_{i_1} \cdot \langle W_{j_2,*}, X_{*,i_0} \rangle))_{i_1}
$$
$$
= - (\alpha(X)_{i_0})^{-1} \cdot f(X)_{i_0,i_1} \cdot u(X)_{i_0,i_2} \cdot \langle W_{j_2,*}, X_{*,i_0} \rangle
$$
$$
+ f(X)_{i_0,i_1} \cdot \langle W_{j_2,*}, X_{*,i_0} \rangle
$$
$$
= - f(X)_{i_0,i_1} \cdot f(X)_{i_0,i_2} \cdot w(X)_{i_0,j_2}
$$
$$
+ f(X)_{i_0,i_1} \cdot w(X)_{i_0,j_2}
$$

where the first step follows from Part 1 of Lemma B.14, the second step follows from simple algebra, the first step follows from Definition B.6. $\qquad\square$

## D.2  DERIVATIVE OF SCALAR FUNCTION $h(X)_{j_0,i_1}$

**Lemma D.2.** *If the following holds:*

- *Let $h(X)_{j_0}$ be defined as Definition B.7*

- *For $i_0 \neq i_2 \in [n]$, $j_1, j_2 \in [d]$*

*We have*

- **Part 1.** *For $i_0 \neq i_2, i_1 = i_2 \in [n]$, $j_1, j_2 \in [d]$*

$$
\frac{\mathrm{d}h(X)_{j_0,i_1}}{\mathrm{d}x_{i_2,j_2}} = v_{j_2,j_0}
$$

- **Part 2.** *For $i_0 \neq i_2, i_1 \neq i_2 \in [n]$, $j_1, j_2 \in [d]$*

$$
\frac{\mathrm{d}h(X)_{j_0,i_1}}{\mathrm{d}x_{i_2,j_2}} = 0
$$

*Proof.* **Proof of Part 1.**

$$
\frac{\mathrm{d}h(X)_{j_0,i_1}}{\mathrm{d}x_{i_2,j_2}} = (e_{i_2} \cdot v_{j_2,j_0})_{i_1}
$$
$$
= v_{j_2,j_0}
$$

where the first step follows from Lemma B.7, the second step follows from $i_1 = i_2$.

**Proof of Part 1.**

$$\frac{\mathrm{d}h(X)_{j_0,i_1}}{\mathrm{d}x_{i_2,j_2}} = (e_{i_2} \cdot v_{j_2,j_0})_{i_1}$$
$$= 0$$

where the first step follows from Lemma B.7, the second step follows from simple algebra. $\square$

## D.3 DERIVATIVE OF SCALAR FUNCTION $\langle f(X)_{i_0}, h(X)_{j_0} \rangle$

**Lemma D.3.** *If the following holds:*

- *Let $f(X)_{i_0}$ be defined as Definition B.6*
- *Let $h(X)_{j_0}$ be defined as Definition B.7*
- *For $i_0 \neq i_2 \in [n]$, $j_1, j_2 \in [d]$*

*We have*

- 
$$\frac{\mathrm{d}\langle f(X)_{i_0}, h(X)_{j_0} \rangle}{\mathrm{d}x_{i_2,j_2}} = \langle -f(X)_{i_0} \cdot f(X)_{i_0,i_2} \cdot \langle W_{j_2,*}, X_{*,i_0} \rangle$$
$$+ f(X)_{i_0} \circ (e_{i_1} \cdot \langle W_{j_2,*}, X_{*,i_0} \rangle), h(X)_{j_0} \rangle + f(X)_{i_0,i_2} \cdot v_{j_2,j_0}$$

*Proof.*

$$\frac{\mathrm{d}\langle f(X)_{i_0}, h(X)_{j_0} \rangle}{\mathrm{d}x_{i_2,j_2}} = \langle \frac{\mathrm{d}f(X)_{i_0}}{\mathrm{d}x_{i_2,j_2}}, h(X)_{j_0} \rangle + \langle f(X)_{i_0}, \frac{\mathrm{d}h(X)_{j_0}}{\mathrm{d}x_{i_2,j_2}} \rangle$$
$$= \langle -(\alpha(X)_{i_0})^{-1} \cdot f(X)_{i_0} \cdot u(X)_{i_0,i_2} \cdot \langle W_{j_2,*}, X_{*,i_0} \rangle$$
$$+ f(X)_{i_0} \circ (e_{i_1} \cdot \langle W_{j_2,*}, X_{*,i_0} \rangle), h(X)_{j_0} \rangle + \langle f(X)_{i_0}, \frac{\mathrm{d}h(X)_{j_0}}{\mathrm{d}x_{i_2,j_2}} \rangle$$
$$= \langle -f(X)_{i_0} \cdot f(X)_{i_0,i_2} \cdot \langle W_{j_2,*}, X_{*,i_0} \rangle$$
$$+ f(X)_{i_0} \circ (e_{i_1} \cdot \langle W_{j_2,*}, X_{*,i_0} \rangle), h(X)_{j_0} \rangle + \langle f(X)_{i_0}, \frac{\mathrm{d}h(X)_{j_0}}{\mathrm{d}x_{i_2,j_2}} \rangle$$
$$= \langle -f(X)_{i_0} \cdot f(X)_{i_0,i_2} \cdot \langle W_{j_2,*}, X_{*,i_0} \rangle$$
$$+ f(X)_{i_0} \circ (e_{i_1} \cdot \langle W_{j_2,*}, X_{*,i_0} \rangle), h(X)_{j_0} \rangle + \langle f(X)_{i_0}, e_{i_2} \cdot v_{j_2,j_0} \rangle$$
$$= \langle -f(X)_{i_0} \cdot f(X)_{i_0,i_2} \cdot \langle W_{j_2,*}, X_{*,i_0} \rangle$$
$$+ f(X)_{i_0} \circ (e_{i_1} \cdot \langle W_{j_2,*}, X_{*,i_0} \rangle), h(X)_{j_0} \rangle + f(X)_{i_0,i_2} \cdot v_{j_2,j_0}$$

where the first step follows from simple differential rule, the second step follows from Lemma B.14, the third step follows from simple algebra and Definition B.6, the fourth step follows from Lemma B.15, the last step follows from simple algebra. $\square$

## D.4 DERIVATIVE OF SCALAR FUNCTION $f(X)_{i_0,i_1} \cdot \langle W_{j_1,*}, X_{*,i_0} \rangle$

**Lemma D.4.** *If the following holds:*

- *Let $f(X)_{i_0}$ be defined as Definition B.6*
- *For $i_0 \neq i_2 \in [n]$, $j_1, j_2 \in [d]$*

*We have*

- 

$$\frac{\mathrm{d}f(X)_{i_0,i_1} \cdot \langle W_{j_1,*}, X_{*,i_0}\rangle}{\mathrm{d}x_{i_2,j_2}}$$
$$= (-f(X)_{i_0,i_2}f(X)_{i_0,i_1} + f(X)_{i_0,i_1}) \cdot \langle W_{j_2,*}, X_{*,i_0}\rangle \cdot \langle W_{j_1,*}, X_{*,i_0}\rangle$$

*Proof.*

$$\frac{\mathrm{d}f(X)_{i_0,i_1} \cdot \langle W_{j_1,*}, X_{*,i_0}\rangle}{\mathrm{d}x_{i_2,j_2}}$$
$$= \frac{\mathrm{d}f(X)_{i_0,i_1}}{\mathrm{d}x_{i_2,j_2}} \cdot \langle W_{j_1,*}, X_{*,i_0}\rangle + \frac{\mathrm{d}\langle W_{j_1,*}, X_{*,i_0}\rangle}{\mathrm{d}x_{i_2,j_2}} \cdot f(X)_{i_0,i_1}$$
$$= (-f(X)_{i_0,i_2}f(X)_{i_0,i_1} + f(X)_{i_0,i_1}) \cdot \langle W_{j_2,*}, X_{*,i_0}\rangle \cdot \langle W_{j_1,*}, X_{*,i_0}\rangle$$
$$\quad + \frac{\mathrm{d}\langle W_{j_1,*}, X_{*,i_0}\rangle}{\mathrm{d}x_{i_2,j_2}} \cdot f(X)_{i_0,i_1}$$
$$= (-f(X)_{i_0,i_2}f(X)_{i_0,i_1} + f(X)_{i_0,i_1}) \cdot \langle W_{j_2,*}, X_{*,i_0}\rangle \cdot \langle W_{j_1,*}, X_{*,i_0}\rangle + \mathbf{0}_d * f(X)_{i_0,i_1}$$
$$= (-f(X)_{i_0,i_2}f(X)_{i_0,i_1} + f(X)_{i_0,i_1}) \cdot \langle W_{j_2,*}, X_{*,i_0}\rangle \cdot \langle W_{j_1,*}, X_{*,i_0}\rangle$$

where the first step follows from simple differential rule, the second step follows from Lemma D.1, the third step follows from $i_0 \neq i_2$, the last step follows from simple algebra. $\qquad\square$

## D.5 DERIVATIVE OF SCALAR FUNCTION $f(X)_{i_0,i_1} \cdot h(X)_{j_0,i_1}$

**Lemma D.5.** *If the following holds:*

- *Let $f(X)_{i_0}$ be defined as Definition B.6*
- *Let $h(X)_{j_0}$ be defined as Definition B.7*

*We have*

- **Part 1** *For $i_0 \neq i_2, i_1 = i_2 \in [n]$, $j_1, j_2 \in [d]$*

$$\frac{\mathrm{d}f(X)_{i_0,i_1} \cdot h(X)_{j_0,i_1}}{\mathrm{d}x_{i_2,j_2}}$$
$$= (-f(X)_{i_0,i_2} + 1 + v_{j_2,j_0}) \cdot f(X)_{i_0,i_1} \cdot \langle W_{j_2,*}, X_{*,i_0}\rangle \cdot h(X)_{j_0,i_1}$$

- **Part 2** *For $i_0 \neq i_2, i_1 \neq i_2 \in [n]$, $j_1, j_2 \in [d]$*

$$\frac{\mathrm{d}f(X)_{i_0,i_0} \cdot h(X)_{j_0,i_0}}{\mathrm{d}x_{i_2,j_2}}$$
$$= (-f(X)_{i_0,i_2}f(X)_{i_0,i_1} + f(X)_{i_0,i_1}) \cdot \langle W_{j_2,*}, X_{*,i_0}\rangle \cdot h(X)_{j_0,i_1}$$

*Proof.* **Proof of Part 1.**

$$\frac{\mathrm{d}f(X)_{i_0,i_1} \cdot h(X)_{j_0,i_1}}{\mathrm{d}x_{i_2,j_2}} = \frac{\mathrm{d}f(X)_{i_0,i_1}}{\mathrm{d}x_{i_2,j_2}} \cdot h(X)_{j_0,i_1} + \frac{\mathrm{d}h(X)_{j_0,i_1}}{\mathrm{d}x_{i_2,j_2}} \cdot f(X)_{i_0,i_1}$$
$$= (-f(X)_{i_0,i_2}f(X)_{i_0,i_1} + f(X)_{i_0,i_1}) \cdot \langle W_{j_2,*}, X_{*,i_0}\rangle \cdot h(X)_{j_0,i_1}$$
$$\quad + \frac{\mathrm{d}h(X)_{j_0,i_1}}{\mathrm{d}x_{i_2,j_2}} \cdot f(X)_{i_0,i_1}$$
$$= (-f(X)_{i_0,i_2}f(X)_{i_0,i_1} + f(X)_{i_0,i_1}) \cdot \langle W_{j_2,*}, X_{*,i_0}\rangle \cdot h(X)_{j_0,i_1}$$
$$\quad + v_{j_2,j_0} \cdot f(X)_{i_0,i_1}$$
$$= (-f(X)_{i_0,i_2} + 1 + v_{j_2,j_0}) \cdot f(X)_{i_0,i_1} \cdot \langle W_{j_2,*}, X_{*,i_0}\rangle \cdot h(X)_{j_0,i_1}$$

where the first step follows from simple differential rule, the second step follows from Lemma D.1, the third step follows from Part 1 of Lemma D.2, the last step follows from simple algebra.

**Proof of Part 2.**

$$\frac{\mathrm{d}f(X)_{i_0,i_1} \cdot h(X)_{j_0,i_1}}{\mathrm{d}x_{i_2,j_2}} = \frac{\mathrm{d}f(X)_{i_0,i_1}}{\mathrm{d}x_{i_2,j_2}} \cdot h(X)_{j_0,i_1} + \frac{\mathrm{d}h(X)_{j_0,i_1}}{\mathrm{d}x_{i_2,j_2}} \cdot f(X)_{i_0,i_1}$$

$$= (-f(X)_{i_0,i_2} f(X)_{i_0,i_1} + f(X)_{i_0,i_1}) \cdot \langle W_{j_2,*}, X_{*,i_0} \rangle \cdot h(X)_{j_0,i_1}$$

$$+ \frac{\mathrm{d}h(X)_{j_0,i_1}}{\mathrm{d}x_{i_2,j_2}} \cdot f(X)_{i_0,i_1}$$

$$= (-f(X)_{i_0,i_2} f(X)_{i_0,i_1} + f(X)_{i_0,i_1}) \cdot \langle W_{j_2,*}, X_{*,i_0} \rangle \cdot h(X)_{j_0,i_1}$$

where the first step follows from simple differential rule, the second step follows from Lemma D.1, the third step follows from Part 2 of Lemma D.2. □

### D.6 DERIVATIVE OF $C_6(X)$

- $C_6(X) := -\langle f(X)_{i_0}, h(X)_{j_0} \rangle \cdot f(X)_{i_0,i_1} \cdot \langle W_{j_1,*}, X_{*,i_0} \rangle$
- $C_7(X) := f(X)_{i_0,i_1} \cdot h(X)_{j_0,i_1} \cdot \langle W_{j_1,*}, X_{*,i_0} \rangle$
- $C_8(X) := f(X)_{i_0,i_1} \cdot v_{j_1,j_0}$

**Lemma D.6.** *If the following holds:*

- *Let $C_6(X) \in \mathbb{R}$ be defined as in Lemma B.16*

- *For $i_0 \neq i_2 \in [n]$, $j_1, j_2 \in [d]$*

*We have*

-

$$\frac{\mathrm{d}C_6(X)}{\mathrm{d}x_{i_2,j_2}}$$

$$= \langle -f(X)_{i_0} \cdot f(X)_{i_0,i_2} \cdot \langle W_{j_2,*}, X_{*,i_0} \rangle$$

$$+ f(X)_{i_0} \circ (e_{i_1} \cdot \langle W_{j_2,*}, X_{*,i_0} \rangle), h(X)_{j_0} \rangle + f(X)_{i_0,i_2} \cdot v_{j_2,j_0} \cdot f(X)_{i_0,i_1} \cdot \langle W_{j_1,*}, X_{*,i_0} \rangle$$

$$+ (-\langle f(X)_{i_0}, h(X)_{j_0} \rangle) \cdot (-f(X)_{i_0,i_2} f(X)_{i_0,i_1} + f(X)_{i_0,i_1}) \cdot \langle W_{j_2,*}, X_{*,i_0} \rangle \cdot \langle W_{j_1,*}, X_{*,i_0} \rangle$$

*Proof.*

$$\frac{\mathrm{d}C_6(X)}{\mathrm{d}x_{i_2,j_2}}$$

$$= \frac{\mathrm{d}}{\mathrm{d}x_{i_2,j_2}} (-\langle f(X)_{i_0}, h(X)_{j_0} \rangle \cdot f(X)_{i_0,i_1} \cdot \langle W_{j_1,*}, X_{*,i_0} \rangle)$$

$$= \frac{\mathrm{d}}{\mathrm{d}x_{i_2,j_2}} (-\langle f(X)_{i_0}, h(X)_{j_0} \rangle) \cdot f(X)_{i_0,i_1} \cdot \langle W_{j_1,*}, X_{*,i_0} \rangle$$

$$+ (-\langle f(X)_{i_0}, h(X)_{j_0} \rangle) \cdot \frac{\mathrm{d}}{\mathrm{d}x_{i_2,j_2}} (f(X)_{i_0,i_1} \cdot \langle W_{j_1,*}, X_{*,i_0} \rangle)$$

$$= \frac{\mathrm{d}}{\mathrm{d}x_{i_2,j_2}} (-\langle f(X)_{i_0}, h(X)_{j_0} \rangle) \cdot f(X)_{i_0,i_1} \cdot \langle W_{j_1,*}, X_{*,i_0} \rangle$$

$$+ (-\langle f(X)_{i_0}, h(X)_{j_0} \rangle) \cdot (-f(X)_{i_0,i_2} f(X)_{i_0,i_1} + f(X)_{i_0,i_1}) \cdot \langle W_{j_2,*}, X_{*,i_0} \rangle \cdot \langle W_{j_1,*}, X_{*,i_0} \rangle$$

$$= \langle -f(X)_{i_0} \cdot f(X)_{i_0,i_2} \cdot \langle W_{j_2,*}, X_{*,i_0} \rangle$$

$$+ f(X)_{i_0} \circ (e_{i_1} \cdot \langle W_{j_2,*}, X_{*,i_0} \rangle), h(X)_{j_0} \rangle + f(X)_{i_0,i_2} \cdot v_{j_2,j_0} \cdot f(X)_{i_0,i_1} \cdot \langle W_{j_1,*}, X_{*,i_0} \rangle$$

$$+ (-\langle f(X)_{i_0}, h(X)_{j_0} \rangle) \cdot (-f(X)_{i_0,i_2} f(X)_{i_0,i_1} + f(X)_{i_0,i_1}) \cdot \langle W_{j_2,*}, X_{*,i_0} \rangle \cdot \langle W_{j_1,*}, X_{*,i_0} \rangle$$

where the first step follows from Lemma B.16, the second step follows from simple differential rule, the third step follows from Lemma D.4, last step follows from Lemma D.3. □

## D.7    DERIVATIVE OF $C_7(X)$

**Lemma D.7.** *If the following holds:*

- *Let $C_7(X) \in \mathbb{R}$ be defined as in Lemma B.16*

*We have*

- **Part 1.** *For $i_0 \neq i_2, i_1 = i_2 \in [n]$, $j_1, j_2 \in [d]$*

$$\frac{\mathrm{d}C_7(X)}{\mathrm{d}x_{i_2,j_2}}$$
$$= (-f(X)_{i_0,i_2} + 1 + v_{j_2,j_0}) \cdot f(X)_{i_0,i_1} \cdot \langle W_{j_2,*}, X_{*,i_0} \rangle \cdot h(X)_{j_0,i_1} \rangle \cdot \langle W_{j_1,*}, X_{*,i_0} \rangle$$

- **Part 2.** *For $i_0 \neq i_2, i_1 \neq i_2 \in [n]$, $j_1, j_2 \in [d]$*

$$\frac{\mathrm{d}C_7(X)}{\mathrm{d}x_{i_2,j_2}}$$
$$= (-f(X)_{i_0,i_2}f(X)_{i_0,i_1} + f(X)_{i_0,i_1}) \cdot \langle W_{j_2,*}, X_{*,i_0} \rangle \cdot h(X)_{j_0,i_1} \cdot \langle W_{j_1,*}, X_{*,i_0} \rangle$$

*Proof.* **Proof of Part 1.**

$$\frac{\mathrm{d}C_7(X)}{\mathrm{d}x_{i_2,j_2}}$$
$$= \frac{\mathrm{d}}{\mathrm{d}x_{i_2,j_2}} (f(X)_{i_0,i_1} \cdot h(X)_{j_0,i_1} \cdot \langle W_{j_1,*}, X_{*,i_0} \rangle)$$
$$= \frac{\mathrm{d}}{\mathrm{d}x_{i_2,j_2}} (f(X)_{i_0,i_1} \cdot h(X)_{j_0,i_1}) \cdot \langle W_{j_1,*}, X_{*,i_0} \rangle + f(X)_{i_0,i_1} \cdot h(X)_{j_0,i_1} \cdot \frac{\mathrm{d}}{\mathrm{d}x_{i_2,j_2}} (\langle W_{j_1,*}, X_{*,i_0} \rangle)$$
$$= (-f(X)_{i_0,i_2} + 1 + v_{j_2,j_0}) \cdot f(X)_{i_0,i_1} \cdot \langle W_{j_2,*}, X_{*,i_0} \rangle \cdot h(X)_{j_0,i_1} \rangle \cdot \langle W_{j_1,*}, X_{*,i_0} \rangle$$
$$\quad + f(X)_{i_0,i_1} \cdot h(X)_{j_0,i_1} \cdot \frac{\mathrm{d}}{\mathrm{d}x_{i_2,j_2}} (\langle W_{j_1,*}, X_{*,i_0} \rangle)$$
$$= (-f(X)_{i_0,i_2} + 1 + v_{j_2,j_0}) \cdot f(X)_{i_0,i_1} \cdot \langle W_{j_2,*}, X_{*,i_0} \rangle \cdot h(X)_{j_0,i_1} \rangle \cdot \langle W_{j_1,*}, X_{*,i_0} \rangle$$
$$\quad + f(X)_{i_0,i_1} \cdot h(X)_{j_0,i_1} \cdot \mathbf{0}_d$$
$$= (-f(X)_{i_0,i_2} + 1 + v_{j_2,j_0}) \cdot f(X)_{i_0,i_1} \cdot \langle W_{j_2,*}, X_{*,i_0} \rangle \cdot h(X)_{j_0,i_1} \rangle \cdot \langle W_{j_1,*}, X_{*,i_0} \rangle$$

where the first step follows from Lemma B.16, the second step follows from differential rule, the third step follows from Part 1 of Lemma D.3, the fourth step follows from $i_0 \neq i_2$, the last step follows from simple algebra.

**Proof of Part 2.**

$$\frac{\mathrm{d}C_7(X)}{\mathrm{d}x_{i_2,j_2}}$$
$$= \frac{\mathrm{d}}{\mathrm{d}x_{i_2,j_2}} (f(X)_{i_0,i_1} \cdot h(X)_{j_0,i_1} \cdot \langle W_{j_1,*}, X_{*,i_0} \rangle)$$
$$= \frac{\mathrm{d}}{\mathrm{d}x_{i_2,j_2}} (f(X)_{i_0,i_1} \cdot h(X)_{j_0,i_1}) \cdot \langle W_{j_1,*}, X_{*,i_0} \rangle + f(X)_{i_0,i_1} \cdot h(X)_{j_0,i_1} \cdot \frac{\mathrm{d}}{\mathrm{d}x_{i_2,j_2}} (\langle W_{j_1,*}, X_{*,i_0} \rangle)$$
$$= (-f(X)_{i_0,i_2}f(X)_{i_0,i_1} + f(X)_{i_0,i_1}) \cdot \langle W_{j_2,*}, X_{*,i_0} \rangle \cdot h(X)_{j_0,i_1} \cdot \langle W_{j_1,*}, X_{*,i_0} \rangle$$
$$\quad + f(X)_{i_0,i_1} \cdot h(X)_{j_0,i_1} \cdot \frac{\mathrm{d}}{\mathrm{d}x_{i_2,j_2}} (\langle W_{j_1,*}, X_{*,i_0} \rangle)$$
$$= (-f(X)_{i_0,i_2}f(X)_{i_0,i_1} + f(X)_{i_0,i_1}) \cdot \langle W_{j_2,*}, X_{*,i_0} \rangle \cdot h(X)_{j_0,i_1} \cdot \langle W_{j_1,*}, X_{*,i_0} \rangle$$
$$\quad + f(X)_{i_0,i_1} \cdot h(X)_{j_0,i_1} \cdot \mathbf{0}_d$$
$$= (-f(X)_{i_0,i_2}f(X)_{i_0,i_1} + f(X)_{i_0,i_1}) \cdot \langle W_{j_2,*}, X_{*,i_0} \rangle \cdot h(X)_{j_0,i_1} \cdot \langle W_{j_1,*}, X_{*,i_0} \rangle$$

where the first step follows from Lemma B.16, the second step follows from differential rule, the third step follows from Part 2 of Lemma D.3, the fourth step follows from $i_0 \neq i_2$, the last step follows from simple algebra. $\qquad\square$

## D.8 DERIVATIVE OF $C_8(X)$

**Lemma D.8.** *If the following holds:*

- *Let $C_8(X) \in \mathbb{R}$ be defined as in Lemma B.16*

- *For $i_0 \neq i_2 \in [n]$, $j_1, j_2 \in [d]$*

*We have*

- 

$$\frac{\mathrm{d}C_8(X)}{\mathrm{d}x_{i_2,j_2}}$$
$$= (-f(X)_{i_0,i_2}f(X)_{i_0,i_1} + f(X)_{i_0,i_1}) \cdot \langle W_{j_2,*}, X_{*,i_0} \rangle \cdot v_{j_1,j_0}$$

*Proof.*

$$\frac{\mathrm{d}C_8(X)}{\mathrm{d}x_{i_2,j_2}} = \frac{\mathrm{d}}{\mathrm{d}x_{i_2,j_2}} f(X)_{i_0,i_1} \cdot v_{j_1,j_0}$$
$$= (-f(X)_{i_0,i_2}f(X)_{i_0,i_1} + f(X)_{i_0,i_1}) \cdot \langle W_{j_2,*}, X_{*,i_0} \rangle \cdot v_{j_1,j_0}$$

where the first step follows from Lemma B.16, the second step follows from differential rule and Lemma D.1. $\square$

## D.9 DERIVATIVE OF $\frac{\mathrm{d}c(X)_{i_0,j_1}}{\mathrm{d}x_{i_1,j_1}}$

**Lemma D.9.** *If the following holds:*

- *Let $c(X)_{i_0,j_1} \in \mathbb{R}$ be defined as in Lemma B.16 and Definition B.8*

*We have*

- **Part 1** *For $i_0 \neq i_2, i_1 = i_2 \in [n]$, $j_1, j_2 \in [d]$*

$$\frac{\mathrm{d}c(X)}{\mathrm{d}x_{i_1,j_1}, \mathrm{d}x_{i_2,j_2}} =$$

- **Part 2** *For $i_0 \neq i_2, i_1 \neq i_2 \in [n]$, $j_1, j_2 \in [d]$*

$$\frac{\mathrm{d}c(X)}{\mathrm{d}x_{i_1,j_1}, \mathrm{d}x_{i_2,j_2}} =$$

*Proof.* **Proof of Part 1.**

$$\frac{\mathrm{d}c(X)_{i_0,j_0}}{\mathrm{d}x_{i_1,j_1}, \mathrm{d}x_{i_2,j_2}}$$
$$= \frac{\mathrm{d}C_6}{\mathrm{d}x_{i_2,j_2}} + \frac{\mathrm{d}C_7}{\mathrm{d}x_{i_2,j_2}} + \frac{\mathrm{d}C_8}{\mathrm{d}x_{i_2,j_2}}$$
$$= \langle -f(X)_{i_0} \cdot f(X)_{i_0,i_2} \cdot \langle W_{j_2,*}, X_{*,i_0} \rangle + f(X)_{i_0} \circ (e_{i_1} \cdot \langle W_{j_2,*}, X_{*,i_0} \rangle), h(X)_{j_0} \rangle$$
$$\quad + f(X)_{i_0,i_2} \cdot v_{j_2,j_0} \cdot f(X)_{i_0,i_1} \cdot \langle W_{j_1,*}, X_{*,i_0} \rangle$$
$$\quad + (-\langle f(X)_{i_0}, h(X)_{j_0} \rangle) \cdot (-f(X)_{i_0,i_2}f(X)_{i_0,i_1} + f(X)_{i_0,i_1}) \cdot \langle W_{j_2,*}, X_{*,i_0} \rangle \cdot \langle W_{j_1,*}, X_{*,i_0} \rangle$$
$$\quad + \frac{\mathrm{d}C_7}{\mathrm{d}x_{i_2,j_2}} + \frac{\mathrm{d}C_8}{\mathrm{d}x_{i_2,j_2}}$$
$$= \langle -f(X)_{i_0} \cdot f(X)_{i_0,i_2} \cdot \langle W_{j_2,*}, X_{*,i_0} \rangle + f(X)_{i_0} \circ (e_{i_1} \cdot \langle W_{j_2,*}, X_{*,i_0} \rangle), h(X)_{j_0} \rangle$$
$$\quad + f(X)_{i_0,i_2} \cdot v_{j_2,j_0} \cdot f(X)_{i_0,i_1} \cdot \langle W_{j_1,*}, X_{*,i_0} \rangle$$
$$\quad + (-\langle f(X)_{i_0}, h(X)_{j_0} \rangle) \cdot (-f(X)_{i_0,i_2}f(X)_{i_0,i_1} + f(X)_{i_0,i_1}) \cdot \langle W_{j_2,*}, X_{*,i_0} \rangle \cdot \langle W_{j_1,*}, X_{*,i_0} \rangle$$

$$+ (-f(X)_{i_0,i_2} + 1 + v_{j_2,j_0}) \cdot f(X)_{i_0,i_1} \cdot \langle W_{j_2,*}, X_{*,i_0} \rangle \cdot h(X)_{j_0,i_1} \rangle \cdot \langle W_{j_1,*}, X_{*,i_0} \rangle + \frac{\mathrm{d}C_8}{\mathrm{d}x_{i_2,j_2}}$$

$$= \langle -f(X)_{i_0} \cdot f(X)_{i_0,i_2} \cdot \langle W_{j_2,*}, X_{*,i_0} \rangle + f(X)_{i_0} \circ (e_{i_1} \cdot \langle W_{j_2,*}, X_{*,i_0} \rangle), h(X)_{j_0} \rangle$$
$$+ f(X)_{i_0,i_2} \cdot v_{j_2,j_0} \cdot f(X)_{i_0,i_1} \cdot \langle W_{j_1,*}, X_{*,i_0} \rangle$$
$$+ (-\langle f(X)_{i_0}, h(X)_{j_0} \rangle) \cdot (-f(X)_{i_0,i_2} f(X)_{i_0,i_1} + f(X)_{i_0,i_1}) \cdot \langle W_{j_2,*}, X_{*,i_0} \rangle \cdot \langle W_{j_1,*}, X_{*,i_0} \rangle$$
$$+ (-f(X)_{i_0,i_2} + 1 + v_{j_2,j_0}) \cdot f(X)_{i_0,i_1} \cdot \langle W_{j_2,*}, X_{*,i_0} \rangle \cdot h(X)_{j_0,i_1} \rangle \cdot \langle W_{j_1,*}, X_{*,i_0} \rangle$$
$$+ (-f(X)_{i_0,i_2} f(X)_{i_0,i_1} + f(X)_{i_0,i_1}) \cdot \langle W_{j_2,*}, X_{*,i_0} \rangle \cdot v_{j_1,j_0}$$

where the first step follows from Lemma B.16, the second step follows from Lemma D.6, the third step follows from Part 1 of Lemma D.7, the last step follows from Lemma D.8.

**Proof of Part 2.**

$$\frac{\mathrm{d}c(X)_{i_0,j_0}}{\mathrm{d}x_{i_1,j_1}, \mathrm{d}x_{i_2,j_2}}$$
$$= \frac{\mathrm{d}C_6}{\mathrm{d}x_{i_2,j_2}} + \frac{\mathrm{d}C_7}{\mathrm{d}x_{i_2,j_2}} + \frac{\mathrm{d}C_8}{\mathrm{d}x_{i_2,j_2}}$$
$$= \langle -f(X)_{i_0} \cdot f(X)_{i_0,i_2} \cdot \langle W_{j_2,*}, X_{*,i_0} \rangle + f(X)_{i_0} \circ (e_{i_1} \cdot \langle W_{j_2,*}, X_{*,i_0} \rangle), h(X)_{j_0} \rangle$$
$$+ f(X)_{i_0,i_2} \cdot v_{j_2,j_0} \cdot f(X)_{i_0,i_1} \cdot \langle W_{j_1,*}, X_{*,i_0} \rangle$$
$$+ (-\langle f(X)_{i_0}, h(X)_{j_0} \rangle) \cdot (-f(X)_{i_0,i_2} f(X)_{i_0,i_1} + f(X)_{i_0,i_1}) \cdot \langle W_{j_2,*}, X_{*,i_0} \rangle \cdot \langle W_{j_1,*}, X_{*,i_0} \rangle$$
$$+ \frac{\mathrm{d}C_7}{\mathrm{d}x_{i_2,j_2}} + \frac{\mathrm{d}C_8}{\mathrm{d}x_{i_2,j_2}}$$
$$= \langle -f(X)_{i_0} \cdot f(X)_{i_0,i_2} \cdot \langle W_{j_2,*}, X_{*,i_0} \rangle + f(X)_{i_0} \circ (e_{i_1} \cdot \langle W_{j_2,*}, X_{*,i_0} \rangle), h(X)_{j_0} \rangle$$
$$+ f(X)_{i_0,i_2} \cdot v_{j_2,j_0} \cdot f(X)_{i_0,i_1} \cdot \langle W_{j_1,*}, X_{*,i_0} \rangle$$
$$+ (-\langle f(X)_{i_0}, h(X)_{j_0} \rangle) \cdot (-f(X)_{i_0,i_2} f(X)_{i_0,i_1} + f(X)_{i_0,i_1}) \cdot \langle W_{j_2,*}, X_{*,i_0} \rangle \cdot \langle W_{j_1,*}, X_{*,i_0} \rangle$$
$$+ (-f(X)_{i_0,i_2} f(X)_{i_0,i_1} + f(X)_{i_0,i_1}) \cdot \langle W_{j_2,*}, X_{*,i_0} \rangle \cdot h(X)_{j_0,i_1} \cdot \langle W_{j_1,*}, X_{*,i_0} \rangle + \frac{\mathrm{d}C_8}{\mathrm{d}x_{i_2,j_2}}$$
$$= \langle -f(X)_{i_0} \cdot f(X)_{i_0,i_2} \cdot \langle W_{j_2,*}, X_{*,i_0} \rangle + f(X)_{i_0} \circ (e_{i_1} \cdot \langle W_{j_2,*}, X_{*,i_0} \rangle), h(X)_{j_0} \rangle$$
$$+ f(X)_{i_0,i_2} \cdot v_{j_2,j_0} \cdot f(X)_{i_0,i_1} \cdot \langle W_{j_1,*}, X_{*,i_0} \rangle$$
$$+ (-\langle f(X)_{i_0}, h(X)_{j_0} \rangle) \cdot (-f(X)_{i_0,i_2} f(X)_{i_0,i_1} + f(X)_{i_0,i_1}) \cdot \langle W_{j_2,*}, X_{*,i_0} \rangle \cdot \langle W_{j_1,*}, X_{*,i_0} \rangle$$
$$+ (-f(X)_{i_0,i_2} f(X)_{i_0,i_1} + f(X)_{i_0,i_1}) \cdot \langle W_{j_2,*}, X_{*,i_0} \rangle \cdot h(X)_{j_0,i_1} \cdot \langle W_{j_1,*}, X_{*,i_0} \rangle$$
$$+ (-f(X)_{i_0,i_2} f(X)_{i_0,i_1} + f(X)_{i_0,i_1}) \cdot \langle W_{j_2,*}, X_{*,i_0} \rangle \cdot v_{j_1,j_0}$$

where the first step follows from Lemma B.16, the second step follows from Lemma D.6, the third step follows from Part 2 of Lemma D.7, the last step follows from Lemma D.8. $\square$

## E  HESSIAN REFORMULATION:

In this section, we provide a reformulation of Hessian formula, which simplifies our calculation and analysis. In Section E.1 we show the way we split the Hessian. In Section E.2 we show the decomposition when $i_0 = i_1 = i_2$.

### E.1  HESSIAN SPLIT

**Definition E.1** (Hessian of functions of matrix). *We define the Hessian of $c(X)_{i_0,j_0}$ by considering its Hessian with respect to $x = \mathrm{vec}(X)$. This means that, $\nabla^2 c(X)_{i_0,j_0}$ is a $nd \times nd$ matrix with its $i_1 \cdot j_1, i_2 \cdot j_2$-th entry being*

$$\frac{\mathrm{d}c(X)_{i_0,j_0}}{\mathrm{d}x_{i_1,j_2} x_{i_2,j_2}}$$

**Definition E.2** (Hessian split). *We split the hessian of $c(X)_{i_0,j_0}$ into following cases*

- $i_0 = i_1 = i_2 : H_1^{(i_1,i_2)}$

- $i_0 = i_1$, $i_0 \neq i_2$ : $H_2^{(i_1,i_2)}$

- $i_0 \neq i_1$, $i_0 = i_2$ : $H_3^{(i_1,i_2)}$

- $i_0 \neq i_1$, $i_0 \neq i_2$ : $H_4^{(i_1,i_2)}$

*In above, $H_i^{(i_1,i_2)}$ is a $d \times d$ matrix with its $j_1, j_2$-th entry being*

$$\frac{\mathrm{d}c(X)_{i_0,j_0}}{\mathrm{d}x_{i_1,j_2} x_{i_2,j_2}}$$

Utilizing above definitions, we split the Hessian to a $n \times n$ partition with its $i_1, i_2$-th component being $H_i(i_1, i_2)$.

**Definition E.3.** *We define $\nabla^2 c(X)_{i_0,j_0}$ to be as following*

$$\nabla^2 c(X)_{i_0,j_0} = \begin{bmatrix} H_4^{(1,1)} & H_4^{(1,2)} & H_4^{(1,3)} & \cdots & H_3^{(1,i_0)} & \cdots & H_4^{(1,n)} \\ H_4^{(2,1)} & H_4^{(2,2)} & H_4^{(2,3)} & \cdots & H_3^{(2,i_0)} & \cdots & H_4^{(2,n)} \\ H_4^{(3,1)} & H_4^{(3,2)} & H_4^{(3,3)} & \cdots & H_3^{(3,i_0)} & \cdots & H_4^{(3,n)} \\ \vdots & \vdots & \vdots & \ddots & \vdots & \ddots & \vdots \\ H_2^{(i_0,1)} & H_2^{(i_0,2)} & H_2^{(i_0,3)} & \cdots & H_1^{(i_0,i_0)} & \cdots & H_2^{(i_0,n)} \\ \vdots & \vdots & \vdots & \ddots & \vdots & \ddots & \vdots \\ H_4^{(n,1)} & H_4^{(n,2)} & H_4^{(n,3)} & \cdots & H_3^{(n,i_0)} & \cdots & H_4^{(n,n)} \end{bmatrix}$$

## E.2 DECOMPOSITION HESSIAN : $i_0 = i_1 = i_2$

**Lemma E.4.** *Under following conditions*

- *Let $D_i(X)$ be defined as Lemma C.14*

- *Let $z(X)_{i_0} := WX \cdot f(X)_{i_0}$*

- *Let $w(X)_{i_0,*} := WX_{*,i_0}$*

*we have*

$$D_1(X) = e_{j_1}^\top \cdot w(X)_{i_0,*} \cdot 2s(X)_{i_0,j_0} \cdot f(X)_{i_0,i_0}^2 \cdot w(X)_{i_0,*}^\top \cdot e_{j_2}$$

$$D_2(X) = e_{j_1}^\top \cdot (w(X)_{i_0,*} \cdot 2f(X)_{i_0,i_0} \cdot s(X)_{i_0,j_0} \cdot z(X)_{i_0}^\top$$
$$+ z(X)_{i_0} \cdot 2f(X)_{i_0,i_0} \cdot s(X)_{i_0,j_0} \cdot w(X)_{i_0,*}^\top) \cdot e_{j_2}$$

$$D_3(X) = - e_{j_1}^\top \cdot w(X)_{i_0,*} \cdot f(X)_{i_0,i_0}^2 \cdot h(X)_{j_0,i_0} \cdot w(X)_{i_0,*}^\top \cdot e_{j_2}$$

$$D_4(X) = - e_{j_1}^\top \cdot W^\top \cdot f(X)_{i_0,i_0} \cdot X \cdot \mathrm{diag}(f(X)_{i_0}) \cdot h(X)_{j_0} \cdot w(X)_{i_0,*}^\top \cdot e_{j_2}$$
$$- e_{j_1}^\top \cdot w(X)_{i_0,*} \cdot f(X)_{i_0,i_0} \cdot h(X)_{j_0}^\top \cdot \mathrm{diag}(f(X)_{i_0}) \cdot X^\top \cdot W \cdot e_{j_2}$$

$$D_5(X) = - e_{j_1}^\top \cdot (w(X)_{i_0,*} \cdot f(X)_{i_0,i_0}^2 \cdot V_{*,j_0}^\top + V_{*,j_0} \cdot f(X)_{i_0,i_0}^2 \cdot w(X)_{i_0,*}^\top) \cdot e_{j_2}$$

$$D_6(X) = - e_{j_1}^\top \cdot w(X)_{i_0,*} \cdot s(X)_{i_0,j_0} \cdot f(X)_{i_0,i_0} \cdot w(X)_{i_0,*}^\top \cdot e_{j_2}$$

$$D_7(X) = - e_{j_1}^\top \cdot w(X)_{i_0,*} \cdot s(X)_{i_0,j_0} \cdot f(X)_{i_0,i_0} \cdot X_{*,i_0}^\top \cdot W \cdot e_{j_2}$$
$$- e_{j_1}^\top \cdot W^\top \cdot X_{*,i_0} \cdot s(X)_{i_0,j_0} \cdot f(X)_{i_0,i_0} \cdot w(X)_{i_0,*}^\top \cdot e_{j_2}$$

$$D_8(X) = e_{j_1}^\top \cdot s(X)_{i_0,j_0} \cdot f(X)_{i_0,i_0} \cdot (W^\top - W) \cdot e_{j_2}$$

$$D_9(X) = e_{j_1}^\top \cdot z(X)_{i_0} \cdot s(X)_{i_0,j_0} \cdot z(X)_{i_0}^\top \cdot e_{j_2}$$

$$D_{10}(X) = - e_{j_1}^\top \cdot (z(X)_{i_0} \cdot f(X)_{i_0,i_0} \cdot h(X)_{j_0,i_0} \cdot w(X)_{i_0,*}^\top$$
$$+ w(X)_{i_0,*} \cdot f(X)_{i_0,i_0} \cdot h(X)_{j_0,i_0} \cdot z(X)_{i_0}^\top) \cdot e_{j_2}$$

$$D_{11}(X) = - e_{j_1}^\top \cdot (z(X)_{i_0} \cdot (h(X)_{j_0}^\top \cdot \mathrm{diag}(f(X)_{i_0}) \cdot X^\top \cdot W$$

$$+ W^\top \cdot X \cdot \operatorname{diag}(f(X)_{i_0}) \cdot h(X)_{j_0} \cdot z(X)_{i_0}^\top \cdot e_{j_2}$$

$$D_{12}(X) = - e_{j_1}^\top \cdot (z(X)_{i_0} \cdot f(X)_{i_0,i_0} \cdot V_{*,j_0}^\top + V_{*,j_0} \cdot f(X)_{i_0,i_0} \cdot z(X)_{i_0}^\top) \cdot e_{j_2}$$

$$D_{13}(X) = e_{j_1}^\top \cdot z(X)_{i_0} \cdot s(X)_{i_0,j_0} \cdot f(X)_{i_0,i_0} \cdot z(X)_{i_0}^\top \cdot e_{j_2}$$

$$D_{14}(X) = - e_{j_1}^\top \cdot W^\top \cdot X \cdot s(X)_{i_0,j_0} \cdot \operatorname{diag}(f(X)_{i_0}) \cdot X^\top \cdot W \cdot e_{j_2}$$

$$D_{15}(X) = - e_{j_1}^\top \cdot w(X)_{i_0,*} \cdot f(X)_{i_0,i_0}^2 \cdot h(X)_{j_0,i_0} \cdot \cdot w(X)_{i_0,*}^\top \cdot e_{j_2}$$

$$D_{16}(X) = e_{j_1}^\top \cdot w(X)_{i_0,*} \cdot f(X)_{i_0,i_0} \cdot h(X)_{j_0,i_0} \cdot \cdot w(X)_{i_0,*}^\top \cdot e_{j_2}$$

$$D_{17}(X) = e_{j_1}^\top \cdot (w(X)_{i_0,*} \cdot f(X)_{i_0,i_0} \cdot X_{*,i_0}^\top \cdot h(X)_{j_0,i_0} \cdot W$$
$$+ W^\top \cdot X_{*,i_0} \cdot f(X)_{i_0,i_0} \cdot \cdot h(X)_{j_0,i_0} \cdot w(X)_{i_0}) \cdot e_{j_2}$$

$$D_{18}(X) = e_{j_1}^\top \cdot (w(X)_{i_0,*} f(X)_{i_0,i_0} \cdot V_{j_2,*}^\top + V_{j_1,*}^\top \cdot f(X)_{i_0,i_0} \cdot w(X)_{i_0,*}^\top) \cdot e_{j_2}$$

$$D_{19}(X) = e_{j_1}^\top \cdot f(X)_{i_0,i_0} \cdot h(X)_{i_0,i_0} \cdot (W + W^\top) \cdot e_{j_2}$$

$$D_{20}(X) := e_{j_1}^\top \cdot W^\top \cdot X \cdot \operatorname{diag}(f(X)_{i_0}) \cdot \operatorname{diag}(h(X)_{j_0}) \cdot X^\top \cdot W \cdot e_{j_2}$$

$$D_{21}(X) := e_{j_1}^\top (\cdot W^\top \cdot X_{*,i_0} \cdot f(X)_{i_0,i_0} \cdot V_{*,j_0}^\top + V_{*,j_0} \cdot f(X)_{i_0,i_0} \cdot X_{*,i_0}^\top \cdot W) \cdot e_{j_2}$$

*Proof.* This lemma is followed by linear algebra calculations. □

Based on above auxiliary lemma, we have following definition.

**Definition E.5.** *Under following conditions*

- *Let $z(X)_{i_0} := WX \cdot f(X)_{i_0}$*

- *Let $w(X)_{i_0,*} := WX_{*,i_0}$*

*We define the* **Case 1** *component of Hessian $c(X)_{i_0,j_0}$ to be*

$$H_1^{(i_0,i_0)}(X) := B(X)$$

*where we have*

$$B(X) := \sum_{i=1}^{21} B_i(X)$$

$$B_1(X) := w(X)_{i_0,*} \cdot 2s(X)_{i_0,j_0} \cdot f(X)_{i_0,i_0}^2 \cdot w(X)_{i_0,*}^\top$$

$$B_2(X) := w(X)_{i_0,*} \cdot 2f(X)_{i_0,i_0} \cdot s(X)_{i_0,j_0} \cdot z(X)_{i_0}^\top$$
$$+ z(X)_{i_0} \cdot 2f(X)_{i_0,i_0} \cdot s(X)_{i_0,j_0} \cdot w(X)_{i_0,*}^\top$$

$$B_3(X) := - w(X)_{i_0,*} \cdot f(X)_{i_0,i_0}^2 \cdot h(X)_{j_0,i_0} \cdot w(X)_{i_0,*}^\top$$

$$B_4(X) := - W^\top \cdot f(X)_{i_0,i_0} \cdot X \cdot \operatorname{diag}(f(X)_{i_0}) \cdot h(X)_{j_0} \cdot w(X)_{i_0,*}^\top$$
$$- w(X)_{i_0,*} \cdot f(X)_{i_0,i_0} \cdot h(X)_{j_0}^\top \cdot \operatorname{diag}(f(X)_{i_0}) \cdot X^\top \cdot W$$

$$B_5(X) := - w(X)_{i_0,*} \cdot f(X)_{i_0,i_0}^2 \cdot V_{*,j_0}^\top - V_{*,j_0} \cdot f(X)_{i_0,i_0}^2 \cdot w(X)_{i_0,*}^\top$$

$$B_6(X) := - w(X)_{i_0,*} \cdot s(X)_{i_0,j_0} \cdot f(X)_{i_0,i_0} \cdot w(X)_{i_0,*}^\top$$

$$B_7(X) := - w(X)_{i_0,*} \cdot s(X)_{i_0,j_0} \cdot f(X)_{i_0,i_0} \cdot X_{*,i_0}^\top \cdot W$$
$$- W^\top \cdot X_{*,i_0} \cdot s(X)_{i_0,j_0} \cdot f(X)_{i_0,i_0} \cdot w(X)_{i_0,*}^\top$$

$$B_8(X) := s(X)_{i_0,j_0} \cdot f(X)_{i_0,i_0} \cdot (W^\top - W)$$

$$B_9(X) := z(X)_{i_0} \cdot s(X)_{i_0,j_0} \cdot z(X)_{i_0}^\top$$

$$B_{10}(X) := - z(X)_{i_0} \cdot f(X)_{i_0,i_0} \cdot h(X)_{j_0,i_0} \cdot w(X)_{i_0,*}^\top$$
$$- w(X)_{i_0,*} \cdot f(X)_{i_0,i_0} \cdot h(X)_{j_0,i_0} \cdot z(X)_{i_0}^\top$$

$$B_{11}(X) := - z(X)_{i_0} \cdot (h(X)_{j_0}^\top \cdot \operatorname{diag}(f(X)_{i_0}) \cdot X^\top \cdot W$$

$$- W^\top \cdot X \cdot \mathrm{diag}(f(X)_{i_0}) \cdot h(X)_{j_0} \cdot z(X)_{i_0}^\top$$

$$B_{12}(X) := - z(X)_{i_0} \cdot f(X)_{i_0,i_0} \cdot V_{*,j_0}^\top + V_{*,j_0} \cdot f(X)_{i_0,i_0} \cdot z(X)_{i_0}^\top$$

$$B_{13}(X) := z(X)_{i_0} \cdot s(X)_{i_0,j_0} \cdot f(X)_{i_0,i_0} \cdot z(X)_{i_0}^\top$$

$$B_{14}(X) := - W^\top \cdot X \cdot s(X)_{i_0,j_0} \cdot \mathrm{diag}(f(X)_{i_0}) \cdot X^\top \cdot W$$

$$B_{15}(X) := - w(X)_{i_0,*} \cdot f(X)_{i_0,i_0}^2 \cdot h(X)_{j_0,i_0} \cdot w(X)_{i_0,*}^\top$$

$$B_{16}(X) := w(X)_{i_0,*} \cdot f(X)_{i_0,i_0} \cdot h(X)_{j_0,i_0} \cdot w(X)_{i_0,*}^\top$$

$$B_{17}(X) := w(X)_{i_0,*} \cdot f(X)_{i_0,i_0} \cdot X_{*,i_0}^\top \cdot h(X)_{j_0,i_0} \cdot W$$
$$+ W^\top \cdot X_{*,i_0} \cdot f(X)_{i_0,i_0} \cdot h(X)_{j_0,i_0} \cdot w(X)_{i_0}$$

$$B_{18}(X) := w(X)_{i_0,*} \cdot f(X)_{i_0,i_0} \cdot V_{j_2,*}^\top + V_{j_1,*}^\top \cdot f(X)_{i_0,i_0} \cdot w(X)_{i_0,*}^\top$$

$$B_{19}(X) := f(X)_{i_0,i_0} \cdot h(X)_{i_0,i_0} \cdot (W + W^\top)$$

$$B_{20}(X) := W^\top \cdot X \cdot \mathrm{diag}(f(X)_{i_0}) \cdot \mathrm{diag}(h(X)_{j_0}) \cdot X^\top$$

$$B_{21}(X) := W^\top \cdot X_{*,i_0} \cdot f(X)_{i_0,i_0} \cdot V_{*,j_0}^\top + V_{*,j_0} \cdot f(X)_{i_0,i_0} \cdot X_{*,i_0}^\top \cdot W$$

Notice that, the Hessian for other cases equals to summation of selected terms in **Case 1**. We do not provide the explicit here since they are not essential for following analysis.

## F  HESSIAN OF LOSS FUNCTION

In this section, we provide the Hessian of our loss function.

**Lemma F.1** (A single entry). *Under following conditions*

- *Let $L(X)$ be defined as Definition B.9*

*we have*

$$\frac{\mathrm{d}L(X)}{\mathrm{d}x_{i_1,j_1} x_{i_2,j_2}} = \sum_{i_0=1}^n \sum_{j_0=1}^d \frac{\mathrm{d}c(X)_{i_0,j_0}}{\mathrm{d}x_{i_1,j_1}} \cdot \frac{\mathrm{d}c(X)_{i_0,j_0}}{\mathrm{d}x_{i_1,j_2}} + c(X)_{i_0,j_0} \cdot \frac{\mathrm{d}c(X)_{i_0,j_0}}{\mathrm{d}x_{i_1,j_1} x_{i_2,j_2}}$$

*Proof.* **Proof of Part 1:** $i_1 = i_2$

$$\frac{\mathrm{d}L(X)}{\mathrm{d}x_{i_1,j_1} x_{i_2,j_2}} = \frac{\mathrm{d}}{\mathrm{d}x_{i_2,j_2}} (\sum_{i_0=1}^n \sum_{j_0=1}^d c(X)_{i_0,j_0} \cdot \frac{\mathrm{d}c(X)_{i_0,j_0}}{\mathrm{d}x_{i_1,j_1}})$$

$$= \sum_{i_0=1}^n \sum_{j_0=1}^d \frac{\mathrm{d}c(X)_{i_0,j_0}}{\mathrm{d}x_{i_1,j_1}} \cdot \frac{\mathrm{d}c(X)_{i_0,j_0}}{\mathrm{d}x_{i_1,j_2}} + c(X)_{i_0,j_0} \cdot \frac{\mathrm{d}c(X)_{i_0,j_0}}{\mathrm{d}x_{i_1,j_1} x_{i_2,j_2}}$$

where the first step is given by chain rule, and the 2nd step are given by product rule. □

**Lemma F.2** (Matrix Representation of Hessian). *Under following conditions*

- *Let $c(X)_{i_0,j_0}$ be defined as Definition B.8*
- *Let $L(X)$ be defined as Definition B.9*

*we have*

$$\nabla^2 L(X) = \sum_{i_0=1}^n \sum_{j_0=1}^d \nabla c(X)_{i_0,j_0} \cdot \nabla c(X)_{i_0,j_0}^\top + c(X)_{i_0,j_0} \cdot \nabla^2 c(X)_{i_0,j_0}$$

*Proof.* This is directly given by the single-entry representation in Lemma F.1. □

## G  BOUNDS FOR BASIC FUNCTIONS

In this section, we prove the upper bound for each function, with following assumption about the domain of parameters. In Section G.1 we bound the basic terms. In Section G.2 we bound the gradient of $f(X)_{i_0}$. In Section G.3 we bound the gradient of $c(X)_{i_0,j_0}$

**Assumption G.1** (Bounded parameters). *Let $W, V, X, B$ be defined as in Section B.2,*

- *Let $R$ be some fixed constant satisfies $R > 1$*
- *We have $\|W\| \leq R$, $\|V\| \leq R$, $\|X\| \leq R$ where $\|\cdot\|$ is the matrix spectral norm*
- *We have $b_{i_0,j_0} \leq R^2$*

### G.1  BOUNDS FOR BASIC FUNCTIONS

**Lemma G.2.** *Under Assumption G.1, for all $i_0 \in [n], j_0 \in [d]$, we have following bounds:*

- *Part 1*
$$\|f(X)_{i_0}\|_2 \leq 1$$

- *Part 2*
$$\|h(X)_{i_0}\|_2 \leq R^2$$

- *Part 3*
$$|c(X)_{j_0}| \leq 2R^2$$

- *Part 4*
$$\|x^\top W_{*,j_0}\|_2 \leq R^2$$

- *Part 5*
$$|w(X)_{i_0,j_0}| \leq R^2$$

- *Part 6*
$$|z(X)_{i_0,j_0}| \leq R^2$$

*Proof.* **Proof of Part 1**

The proof is similar to Deng et al. (2023d), and hence is omitted here.

**Proof of Part 2**
$$\|h(X)_{j_0}\|_2 = \|X^\top V_{*,j_0}\|_2$$
$$\leq \|V\| \cdot \|X\|$$
$$\leq R^2$$

where the first step is by Definition B.7, the 2nd step is by basic algebra, the 3rd follows by Assumption G.1.

**Proof of Part 3**
$$|c(X)_{j_0}| = |\langle f(X)_{i_0}, h(X)_{j_0}\rangle - b_{i_0,j_0}|$$
$$\leq |\langle f(X)_{i_0}, h(X)_{j_0}\rangle| + |b_{i_0,j_0}|$$
$$\leq \|f(X)_{i_0}\|_2 \cdot \|h(X)_{j_0}\|_2 + |b_{i_0,j_0}|$$
$$\leq 2R^2$$

where the first step is by Definition B.8, the 2nd step uses triangle inequality, the 3rd step uses Cauchy-Schwartz inequality, the 4th step is by Assumption G.1 and **Part 2**.

**Proof of Part 4**

$$\|x^\top W_{*,j_0}\|_2 \leq \|x\| \cdot \|W\|$$
$$\leq R^2$$

where the first step is by basic algebra, the second is by Assumption G.1.

**Proof of Part 5**

$$|w(X)_{i_0,j_0}| = |\langle W_{j_0,*}, X_{*,i_0}|$$
$$\leq \|W_{j_0,*}\|_2 \cdot \|X_{*,i_0}\|_2$$
$$\leq R^2$$

where the first step is by the definition of $w(X)_{i_0,j_0}$, the 2nd step is Cauchy-Schwartz inequality, the 3rd step is by Assumption G.1.

**Proof of Part 6**

$$|z(X)_{i_0,j_0}| = |\langle f(X)_{i_0}, X^\top W_{*,j_0}\rangle|$$
$$\leq \|f(X)_{i_0}\|_2 \cdot \|X\| \cdot \|W_{*,j_0}\|$$
$$\leq R^2$$

where the first step is by the definition of $z(X)_{i_0,j_0}$, the 2nd step is Cauchy-Schwartz inequality, the 3rd step is by Assumption G.1. □

## G.2 BOUNDS FOR GRADIENT OF $f(X)_{i_0}$

**Lemma G.3.** *Under Assumption G.1, for all $i_0, i_1 \in [n], j_1 \in [d]$, we have:*

$$\|\frac{\mathrm{d}f(X)_{i_0}}{\mathrm{d}x_{i_1,j_1}}\|_2 \leq 4R^2$$

*Proof.*

$$\frac{\mathrm{d}f(X)_{i_0}}{\mathrm{d}x_{i_1,j_1}} = \| -f(X)_{i_0} \cdot (f(X)_{i_0,i_0} \cdot \langle W_{j_1,*}, X_{*,i_0}\rangle + \langle f(X)_{i_0}, X^\top W_{*,j_1}\rangle)$$
$$+ f(X)_{i_0} \circ (e_{i_0} \cdot \langle W_{j_1,*}, X_{*,i_0}\rangle + X^\top W_{*,j_1})\|_2$$
$$\leq \|f(X)_{i_0}\|_2^2 \cdot |\langle W_{j_1,*}, X_{*,i_0}\rangle| + \|f(X)_{i_0}\|_2^2 \cdot \|X^\top W_{*,j_1}\|$$
$$+ \|f(X)_{i_0}\|_2 \cdot |\langle W_{j_1,*}, X_{*,i_0}\rangle| + \|f(X)_{i_0}\|_2 \cdot \|X^\top W_{*,j_1})\|_2$$
$$\leq 4R^2$$

where the 1st step is by Lemma B.14, the 2nd step is by Fact B.1, the 3rd step is by Lemma G.2. □

## G.3 BOUNDS FOR GRADIENT OF $c(X)_{i_0,j_0}$

**Lemma G.4.** *Under Assumption G.1, for all $i_0, i_1 \in [n], j_0, j_1 \in [d]$, we have:*

$$|\frac{\mathrm{d}c(X)_{i_0,j_0}}{\mathrm{d}x_{i_1,j_1}}| \leq 5R^4$$

*Proof.*

$$|\frac{\mathrm{d}c(X)_{i_0,j_0}}{\mathrm{d}x_{i_1,j_1}}| = |C_1(X) + C_2(X) + C_3(X) + C_4(X) + C_5(X)|$$
$$\leq |C_1(X)| + |C_2(X)| + |C_3(X)| + |C_4(X)| + |C_5(X)|$$
$$\leq \|f(X)_{i_0}\|_2^2 \cdot \|h(X)_{j_0}\|_2 \cdot |w(X)_{i_0,j_0}| + \|f(X)_{i_0}\|_2 \cdot \|h(X)_{j_0}\|_2 \cdot |z(X)_{i_0,j_1}|$$
$$+ \|f(X)_{i_0}\|_2 \cdot \|h(X)_{j_0}\|_2 \cdot |w(X)_{i_0,j_0}|$$
$$+ \|f(X)_{i_0}\|_2 \cdot \|X\| \cdot \|W_{*,j_1}\|_2 \cdot \|h(X)_{j_0}\|_2 + \|f(X)_{i_0}\|_2 \cdot \|V\|$$

$$\leq R^4 + R^4 + R^4 + R^4 + R^2 \leq 5R^4$$

where the first step is by Lemma B.16, the 2nd step is by triangle inequality, the 3rd step is by Fact B.1, the 4th step is by Lemma G.2, the 5th step holds by $R > 1$.

□

## H Lipschitz of Hessian

In Section H.1 we provide tools and facts. In Sections H.2, H.3, H.4, H.6, H.6, H.7 and H.8 we provide proof of lipschitz property of several important terms. And finally in Section H.9 we provide proof for Lipschitz property of Hessian of $L(X)$.

### H.1 Facts and Tools

In this section, we introduce 2 tools for effectively calculate the Lipschitz for Hessian.

**Fact H.1** (Mean value theorem for vector function, Fact 34 in Deng et al. (2023d)). *Under following conditions,*

- *Let $x, y \in C \subset \mathbb{R}^n$ where $C$ is an open convex domain*

- *Let $g(x) : C \to \mathbb{R}^n$ be a differentiable vector function on $C$*

- *Let $\|g'(a)\| \leq M$ for all $a \in C$, where $g'(a)$ denotes a matrix which its $(i, j)$-th term is $\frac{\mathrm{d}g(a)_j}{\mathrm{d}a_i}$*

*then we have*

$$\|g(y) - g(x)\|_2 \leq M\|y - x\|_2$$

**Fact H.2** (Lipschitz for product of functions). *Under following conditions*

- *Let $\{f_i(x)\}_{i=1}^n$ be a sequence of function with same domain and range*

- *For each $i \in [n]$ we have*

  - *$f_i(x)$ is bounded: $\forall x, \|f_i(x)\| \leq M_i$ with $M_i \geq 1$*
  - *$f_i(x)$ is Lipschitz continuous: $\forall x, y, \|f_i(x) - f_i(y)\| \leq L_i\|x - y\|$*

*Then we have*

$$\|\prod_{i=1}^n f_i(x) - \prod_{i=1}^n f_i(y)\| \leq 2^{n-1} \cdot \max_{i \in [n]}\{L_i\} \cdot (\prod_{i=1}^n M_i) \cdot \|x - y\|$$

*Proof.* We prove it by mathematical induction. The case that $i = 1$ obviously.

Now assume the case holds for $i = k$. Consider $i = k + 1$, we have.

$$\|\prod_{i=1}^{k+1} f_i(x) - \prod_{i=1}^{k+1} f_i(y)\|$$

$$\leq \|\prod_{i=1}^{k+1} f_i(x) - f_{k+1}(x) \cdot \prod_{i=1}^{k} f_i(y)\| + \|f_{k+1}(x) \cdot \prod_{i=1}^{k} f_i(y) - \prod_{i=1}^{k+1} f_i(y)\|$$

$$\leq \|f_{k+1}(x)\| \cdot \|\prod_{i=1}^{k} f_i(x) - \prod_{i=1}^{k} f_i(y)\| + \|f_{k+1}(x) - f_{k+1}(y)\| \cdot \|\prod_{i=1}^{k} f_i(y) - \prod_{i=1}^{k} f_i(y)\|$$

$$\leq M_{k+1} \cdot \|\prod_{i=1}^{k} f_i(x) - \prod_{i=1}^{k} f_i(y)\| + (\prod_{i=1}^{k} M_i) \cdot \|f_{k+1}(x) - f_{k+1}(y)\|$$

$$\leq 2^{k-1}(\prod_{i=1}^{k+1} M_i) \cdot \max_{i \in [k]}\{L_i\}\|x-y\| + (\prod_{i=1}^{k} M_i) \cdot \|f_{k+1}(x) - f_{k+1}(y)\|$$

$$\leq 2^{k-1}(\prod_{i=1}^{k+1} M_i) \cdot \max_{i \in [k]}\{L_i\}\|x-y\| + (\prod_{i=1}^{k} M_i) \cdot L_{k+1}\|x-y\|$$

$$\leq 2^{k-1}(\prod_{i=1}^{k+1} M_i) \cdot \max_{i \in [k]}\{L_i\}\|x-y\| + (\prod_{i=1}^{k+1} M_i) \cdot L_{k+1}\|x-y\|$$

$$\leq 2^{k}(\prod_{i=1}^{k+1} M_i) \cdot \max_{i \in [k+1]}\{L_i\}\|x-y\|$$

where the first step is by triangle inequality, the 2nd step is by property of norm, the 3rd step is by upper bound of functions, the 4th step is by induction hypothesis, the 5th step is by Lipschitz of $f_{k+1}(x)$, the 6th step is by $M_{k+1} \geq 1$, the 7th step is a rearrangement.

Since the claim holds for $i = k + 1$, we prove the desired result. $\qquad\square$

## H.2  LIPSCHITZ FOR $f(X)_{i_0}$

**Lemma H.3.** *Under following conditions*

- *Assumption G.1 holds*

- *Let $f(X)_{i_0}$ be defined as Definition B.6*

*For $X, Y \in \mathbb{R}^{d \times n}$, we have*

$$\|f(X)_{i_0} - f(Y)_{i_0}\|_2 \leq 4R^2\|X-Y\|$$

*Proof.* This lemma is directly given by Mean Value Theorem (Lemma H.1) and upper bound for gradient of $f(X)_{i_0}$ (Lemma G.3). $\qquad\square$

## H.3  LIPSCHITZ FOR $c(X)_{i_0,j_0}$

**Lemma H.4.** *Under following conditions*

- *Assumption G.1 holds*

- *Let $c(X)_{i_0,j_0}$ be defined as Definition B.8*

*For $X, Y \in \mathbb{R}^{d \times n}$, we have*

$$\|c(X)_{i_0,j_0} - c(Y)_{i_0,j_0}\|_2 \leq 5R^4\|X-Y\|$$

*Note that here we abuse notation, $\|X-Y\|$ denotes $\|\operatorname{vec}(X) - \operatorname{vec}(Y)\|_2$.*

*Proof.* This lemma is directly given by Mean Value Theorem (Lemma H.1) and upper bound for gradient of $c(X)_{i_0,j_0}$ (Lemma G.4). $\qquad\square$

## H.4  LIPSCHITZ FOR $h(X)_{j_0}$

**Lemma H.5.** *Under following conditions*

- *Assumption G.1 holds*

- *Let $h(X)_{j_0}$ be defined as Definition B.7*

*For $X, Y \in \mathbb{R}^{d \times n}$, we have*

$$\|h(X)_{j_0} - h(Y)_{j_0}\|_2 \leq R\|X-Y\|$$

*Proof.*

$$\|h(X)_{j_0} - h(Y)_{j_0}\| = \|V_{*,j_0}\|_2 \cdot \|X - Y\|$$
$$\leq R \cdot \|X - Y\|$$

where the first step is from the definition of $h(X)_{j_0}$ (see Definition B.7), the 2nd step is by Assumption G.1. □

### H.5   LIPSCHITZ FOR $w(X)_{i_0,j_0}$

**Lemma H.6.** *Under following conditions*

- *Assumption G.1 holds*

*For $X, Y \in \mathbb{R}^{d \times n}$, we have*

$$|w(X)_{i_0,j_0} - w(Y)_{i_0,j_0}| \leq R\|X - Y\|$$

*Proof.*

$$|w(X)_{i_0,j_0} - w(Y)_{i_0,j_0}| = |\langle W_{j_0,*}, X_{*,i_0} - Y_{*,i_0}\rangle|$$
$$\leq \|W_{j_0,*}\|_2 \cdot \|X - Y\|$$
$$\leq R \cdot \|X - Y\|$$

where the first step is from the definition of $w(X)_{i_0,j_0}$, the 2nd step is by Fact B.1, the 3rd step holds since Assumption G.1. □

### H.6   LIPSCHITZ FOR $z(X)_{i_0,j_0}$

**Lemma H.7.** *Under following conditions*

- *Assumption G.1 holds*

*For $X, Y \in \mathbb{R}^{d \times n}$, we have*

$$|z(X)_{i_0,j_0} - z(Y)_{i_0,j_0}| \leq 5R^4 \cdot \|X - Y\|$$

*Proof.*

$$|z(X)_{i_0,j_0} - z(Y)_{i_0,j_0}| = |\langle f(X)_{i_0}, X^\top W_{*,j_0}\rangle - \langle f(Y)_{i_0}, Y^\top W_{*,j_0}\rangle|$$
$$\leq |\langle f(X)_{i_0}, X^\top W_{*,j_0}\rangle - \langle f(X)_{i_0}, Y^\top W_{*,j_0}\rangle|$$
$$+ |\langle f(X)_{i_0}, Y^\top W_{*,j_0}\rangle - \langle f(Y)_{i_0}, Y^\top W_{*,j_0}\rangle|$$
$$\leq \|f(X)_{i_0}\|_2 \cdot \|X - Y\| \cdot \|W_{*,j_0}\|_2 + \|f(X)_{i_0} - f(Y)_{i_0}\| \cdot \|Y\| \cdot \|W_{*,j_0}\|$$
$$\leq R \cdot \|X - Y\| + R^2\|f(X)_{i_0} - f(Y)_{i_0}\|$$
$$\leq 5R^4 \cdot \|X - Y\|$$

where the first step is from the definition of $w(X)_{i_0,j_0}$, the 2nd step is by Fact B.1, the 3rd step holds since Assumption G.1, the 4th step uses Lemma H.3. □

### H.7   LIPSCHITZ FOR FIRST ORDER DERIVATIVE OF $c(X)_{i_0,j_0}$

**Lemma H.8.** *Under following conditions*

- *Assumption G.1 holds*

- *Let $c(X)_{i_0,j_0}$ be defined as Definition B.8*

*For $X, Y \in \mathbb{R}^{d \times n}$, we have*

$$|\frac{c(X)_{i_0,j_0}}{\mathrm{d}x_{i_1,j_1}} - \frac{c(Y)_{i_0,j_0}}{\mathrm{d}y_{i_1,j_1}}| \leq 320R^6 \cdot \|X - Y\|$$

*Proof.* Notice that

- $\max_{i \in [5]}\{\|C_i(X)\|\} \le R^4$
- $\max_{f \in S}\{\text{Lipschitz}(f)\} = 4R^2$ where $S$ is the set of basic functions in $C_i(X)$
- $n = 4$

The result is directly given by applying Fact H.2. $\quad\square$

### H.8 LIPSCHITZ FOR SECOND ORDER DERIVATIVE OF $c(X)_{i_0,j_0}$

**Lemma H.9.** *Under following conditions*

- *Assumption G.1 holds*
- *Let $c(X)_{i_0,j_0}$ be defined as Definition B.8*

*For $X, Y \in \mathbb{R}^{d \times n}$, we have*

$$|\frac{c(X)_{i_0,j_0}}{\mathrm{d}x_{i_1,j_1}x_{i_2,j_2}} - \frac{c(Y)_{i_0,j_0}}{\mathrm{d}y_{i_1,j_1}y_{i_2,j_2}}| \le 2688 \cdot R^8 \|X - Y\|$$

*Proof.* Notice that

- $\max_{i \in [21]}\{\|D_i(X)\|\} \le R^6$
- $\max_{f \in S}\{\text{Lipschitz}(f)\} = 4R^2$ where $S$ is the set of basic functions in $D_i(X)$
- $n = 6$

The result is directly given by applying Fact H.2. $\quad\square$

### H.9 LIPSCHITZ FOR HESSIAN OF $L(X)$

**Lemma H.10.** *Under following conditions*

- *Assumption G.1 holds*
- *Let $c(X)_{i_0,j_0}$ be defined as Definition B.8*

*For $X, Y \in \mathbb{R}^{d \times n}$, we have*

$$\|\nabla^2 L(X) - \nabla^2 L(Y)\| \le O(n^3 d^3 R^{10})\|X - Y\|$$

*Proof.*

$$|\frac{L(X)}{\mathrm{d}x_{i_1,j_1}x_{i_2,j_2}} - \frac{L(Y)}{\mathrm{d}y_{i_1,j_1}y_{i_2,j_2}}|$$

$$= |\sum_{i_0=1}^{n}\sum_{j_0=1}^{d} \frac{\mathrm{d}c(X)_{i_0,j_0}}{\mathrm{d}x_{i_1,j_1}} \cdot \frac{\mathrm{d}c(X)_{i_0,j_0}}{\mathrm{d}x_{i_1,j_2}} + c(X)_{i_0,j_0} \cdot \frac{\mathrm{d}c(X)_{i_0,j_0}}{\mathrm{d}x_{i_1,j_1}x_{i_2,j_2}}$$

$$- \sum_{i_0=1}^{n}\sum_{j_0=1}^{d} \frac{\mathrm{d}c(Y)_{i_0,j_0}}{\mathrm{d}y_{i_1,j_1}} \cdot \frac{\mathrm{d}c(Y)_{i_0,j_0}}{\mathrm{d}y_{i_1,j_2}} + c(Y)_{i_0,j_0} \cdot \frac{\mathrm{d}c(Y)_{i_0,j_0}}{\mathrm{d}y_{i_1,j_1}y_{i_2,j_2}}|$$

$$\le \sum_{i_0=1}^{n}\sum_{j_0=1}^{d}(|\frac{\mathrm{d}c(X)_{i_0,j_0}}{\mathrm{d}x_{i_1,j_1}} \cdot \frac{\mathrm{d}c(X)_{i_0,j_0}}{\mathrm{d}x_{i_1,j_2}} - \frac{\mathrm{d}c(X)_{i_0,j_0}}{\mathrm{d}x_{i_1,j_1}} \cdot \frac{\mathrm{d}c(Y)_{i_0,j_0}}{\mathrm{d}y_{i_1,j_2}}|$$

$$+ \left| \frac{\mathrm{d}c(X)_{i_0,j_0}}{\mathrm{d}x_{i_1,j_1}} \cdot \frac{\mathrm{d}c(Y)_{i_0,j_0}}{\mathrm{d}y_{i_1,j_2}} - \frac{\mathrm{d}c(Y)_{i_0,j_0}}{\mathrm{d}y_{i_1,j_1}} \cdot \frac{\mathrm{d}c(Y)_{i_0,j_0}}{\mathrm{d}y_{i_1,j_2}} \right|$$

$$+ \left| c(X)_{i_0,j_0} \cdot \frac{\mathrm{d}c(X)_{i_0,j_0}}{\mathrm{d}x_{i_1,j_1} x_{i_2,j_2}} - c(Y)_{i_0,j_0} \cdot \frac{\mathrm{d}c(Y)_{i_0,j_0}}{\mathrm{d}y_{i_1,j_1} y_{i_2,j_2}} \right|$$

$$+ \left| c(X)_{i_0,j_0} \cdot \frac{\mathrm{d}c(Y)_{i_0,j_0}}{\mathrm{d}y_{i_1,j_1} y_{i_2,j_2}} - c(Y)_{i_0,j_0} \cdot \frac{\mathrm{d}c(Y)_{i_0,j_0}}{\mathrm{d}y_{i_1,j_1} y_{i_2,j_2}} \right| \Big)$$

$$= \sum_{i_0=1}^{n} \sum_{j_0=1}^{d} \Big( \left| \frac{\mathrm{d}c(X)_{i_0,j_0}}{\mathrm{d}x_{i_1,j_1}} \right| \cdot \left| \frac{\mathrm{d}c(X)_{i_0,j_0}}{\mathrm{d}x_{i_1,j_2}} - \frac{\mathrm{d}c(Y)_{i_0,j_0}}{\mathrm{d}y_{i_1,j_2}} \right|$$

$$+ \left| \frac{\mathrm{d}c(X)_{i_0,j_0}}{\mathrm{d}x_{i_1,j_1}} \cdot - \frac{\mathrm{d}c(Y)_{i_0,j_0}}{\mathrm{d}y_{i_1,j_1}} \right| \cdot \left| \frac{\mathrm{d}c(Y)_{i_0,j_0}}{\mathrm{d}y_{i_1,j_2}} \right|$$

$$+ |c(X)_{i_0,j_0}| \cdot \left| \frac{\mathrm{d}c(X)_{i_0,j_0}}{\mathrm{d}x_{i_1,j_1} x_{i_2,j_2}} - \frac{\mathrm{d}c(Y)_{i_0,j_0}}{\mathrm{d}y_{i_1,j_1} y_{i_2,j_2}} \right|$$

$$+ |c(X)_{i_0,j_0} - c(Y)_{i_0,j_0}| \cdot \left| \frac{\mathrm{d}c(Y)_{i_0,j_0}}{\mathrm{d}y_{i_1,j_1} y_{i_2,j_2}} \right| \Big)$$

$$\leq \sum_{i_0=1}^{n} \sum_{j_0=1}^{d} \Big( 10R^4 \cdot \left| \frac{\mathrm{d}c(X)_{i_0,j_0}}{\mathrm{d}x_{i_1,j_2}} - \frac{\mathrm{d}c(Y)_{i_0,j_0}}{\mathrm{d}y_{i_1,j_2}} \right|$$

$$+ |c(X)_{i_0,j_0}| \cdot \left| \frac{\mathrm{d}c(X)_{i_0,j_0}}{\mathrm{d}x_{i_1,j_1} x_{i_2,j_2}} - \frac{\mathrm{d}c(Y)_{i_0,j_0}}{\mathrm{d}y_{i_1,j_1} y_{i_2,j_2}} \right|$$

$$+ |c(X)_{i_0,j_0} - c(Y)_{i_0,j_0}| \cdot \left| \frac{\mathrm{d}c(Y)_{i_0,j_0}}{\mathrm{d}y_{i_1,j_1} y_{i_2,j_2}} \right| \Big)$$

$$\leq \sum_{i_0=1}^{n} \sum_{j_0=1}^{d} \Big( 3200R^{10} \cdot \|X - Y\|$$

$$+ |c(X)_{i_0,j_0}| \cdot \left| \frac{\mathrm{d}c(X)_{i_0,j_0}}{\mathrm{d}x_{i_1,j_1} x_{i_2,j_2}} - \frac{\mathrm{d}c(Y)_{i_0,j_0}}{\mathrm{d}y_{i_1,j_1} y_{i_2,j_2}} \right|$$

$$+ |c(X)_{i_0,j_0} - c(Y)_{i_0,j_0}| \cdot \left| \frac{\mathrm{d}c(Y)_{i_0,j_0}}{\mathrm{d}y_{i_1,j_1} y_{i_2,j_2}} \right| \Big)$$

$$\leq \sum_{i_0=1}^{n} \sum_{j_0=1}^{d} \Big( 3200R^{10} \cdot \|X - Y\|$$

$$+ 2R^2 \cdot \left| \frac{\mathrm{d}c(X)_{i_0,j_0}}{\mathrm{d}x_{i_1,j_1} x_{i_2,j_2}} - \frac{\mathrm{d}c(Y)_{i_0,j_0}}{\mathrm{d}y_{i_1,j_1} y_{i_2,j_2}} \right|$$

$$+ |c(X)_{i_0,j_0} - c(Y)_{i_0,j_0}| \cdot \left| \frac{\mathrm{d}c(Y)_{i_0,j_0}}{\mathrm{d}y_{i_1,j_1} y_{i_2,j_2}} \right| \Big)$$

$$\leq \sum_{i_0=1}^{n} \sum_{j_0=1}^{d} \Big( 3200R^{10} \cdot \|X - Y\|$$

$$+ 5376R^{10} \cdot \|X - Y\|$$

$$+ 23R^6 \cdot |c(X)_{i_0,j_0} - c(Y)_{i_0,j_0}| \Big)$$

$$\leq \sum_{i_0=1}^{n} \sum_{j_0=1}^{d} \Big( 3200R^{10} \cdot \|X - Y\|$$

$$+ 5376R^{10} \cdot \|X - Y\|$$

$$+ 105R^{10} \cdot \|X - Y\| \Big)$$

$$= 8681ndR^{10} \cdot \|X - Y\|$$

where the first step is by Lemma F.1, the 2nd step is by triangle inequality, the 3rd step is basic algebra, the 4th step uses Lemma G.4, the 5th step uses Lemma H.8, the 6th step uses Lemma G.2, the 7th step uses Lemma H.9, the 8th step uses Lemma H.4.

Then, we have

$$\|\nabla^2 L(X) - \nabla^2 L(Y)\| \leq \|\nabla^2 L(X) - \nabla^2 L(Y)\|_F$$
$$\leq n^2 d^2 \cdot 9193nd R^{10}\|X - Y\| = 9193 n^3 d^3 R^{10} \cdot \|X - Y\|$$

where the 1st step is by matrix calculus, the 2nd is by above result. $\square$

## I  STRONGLY CONVEXITY

In this section, we provide proof for PSD bounds for $H_i$.

### I.1  PSD BOUNDS FOR $H_i$

**Lemma I.1** (PSD bounds for $H_i$). *Under following conditions,*

- *Let $H_i^{(i_1, i_2)}$ be defined as in Definition E.5*
- *Let Assumption G.1 be satisfied*

*For all $i \in [4]$ (i.e., the 4 cases), we have*

$$-21R^6 \cdot \mathbf{I}_d \preceq H_i^{(i_1, i_2)} \preceq 21R^6 \cdot \mathbf{I}_d$$

*Proof.* Considering $H_1^{(i_0, i_0)}$, we have

$$-21R^6 \cdot \mathbf{I}_n \preceq H_1^{(i_0, i_0)} \preceq 21R^6 \cdot \mathbf{I}_n$$

This is given by the upper bound of $B_i(X)$ is smaller than $R^6$ for all $i \in [21]$.

Notice that in other cases, the Hessian has less terms than **Case 1**. Also, those terms are included in **Case 1** (equivalence by changing coordinates). Therefore, the PSD bound is suited for all cases. $\square$

**Lemma I.2** (PSD bounds for $\nabla^2 c(X)_{i_0, j_0}$). *Under following conditions,*

- *Let $c_{i_0, j_0}$ be defined as in Definition B.8*
- *Let Assumption G.1 be satisfied*

*For all $i_0 \in [n], j_0 \in [d]$, we have*

$$-21R^6 \cdot \mathbf{I}_{nd} \preceq \nabla^2 c(X)_{i_0, j_0} \preceq 21R^6 \cdot \mathbf{I}_{nd}$$

*Proof.* Recall in Definition E.3, each partition component of $\nabla^2 c(X)_{i_0, j_0}$ is $H_i^{(i_1, i_2)}$, which has PSD bounds as in Lemma I.1

Then by the definition of PSD, $\nabla^2 c(X)_{i_0, j_0}$ has the same PSD bounds. $\square$

**Lemma I.3** (PSD bounds for $\nabla^2 L(X)$). *Under following conditions,*

- *Let $L(X)$ be defined as in Definition B.9*
- *Let Assumption G.1 be satisfied*

*we have*

$$\nabla^2 L(X) \succeq -O(nd R^8) \cdot \mathbf{I}_{nd}$$

*Proof.* Recall in Lemma F.2, we have

$$\nabla^2 L(X) = \sum_{i_0=1}^{n} \sum_{j_0=1}^{d} \nabla c(X)_{i_0,j_0} \cdot \nabla c(X)_{i_0,j_0}^{\top} + c(X)_{i_0,j_0} \cdot \nabla^2 c(X)_{i_0,j_0} \tag{2}$$

Notice that the first term is PSD, so we omit it.

By Lemma G.2, we have

$$|c(X)_{i_0,j_0}| \leq 2R^2$$

Therefore, we have

$$\nabla^2 L(X) \succeq -42ndR^8 \cdot \mathbf{I}_{nd}$$

which is given by combining Lemma I.2 and Eq. (2).

$\square$

## J   FINAL RESULT

**Theorem J.1** (Formal of Theorem 1.3, Main Result). *We assume our model satisfies the following conditions*

- *Bounded parameters: there exists $R > 1$ such that*
  - $\|W\|_F \leq R$, $\|V\|_F \leq R$
  - $\|X\|_F \leq R$
  - $\forall i \in [n], j \in [d], |b_{i,j}| \leq R$ where $b_{i,j}$ denotes the $i,j$-th entry of $B$

- *Regularization: we consider the following problem:*
  $$\min_{X \in \mathbb{R}^{n \times d}} \|D(X)^{-1} \exp(X^{\top} W X) X^{\top} V - B\|_F^2 + \gamma \cdot \|\operatorname{vec}(X)\|_2^2$$

- *Good initial point: We choose an initial point $X_0$ such that $M \cdot \|X_0 - X^*\|_F \leq 0.1l$, where $M = O(n^3 d^3 R^{10})$*

*Then, for any accuracy parameter $\epsilon \in (0, 0.1)$ and a failure probability $\delta \in (0, 0.1)$, an algorithm based on the Newton method can be employed to recover the initial data. The result of this algorithm guarantee within $T = O(\log(|X_0 - X^*|_F/\epsilon))$ executions, it outputs a matrix $\widetilde{X} \in \mathbb{R}^{d \times n}$ satisfying $\|\widetilde{X} - X^*\|_F \leq \epsilon$ with a probability of at least $1 - \delta$. The execution time for each iteration is $\operatorname{poly}(n, d)$.*

*Proof.* Choosing $\gamma \geq O(ndR^8)$, by Lemma I.3, we have the PD property of Hessian.

By Lemma H.10, we have the Lipschitz property of Hessian.

Since $M$ is bounded (in the condition of Theorem), then by iterative shrinking lemma (see Lemma 6.9 in Li et al. (2023c) as an example), we prove the convergence. $\square$