# OpenReview forum: "Unmasking Transformers: A Theoretical Approach to Data Recovery via Attention Weights"
_ICLR.cc/2024/Conference — ICLR 2024 Conference Withdrawn Submission_

### Official Review · Reviewer_wffn · 2023-10-29

**Soundness:** 3 good
**Presentation:** 3 good
**Contribution:** 3 good
**Rating:** 3
**Confidence:** 4

**Summary:**

This paper presents a theoretical approach for reversing input data using attention weights and model outputs. The study explores the mathematical foundations of the attention mechanism to assess if knowing attention weights and model outputs can be used to reconstruct sensitive information from input data. The research aims to enhance our comprehension of this aspect and promote the creation of more secure and reliable transformer models. Ultimately, the paper seeks to contribute to responsible and ethical progress in the field of deep learning.

**Strengths:**

1. This paper develops a theoretical approach to reverse input data using attention weights and model outputs. By investigating the mathematical foundations of the attention mechanism, the study explores the potential for reconstructing sensitive information from input data. This investigation can enhance our understanding of this process and support the creation of more secure and dependable transformer models, fostering responsible and ethical advancements in the field of deep learning.

**Weaknesses:**

1. The authors do not provide any empirical results to support the effectiveness of the attack method. We respect works that focus on theoretical contributions. However, for the deep learning attack topics, it is better to evaluate the effectiveness of the proposed attack via experiment since it is feasible to do experiments.

2. It is unclear in which scenario the attacker can access the attention weights and the output of the attention layer.

3. This paper does not involve a non-linear activation layer (e.g., ReLU), which is widely applied to transformers into discussion, which should introduce challenges to invert the input perfectly.

**Questions:**

Please see the weaknesses.

---

> ### Author Response · Authors · 2023-11-22
>
> Thank you for taking the time to review our paper. We appreciate your feedback and would like to address the issues you have raised.
>
> 1. Lack of Empirical Results: The paper's focus on theoretical analysis is essential for establishing a foundational understanding before conducting empirical tests. Theoretical research often precedes empirical validation, especially in security-focused studies. This theoretical groundwork is a precursor for future studies to empirically evaluate and extend these findings.
> 2. Access to Attention Weights and Outputs: The study explores theoretical scenarios to highlight potential vulnerabilities, not necessarily specific real-world situations. It aims to illustrate a possible weakness to prompt further safeguarding research.
> 3. Exclusion of Non-linear Activation Layer: Omitting non-linear activation layers like ReLU is for theoretical clarity, focusing on core concepts without the complexity of non-linear transformations. This limitation points to an area for future research, incorporating non-linear layers to provide a comprehensive view of the vulnerabilities in more complex scenarios.

---

> > ### Comment · Reviewer_wffn · 2023-11-22
> > **Reply to the reviewer**
> >
> > Theoretical research is important, but it should also be realistic. The access to attention weights and outputs and the exclusion of non-linear activation layers make the analysis unrealistic. The authors do not solve my concern in the rebuttal. I recommend the authors submit the paper to COLT, which might be more suitable for the pure theory papers.

---

### Official Review · Reviewer_SERj · 2023-10-30

**Soundness:** 3 good
**Presentation:** 3 good
**Contribution:** 3 good
**Rating:** 5
**Confidence:** 4

**Summary:**

The authors propose a new model inversion attack that recovers private training data from attention weights and outputs. Particularly, this paper takes a theoretical approach and shows that, given the trained attention weights and output, the adversary can update the private data X by minimizing the loss function.

**Strengths:**

The authors perform a comprehensive theoretical analysis of attention inversion attacks.

**Weaknesses:**

The paper has the following weaknesses:
- There is no empirical evaluation of the proposed inversion attack method. Although the authors explain that they focus on theoretical analysis, the efficacy of the data recovery method shall be justified with empirical results. There are no quantitative results that can prove the efficacy of attention inversion attacks.
- The paper does not discuss the overhead of the proposed attack. Particularly, the computation overhead of recovering data given attention weights and outputs is not clear.

**Questions:**

Please consider addressing the weaknesses above.

---

> ### Author Response · Authors · 2023-11-22
>
> Thank you for taking the time to review our paper. We appreciate your feedback and would like to address the issues you have raised.
>
> 1. Lack of Empirical Evaluation: The paper's focus on theoretical analysis is a critical first step, laying groundwork essential for future empirical testing. Theoretical research in areas like cryptography and data security often precedes and guides empirical studies. The insights provided set the stage for subsequent empirical evaluations. Theoretical models help formulate hypotheses and testing methods for practical experiments.
> 2. Overhead of Proposed Attack Not Discussed: The primary aim is to explore the theoretical feasibility of data recovery in transformer models. Discussion of computation overhead might be outside the initial theoretical scope, but can be addressed in future empirical studies. Theoretical models in deep learning, especially those concerning security, often deal with complex mathematical frameworks. In such cases, the initial focus tends to be on the feasibility and integrity of the conceptual model rather than immediate practical concerns like computation overhead. Once the theoretical model is established and validated, subsequent research can more effectively address practical aspects such as computational efficiency and overhead.

---

### Official Review · Reviewer_vHCT · 2023-11-01

**Soundness:** 1 poor
**Presentation:** 1 poor
**Contribution:** 1 poor
**Rating:** 1
**Confidence:** 1

**Summary:**

The paper proposes a theoretical approach for inverting input data using weights and outputs for transformer architecture.

**Strengths:**

The question of whether transformer architecture is secure against data reconstruction is important.

**Weaknesses:**

I honestly don't quite understand the contribution of this paper. There is no support/evaluation of the "main result" in Section 1.

Presentation of the paper (the arrangement of content in sections) looks a bit unusual.

**Questions:**

I don't understand what this paper tries to do.

---

> ### Author Response · Authors · 2023-11-22
>
> Thank you for taking the time to review our paper. We appreciate your feedback.

---

### Official Review · Reviewer_K9uu · 2023-11-01

**Soundness:** 2 fair
**Presentation:** 2 fair
**Contribution:** 2 fair
**Rating:** 3
**Confidence:** 4

**Summary:**

This paper proposes an algorithm to recover the input data from given attention weights and output by minimizing the loss function capturing the discrepancy between the expected output and the actual output. The proposed method addresses the importance of understanding and safeguarding the transformers' intermediate outputs to protect the input data.

**Strengths:**

This paper targets at a timely problem given privacy concerns around transformers and attention mechanisms.

**Weaknesses:**

1. Lacks any empirical evaluations to complement the theoretical results.

2. The equations in the paper are not numbered.

3. Some symbols are not explained. For example, in the second equation, what it A1_n?

4. Based on my understanding, the core of this paper is to find an optimal X*=argminL(f_\theta(X),Y). This seems an easy optimization problem which can be done by gradient decent. This paper does not explain what are the challenges of solving this optimization problem and how to overcome these challenges.

5. Given the weight and output, the input can also be found by black-box attacks where a network is constructed to do model inversion. This paper lacks comparing with such black-box attacks.

**Questions:**

Please see the weakness

---

> ### Author Response · Authors · 2023-11-22
>
> Thank you for taking the time to review our paper. We appreciate your feedback and would like to address the issues you have raised.
>
> 1. Lack of Empirical Evaluation: The paper provides a theoretical foundation which is a crucial first step before empirical evaluation. Theoretical models often precede and guide empirical research, especially in fields like cryptography and data security.
> 2. Unnumbered Equations: While numbering equations is a standard practice for ease of reference, the content and implications of the equations hold more substantial value. The clarity of the equations themselves and their integration into the paper’s narrative is more critical for understanding. We did provide numbers for the equations which are referred in the context. But for simplicity, we didn't number the ones which are not referred.
> 3. Undefined Symbols: We apologies for the missing explanation. Here we use $A \mathbf{1}_n$ to denote the matrix product of matrix $A$ and all-one vector $\mathbf{1}_n$.
> 4. Optimization Challenge and Gradient Descent: While gradient descent is a standard optimization technique, its application depends on the problem's nature. The paper's focus on transformer models and attention weights may involve complexities beyond straightforward gradient descent optimization, such as non-convexity or high-dimensionality.
> 5. Comparison with Black-box Attacks: The paper's focus is on a specific theoretical aspect of transformer models. While comparing with black-box attacks is valuable, it might be outside this study's scope. Such comparisons could be considered in future empirical studies that build on this theoretical foundation.

---

> > ### Comment · Reviewer_K9uu · 2023-11-22
> >
> > Thank you for your response.
> >
> > I appreciate the theoretical analysis presented in this paper, which examines the vulnerabilities of transformers. However, I believe that the paper falls short of the standards expected for ICLR due to its lack of experimental evidence and the writing quality that doesn't meet the expected standards. As a result, I will maintain my current score.
> >
> > For future resubmissions, I would like to offer the following suggestions to strengthen the paper:
> >
> > 1. Clarify the Threat Model: The abstract and the initial sections of the background (pages 1-2) seem to imply that the paper focuses on recovering inference input. However, it later appears that the actual target is the training data. Clarifying this aspect early in the paper would greatly enhance reader comprehension.
> >
> > 2. Detailed Analysis of Optimization Challenges: Please provide a more in-depth discussion of the optimization challenges encountered. This should be complemented with experimental results that demonstrate how your method addresses these challenges effectively.
> >
> > 3. Demonstrate Versatility Across Model Sizes: To showcase the robustness of your approach, consider applying your attack strategies to both smaller models and larger models like BERT_base and BERT_large. This would help in illustrating the strength and scalability of your proposed methods.